# Targeting G protein-coupled receptor signaling at the G protein level with a selective nanobody inhibitor

Sahil Gulati [1,2], Hui Jin[1], Ikuo Masuho[3], Tivadar Orban[1], Yuan Cai[4,5], Els Pardon [6,7], Kirill A. Martemyanov[3], Philip D. Kiser[1,8], Phoebe L. Stewart[1,2], Christopher P. Ford[5], Jan Steyaert [6,7] & Krzysztof Palczewski[1,2]

G protein-coupled receptors (GPCRs) activate heterotrimeric G proteins by mediating a GDP to GTP exchange in the Gα subunit. This leads to dissociation of the heterotrimer into Gα-GTP and Gβγ dimer. The Gα-GTP and Gβγ dimer each regulate a variety of downstream pathways to control various aspects of human physiology. Dysregulated Gβγ-signaling is a central element of various neurological and cancer-related anomalies. However, Gβγ also serves as a negative regulator of Gα that is essential for G protein inactivation, and thus has the potential for numerous side effects when targeted therapeutically. Here we report a llama-derived nanobody (Nb5) that binds tightly to the Gβγ dimer. Nb5 responds to all combinations of β-subtypes and γ-subtypes and competes with other Gβγ-regulatory proteins for a common binding site on the Gβγ dimer. Despite its inhibitory effect on Gβγ-mediated signaling, Nb5 has no effect on $G\alpha_q$-mediated and $G\alpha_s$-mediated signaling events in living cells.

[1] Department of Pharmacology, School of Medicine, Case Western Reserve University, 10900 Euclid Avenue, Cleveland, OH 44106, USA. [2] Cleveland Center for Membrane and Structural Biology, Case Western Reserve University, 1819 East 101st Street, Cleveland, OH 44106, USA. [3] Department of Neuroscience, The Scripps Research Institute, 130 Scripps Way, Jupiter, FL 33458, USA. [4] Department of Physiology and Biophysics, School of Medicine, Case Western Reserve University, 10900 Euclid Ave, Cleveland, OH 44106, USA. [5] Department of Pharmacology, University of Colorado School of Medicine, 12800 East 19th Ave, Aurora, CO 80045, USA. [6] Structural Biology Brussels, Vrije Universiteit Brussel (VUB), Brussels 1050, Belgium. [7] VIB-VUB Center for Structural Biology, VIB, Brussels 1050, Belgium. [8] Research Service, Louis Stokes Cleveland VA Medical Center, Cleveland, OH 44106, USA. Correspondence and requests for materials should be addressed to K.P. (email: kxp65@case.edu)

Gprotein-coupled receptors (GPCRs) function by translating extracellular stimuli across the plasma membrane into intracellular signaling events[1]. The latter are accomplished by a ligand-induced conformational change in the GPCR that activates a downstream heterotrimeric G protein[2]. Heterotrimeric G proteins, consisting of three subunits, Gα, Gβ and Gγ, undergo a Gα-GDP/GTP exchange that leads to dissociation of the Gβγ dimer from the heterotrimer[3]. Both these moieties then become free to act upon their downstream effector molecules and thereby initiate a variety of intracellular signaling events that constitute the overall GPCR-mediated response. Notably, heterotrimeric G proteins maintain an equilibrium between their heterotrimeric and dissociated states by undergoing spontaneous GPCR-independent association/dissociation[4].

While GPCRs serve as the largest class of drug-targeted membrane proteins, producing the majority of FDA-approved drugs available on the market, only a handful of GPCR-mediated signaling events have been targeted therapeutically with either small molecule or peptide modulators[5]. Such modulators can be classified as either agonists, antagonists, or inverse agonists, depending on their ability to either stabilize the activated state of receptors, inhibit agonist competitively, or reduce the basal spontaneous coupling to G proteins, respectively[6]. Some biased signaling ligands stabilize an intermediate receptor conformation that mimics the active conformation with respect to one signaling pathway, while simultaneously mimicking the inactive conformation for another signaling pathway[7]. However, drug development for GPCR signaling pathways has been hampered by difficulties in identifying molecules with suitable selectivity[8]. Related GPCRs share a promiscuous ligand-binding site for lipophilic ligands, which allows several off-target effects when pursued therapeutically. Additionally, lipophilic ligands can penetrate the central nervous system (CNS), and thereby interact with undesirable neuronal receptor targets. As an alternative strategy, functional monoclonal antibodies (mAbs) are currently being used to target less well-conserved allosteric sites of GPCRs to selectively modulate their signaling[9]. Although mAbs and their fragments have played an instrumental role in selectively targeting GPCR signaling pathways, their commercial success is limited due to their high costs of production and treatment, especially when their efficacy in prolonging survival is similar to that of less expensive alternatives.

Antibody alternatives are known to be versatile tools for exploring the determinants of GPCR recognition. However, nanobodies (Nbs) derived from the variable region of camelid heavy chain are endowed with favorable characteristics in terms of size, solubility, affinity, and ease of production[10]. Nbs are single-domain entities which form the smallest antigen binding fragment that completely retains the binding affinity and specificity of a full-length antibody[10]. Nbs possess exceptionally long complementarity-determining region 3 (CDR3) loops and a convex paratope, which allow them to penetrate into hidden cavities of target antigens[11]. These unique biochemical and biophysical properties of Nbs render them superior to conventional mAbs or antibody fragments, and ideally suit them for a myriad of biotechnological applications. In addition to their use in GPCR structural studies, therapeutic Nbs are being discovered that target GPCR signaling[12]. A biparatopic nanobody targeting different binding sites on the chemokine receptor CXCR2 recently entered phase I studies for therapy of inflammation[13]. Additionally, Nbs targeting chemokine receptor CXCR4 were found to inhibit chemotaxis and HIV-1 replication in vivo[12]. vNARs[14], the variable domain of shark-derived new antigen receptors and their human equivalents, i-bodies[15] derived from the I-set immunoglobulin superfamily are other examples of single-domain antibody-like molecules with biochemical and biophysical

properties similar to Nbs. Recently, i-bodies that selectively block CXCR4 β-arrestin recruitment were reported[15]. With the exception of β-arrestin specific antibody fragments[16], all small-molecule-based, antibody-based, and nanobody-based approaches developed to date target GPCR-mediated signaling at the GPCR level[12], and thereby are GPCR-specific and cannot be used universally. Additionally, these approaches activate both Gα-GTP-mediated and Gβγ-mediated signaling pathways that lead to the activation of undesired cellular signaling events.

The liberated Gβγ dimer is a very efficient signal transducer[17–21] and can dysregulate various cellular functions leading to numerous side effects associated with dysregulated ion channels, phosphoinositide-3-kinase (PI3K), adenylyl cyclase (AC) and mitogen-activated protein kinase (MAPK) pathways. These diverse Gβγ functions make it an attractive target for the treatment of many medical conditions[22]. However, the ability of Gβγ to play essential roles in various cellular functions, including the formation of heterotrimeric G proteins, necessitates highly-specific Gβγ inhibitors that modulate its function while keeping Gα-mediated signaling intact. Here we describe a nanobody (Nb5) that specifically binds to the Gβγ dimer and shifts the association/dissociation equilibrium of the heterotrimeric visual G protein ($G_t$) towards dissociated $Gα_t$ and Nb5-bound $Gβ_1γ_1$ subunits. Differential hydrogen/deuterium exchange (HDX) and crystallography studies suggest a competition between Nb5 and other Gβγ-regulatory proteins for a common binding site on the Gβγ dimer. Interestingly, Nb5 binds and inhibits Gβ subtypes 1–4, showing its broad applicability to various cell types. Whole-cell patch clamping of striatum neurons demonstrates an inhibitory effect of Nb5 on the activation of Gβγ-regulated G-protein-gated inward rectifier potassium (GIRK) channels. Finally, Nb5 acts as a control-switch of GPCR-mediated cellular signaling by modulating PI3K-protein kinase B/AKT and MAPK-extracellular signal-regulated kinase (ERK) pathways. However, Nb5 does not alter $Gα_i$-GTP-governed cellular levels of secondary messenger cyclic adenosine monophosphate (cAMP). Additionally, Nb5 shows no effect on either $Gα_q$-induced intracellular $Ca^{2+}$ elevation or $Gα_s$-induced cAMP production. This is the first demonstration of a selective inhibitory effect of a nanobody on Gβγ-mediated signaling events that does not affect Gα signaling on a cellular level. While serving as a rare example wherein a nanobody has an inhibitory role, this study also opens potential avenues for nanobody-mediated Gβγ-signaling modulation to treat various excitatory neurological conditions and block cancer progression.

## Results

**Generation of Nbs directed towards $Gβ_1γ_1$.** A total of 17 different nanobody clones were identified after ELISA selection wherein the wells were coated with purified bovine $Gβ_1γ_1$ dimer. Nbs were divided into 14 families based on their amino acid sequence. All Nbs were produced as soluble His-tagged protein products in the E. coli periplasmic region. Initial binding analyses performed with an immobilized-$Ni^{2+}$ affinity chromatography pull down assay identified 3 Nbs that bound the $Gβ_1γ_1$ dimer. Interestingly, all 3 $Gβ_1γ_1$-positive Nbs belonged to the same nanobody family with an identical complementarity determining region 3 (CDR3) and displayed similar biochemical properties (Supplementary Figure 1). The nanobody with highest expression levels, Nb5, was used for further characterization. Additionally, an irrelevant nanobody, Nb17 was chosen as a negative control due to its non-reactivity with bovine rod outer segments (ROS) proteins.

**Nb5 shifts the dynamic equilibrium of heterotrimeric $G_t$.** The effect of Nb5 on the heterotrimeric state equilibrium of $G_t$ was

determined by assessing its ability to bind and pull out the $\beta_1\gamma_1$ subunits of $G_t$ ($G\beta_1\gamma_1$) from no-salt extracts of ROS. No-salt extracts were chosen over purified $G_t$ to ensure its heterotrimeric configuration as the purification procedure could cause partial dissociation of $G_t$ subunits. As expected, complete $G_t$ heterotrimer dissociation was observed in the presence of light-activated rhodopsin (Rh*) and GTP. As a result, Nb5 bound to the released $G\beta_1\gamma_1$ dimer and the $G\beta_1\gamma_1$-Nb5 complex was obtained in the eluate upon immobilized-Ni$^{2+}$ affinity chromatography purification (Fig. 1a, lane 1 and Fig. 1b). In contrast, the displaced alpha subunit of $G_t$ ($G\alpha_t$) was found in the flow-through (Fig. 1a, lane 2). Interestingly, similar results were obtained in the absence of either Rh* or Rh* and GTP during the purification (Fig. 1a, lanes 3–6). Moreover, an irrelevant nanobody, Nb17 failed to cause dissociation of the $G_t$ heterotrimer (Fig. 1a, compare lanes marked with asterisks). Additionally, no non-specific interactions were seen between $G\beta_1\gamma_1$ and the Ni$^{2+}$-NTA purification resin (Fig. 1a, lanes 9 and 10). These experiments indicate an Nb5-mediated shift in the equilibrium of heterotrimeric $G_t$ in solution towards its dissociated subunits.

Next, the effect of Nb5 on the heterotrimeric state of membrane-bound $G_t$ was investigated. Here, high-salt extracts from dark-adapted ROS were used that contain an excess of free $G\beta_1\gamma_1$ (Fig. 1c, lane 5). As expected, light activation of ROS

resulted in dissociation of heterotrimeric $G_t$ and release of $G\alpha_t$ into the high-salt extracts (Fig. 1c, lane 1). When ROS were washed with high-salt buffer containing a 2 molar excess of Nb5 with or without GTP, a similar release of $G\alpha_t$ was observed. In contrast, high-salt extracts of ROS treated with Nb17 had no effect on either the heterotrimeric state of $G_t$ or the release of $G\alpha_t$ from ROS membranes (Fig. 1c, compare lanes marked with asterisks).

The effect of Nb5 on the $G_t$ activation ability of Rh* was evaluated by monitoring the increase of intrinsic tryptophan fluorescence in $G\alpha_t$ in the presence of Rh* at pH 7.0. Assay conditions were chosen such that the $G_t$ activation rate was the same as that determined by GTP$\gamma$S-induced complex dissociation[23]. A typical elevation of $G\alpha_t$ intrinsic fluorescence was noted upon GTP$\gamma$S-induced complex dissociation with Rh* (Fig. 1d). Interestingly, a 5 min pre-treatment of heterotrimeric $G_t$ with a 2-fold molar excess of Nb5 resulted in a significant decrease in the $G_t$ activation rate (Fig. 1d,e). In contrast, no significant effect was observed upon pre-treatment with Nb17 (Fig. 1d,e). This indicates that Nb5 alone can trap $G\beta_1\gamma_1$ after its spontaneous dissociation from $G\alpha_t$, and thereby reduce the effective heterotrimeric population of $G_t$ during $G_t$ activation assays. Additionally, the observed changes in $G_t$ activation kinetics with Nb5 are consistent with single turnover experiments of Rh* with different

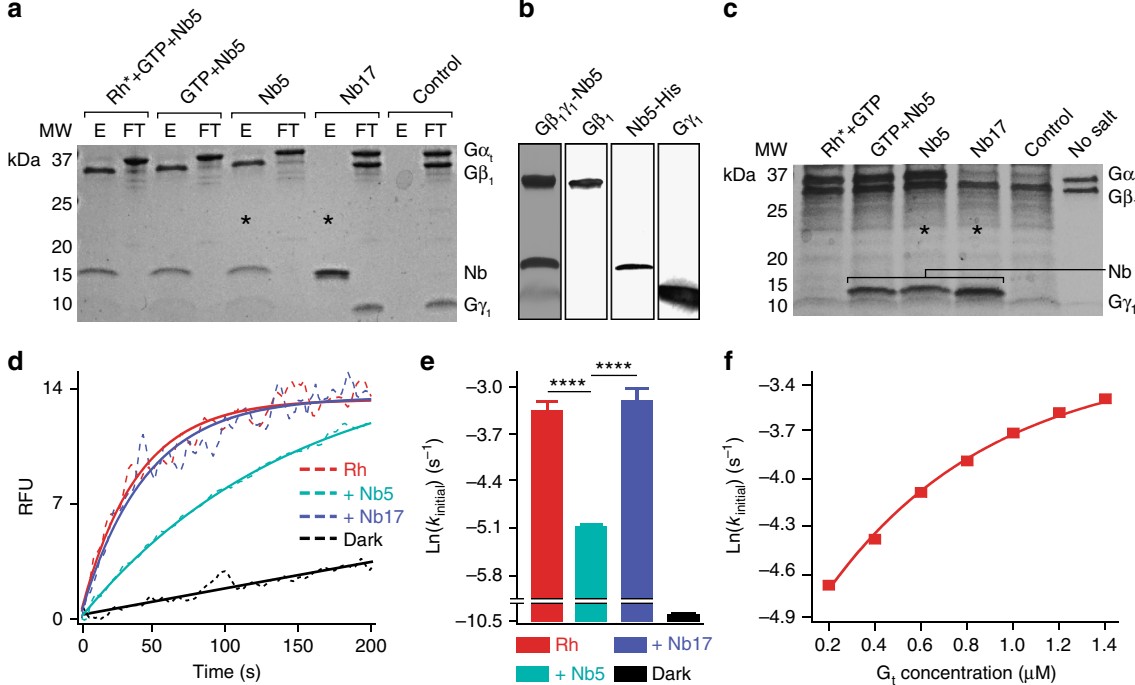

**Fig. 1** Nanobody-mediated shift in heterotrimeric $G_t$ equilibrium. **a** Effect of Nb5 on the heterotrimeric state of $G_t$ in solution. Lane 1 shows the dissociation of heterotrimeric $G_t$ in the presence of light-activated Rh (Rh*), GTP, and Nb5. $G\beta_1\gamma_1$ was later co-purified with Nb5 by immobilized-Ni$^{2+}$ affinity chromatography and $G\alpha_t$ emerged as an unbound protein in the flow-through (lane 2). Note the decrease in $G\gamma_1$ Coomassie staining in the presence of 300 mM imidazole during protein elution. Similar results were obtained from the reaction mixture in the absence of either Rh* alone (lanes 3 and 4) or both Rh* and GTP (lanes 5 and 6). However, Nb17 did not affect the heterotrimeric configuration of $G_t$ (lanes 7 and 8, compare lanes marked with asterisks). Additionally, non-specific interactions between $G\beta_1\gamma_1$ and Ni$^{2+}$-NTA resin were not seen (lanes 9 and 10). The eluate and flow-through obtained from affinity chromatography purification are denoted as E and FT, respectively. **b** Proteins in typical eluates obtained after immobilized-Ni$^{2+}$ affinity chromatography were analyzed by SDS-PAGE (lane 1) and immunoblotting with antibodies specific to $G\beta_1$, His-tag, and $G\gamma_1$, respectively (lanes 2–4). **c** Light activation of ROS membranes in the presence of GTP released $G\alpha_t$ in the high salt wash (lane 1). Similar $G\alpha_t$ release was obtained with Nb5 with or without GTP during the high salt wash (lanes 2 and 3). Treatment of ROS membranes with Nb17 had no effect on $G\alpha_t$ release (lane 4, compare lanes marked with asterisks). Untreated ROS membranes showed no $G\alpha_t$ release during the high salt wash (lane 5). **d** The initial $G_t$ activation rate of Rh* was reduced upon pre-treatment of heterotrimeric $G_t$ with Nb5 (greencyan) as compared to either untreated (red) or Nb17-treated $G_t$ (purple). **e** Quantification of the initial $G_t$ activation rates of Rh* with and without nanobody pre-treatment. Results are expressed as the mean ± SD, $n = 3$ replicates, ****$P < 0.0001$, Student's $t$-test. **f** The decrease in $G_t$ activation rates after Nb5 treatment were explained by Rh single turnover experiments that exhibited an exponential increase in $G_t$ activation rates with increasing concentrations of $G_t$

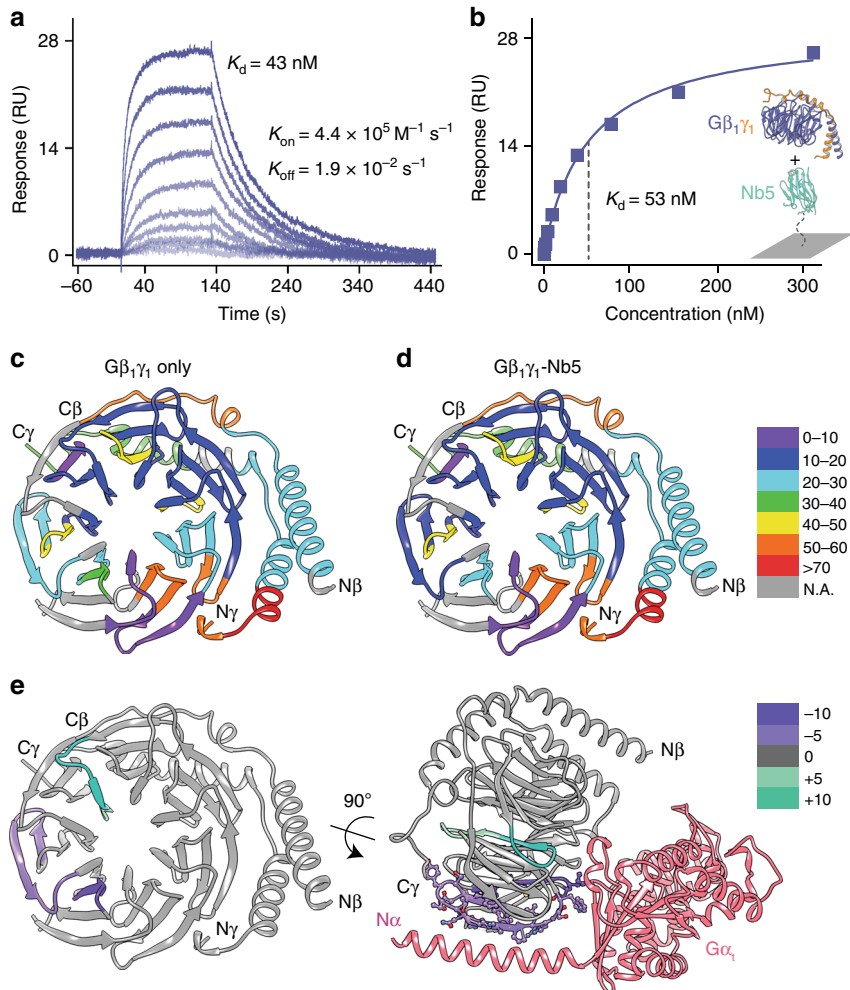

**Fig. 2** Kinetic profiling and hydrogen-deuterium exchange (HDX) of the $G\beta_1\gamma_1$-Nb5 complex. Binding of $G\beta_1\gamma_1$ to immobilized Nb5 in SPR equilibrium binding experiments. Association of the $G\beta_1\gamma_1$ dimer with Nb5 was investigated by single-cycle kinetics (**a**) and affinity based analysis (**b**). Resonance signals are indicated in response units (RU). The determined dissociation constants ($K_d$) and kinetic parameters ($K_{on}$ and $K_{off}$) are shown as insets. HDX of the $G\beta_1\gamma_1$ dimer without (**c**) and with (**d**) Nb5 is shown with all the identified peptides colored by their percentage of deuterium exchange. **e** Differential HDX data mapped into the crystal structure of $G\beta_1\gamma_1$ (PDB ID: 5KDO) indicate peptides with increased (greencyan) and reduced (purple) deuterium incorporation upon Nb5 binding (left). Interestingly, $G\beta_1\gamma_1$ peptides that revealed changes in their solvent accessibility during HDX analyses displayed a partial interface with the $G\alpha$ subunit (right, pink). The $G\alpha$ subunit was omitted from the left panel for clarity

concentrations of heterotrimeric $G_t$, that demonstrate a decrease in $G_t$ activation rates with decreasing $G_t$ concentrations (Fig. 1f). Overall, these results indicate that Nb5 has a high affinity towards the $G\beta_1\gamma_1$ dimer which shifts the dynamic equilibrium of heterotrimeric $G_t$ towards dissociated $G\alpha_t$ and Nb5-bound $G\beta_1\gamma_1$ complex in both solution and membranes.

**Nb5 competes with Gβγ regulatory proteins for Gβγ binding.** To examine the kinetics underlying the assembly of the $G\beta_1\gamma_1$-Nb5 complex, surface plasmon resonance (SPR) was employed with varied concentrations of native bovine $G\beta_1\gamma_1$. The binding kinetics analysis revealed a rapid on-rate constant ($K_{on} = 4.4 \times 10^5 \, M^{-1} \, s^{-1}$) and a slow off-rate constant ($K_{off} = 1.9 \times 10^{-2} \, s^{-1}$) with a $K_d$ of 43 nM (Fig. 2a). Analysis of the protein–protein interactions gave a similar $K_d$ of 53 nM (Fig. 2b). The low-nanomolar affinity of Nb5 for $G\beta_1\gamma_1$ means a possible competition between Nb5 and other $G\beta_1\gamma_1$ regulatory proteins that bind $G\beta_1\gamma_1$ with similar affinities[24,25]. Differential HDX was used to investigate the binding dynamics between $G\beta_1\gamma_1$ and Nb5. Full HDX profiles of $G\beta_1\gamma_1$ alone were compared to those of the

$G\beta_1\gamma_1$-Nb5 complex (Fig. 2c,d). Binding of Nb5 to the $G\beta_1\gamma_1$ dimer induced a statistically significant ($P < 0.01$) increase in protection from solvent exchange in the regions of residues 80–99 and 111–118 of the $G\beta_1$ subunit versus the $G\beta_1\gamma_1$ dimer alone (Supplementary Table 1). In addition, regions with a statistically significant decrease in solvent protection were also observed (Fig. 2e, left). The differential HDX data indicate that the binding interface between Nb5 and the $G\beta_1\gamma_1$ dimer involves at a minimum, residues 80–99 and 111–118 of $G\beta_1$ (Fig. 2e, left). Interestingly, the binding site of Nb5 on the $G\beta_1\gamma_1$ dimer implied by the HDX data closely resembles that of $G\alpha_t$ with the $G\beta_1\gamma_1$ dimer (Fig. 2e, right). This suggests a possible binding interface overlap between Nb5 and $G\alpha_t$ for the $G\beta_1\gamma_1$ dimer.

To further investigate the binding between $G\beta_1\gamma_1$ and Nb5, the complex between $G\beta_1\gamma_1$ and Nb5 was crystallized. The crystal structure of the $G\beta_1\gamma_1$-Nb5 complex refined at 2.34 Å bore a strong resemblance to a heterotrimeric G protein structure. Two molecules of $G\beta_1\gamma_1$-Nb5 complex formed an asymmetric unit (Fig. 3a) wherein most of the crystallographic contacts were mediated by Nb5 (Supplementary Figure 2a). This agrees with a role of nanobodies as crystallization chaperones in the structural

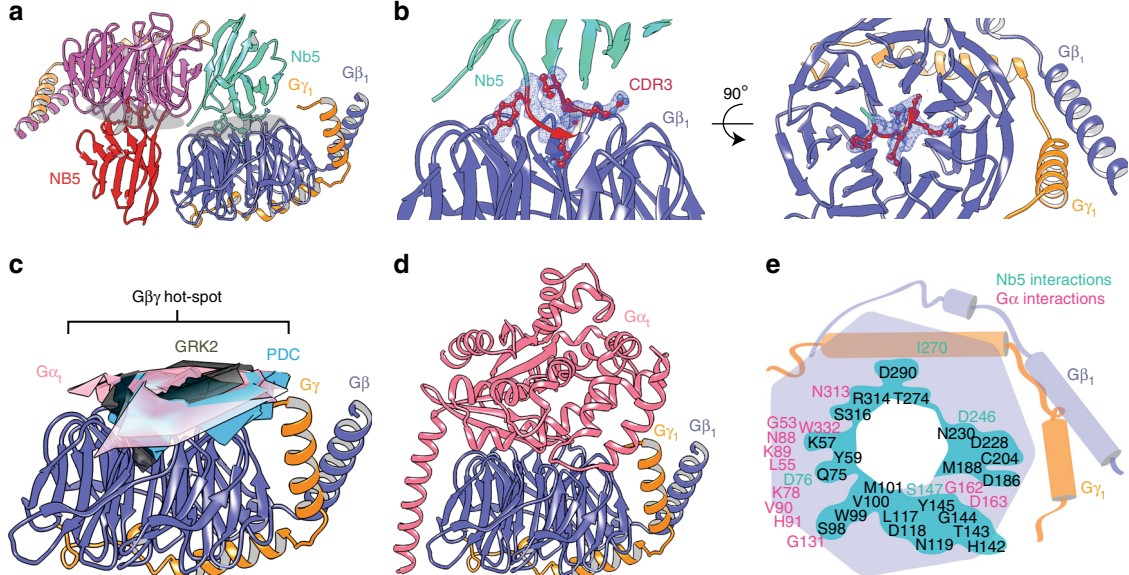

**Fig. 3** Crystal structure of Gβ₁γ₁ in complex with Nb5. **a** The asymmetric unit contains two molecules of both Gβ₁γ₁ (blue and pink) and Nb5 (greencyan and red) shown in cartoon representation. Key interfaces between Gβ₁γ₁ and Nb5 are denoted by grey ellipsoids. **b** Side and top views of the Gβ₁γ₁-Nb5 complex showing the CDR3 region (red) of Nb5 inserted into the Gβ₁-propeller. The 2|$F_o$| - |$F_c$| density (blue) around the CDR3 region was calculated after the final refinement and is contoured at 1.5σ. **c** Gβγ hot-spot is shared by several Gβγ regulatory proteins, including Gα$_t$ (pink), phosducin (PDC, cyan), and G Protein-Coupled Receptor Kinase 2 (GRK2, dark grey). **d** Overall structure of the heterotrimeric G protein (PDB ID: 5KDO). **e** Overlay of the interface between the Gβ₁γ₁-Nb5 complex and heterotrimeric G protein. Schematic representation shows the extent of overlapping (in greencyan background, black residues) between the non-canonical Gβ₁γ₁-Nb5 complex interface (greencyan residues) and the interface between Gβ₁γ₁ and Gα$_t$ (pink residues)

determination of challenging target proteins[26]. An intriguing feature of the Gβ₁γ₁-Nb5 complex is insertion of the CDR3 loop of Nb5 into the Gβ₁-propeller, which occupies most of the binding interface (Fig. 3b). This finding is consistent with the nanomolar affinity observed in the SPR kinetics profiling of the Gβ₁γ₁-Nb5 complex (Fig. 2a, b). Interestingly, Nb5 occupies the same hot-spot region on the Gβ₁γ₁ dimer that is shared by other Gβγ regulatory proteins, including phosducin (PDC) and G protein-coupled receptor kinase 2 (GRK2) (Fig. 3a–c). Also, insertion of the Nb5 CDR3 loop into the Gβ₁-propeller (interface area of 1030 Å²) is reminiscent of the interaction between the C-terminal loop of GRK2 and the Gβ₁γ₂ dimer (interface area of 1080 Å²; PDB accession: 1OMW[27]). Notably, Arg-101 in the Nb5 CDR3 loop serves as a key that locks the Gβ₁-propeller. A similar interaction mechanism is mediated by Lys-663 of GRK2 in the Gβ₁γ₂-GRK2 complex (Supplementary Figure 2b,c). Complexation significance analysis[28] that indicates the significance of a protein assembly formation showed a score of 0.27 for the Gβ₁γ₁-Nb5 complex versus 0.10 for the Gβ₁γ₂-GRK2 complex. The shape complementarity (Sc) index[29] analysis of the binding interface formed by Nb5 and GRK2 with the Gβγ dimer displayed Sc scores of 0.80 and 0.57, respectively (1.0 is a perfect match). Although these results indicate that both Nb5 and GRK2 have a similar mode of binding with the Gβγ dimer, Nb5 is likely to have a structural advantage over GRK2. Similarly, Nb5 had a slight structural advantage over PDC for Gβ₁γ₁ binding (Sc index = 0.73). Interestingly, Arg-101 in the Nb5 CDR3 loop also forms an intricate hydrogen-bonding network with water molecules that extends throughout the Gβ₁-propeller cavity (Supplementary Figure 2b). Consistent with the differential HDX analyses, the crystal structure of the Gβ₁γ₁-Nb5 complex revealed a significant overlap between the binding interfaces formed by Nb5 and Gα$_t$ with Gβ₁γ₁ (Fig. 3b–e). Further, Sc analyses of the interface formed by Gα$_t$ with Gβ₁γ₁ displayed a Sc index of 0.76 and an interface area of 1080 Å² that are comparable to the Gβ₁γ₁-Nb5 complex. Overall, these results suggest a structural advantage of

Nb5 over most Gβγ regulatory proteins except Gα, for Gβγ binding. Furthermore, a higher affinity of Gα (apparent $K_d$ = 1–10 nM[30–32]) for the Gβγ dimer as compared to Nb5 ($K_d$ = 43 nM) supports the inability of Nb5 to affect Gα signaling.

**Nb5 binds to various Gβ subtypes.** Assessment of the binding interface between the Gβ₁γ₁ dimer and Nb5 revealed critical amino acid residues that are highly conserved among Gβ subtypes 1–4 (Fig. 4a). To further investigate the putative binding of Nb5 with different Gβ subtypes, Nb5-mediated Gβ purification was carried out with C57BL/6J mouse brain. As expected, Nb5 bound and pulled out the different Gβ subtypes from mouse brain whereas Nb17 had no binding partner (Fig. 4b). In-gel protein digestion followed by mass-spectrophotometry (MS) analyses of the Gβ gel band (Fig. 4b, asterisks) identified unique peptides from Gβ₁, Gβ₂, Gβ₃, and Gβ₄ subtypes (Fig. 4c, Supplementary Table 3). However, unique peptides from the Gβ₅ subtype were not observed due to its low sequence similarity with other Gβ subtypes. Notably, MS-identified peptides were searched against both the full mouse proteome and the primary sequences of Gβ subtypes to eliminate false-positives.

To further examine Gβ selectivity in a living cell, an optical assay was employed to monitor the interaction of Venus-tagged Gβγ and the nano Luciferase tagged C-terminal Gβ₁γ₂-interacting domain of G protein-coupled Receptor Kinase 3 (masGRK3ct-Nluc)[33]. Direct interaction of these sensor pairs increases the bioluminescence resonance energy transfer (BRET) ratio in transfected cells. As a result, co-transfection with a Gβγ-binding protein such as Gα, competes with the masGRK3ct-Nluc sensor and decreases the BRET ratio (Fig. 4d, pink). Indeed, exogenous Gα$_o$ markedly diminished the BRET ratio (Fig. 4e, pink). Similarly, but to a lesser extent, Nb5 suppressed the BRET ratio in cells transfected with all Venus-Gβγ complexes tested (Fig. 4e, greencyan) suggesting an interaction

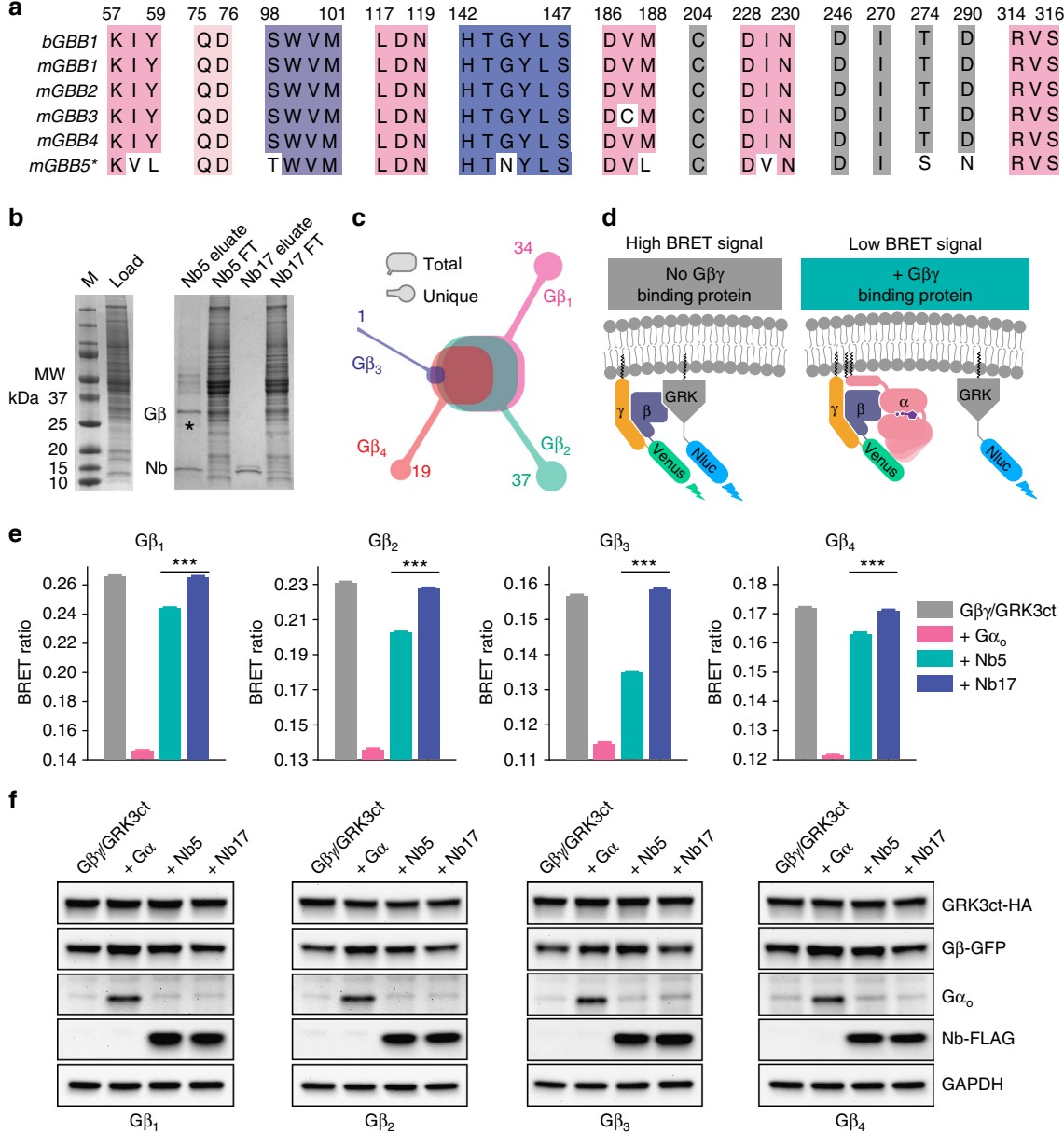

**Fig. 4** Gβ selectivity of Nb5. **a** Multiple sequence alignment of key amino acid residues of Gβ that interact with Nb5. *The amino acid numbering of Gβ5 has an off-set of −50. **b** Nb5-mediated Gβγ extraction from mouse brain. Solubilized and partially purified extracts from mouse brain (left) were subjected to either Nb5- (right, lanes 1 and 2) or Nb17- (right, lanes 3 and 4) mediated immobilized-metal affinity purification of Gβγ. **c** In-gel protein digestion of Gβ subtypes (band marked in **b** with asterisks) purified from mouse brain. Peptides were separated, analyzed and searched against a full mouse proteome to identify unique peptides from Gβ1, Gβ2, Gβ3, and Gβ4. **d** Schematic diagram of the effect of Gβγ-binding proteins on the BRET assay. Co-transfection of HEK293T/17 cells with Venus-Gβγ and masGRK3ct-Nluc-HA produced a high BRET signal through their direct interaction (left). Introduction of Gβγ-binding proteins (e.g., the Gα subunit) competed with masGRK3ct-Nluc-HA, lowering the BRET signal (right). **e** Effects of Nb5 on the interaction of Gβγ and the C-terminus of GRK3. The maximum BRET signal was determined by co-transfection of different Venus-Gβ subtypes + Gγ2 pairs and masGRK3ct-Nluc-HA (grey). A minimum BRET signal also was determined after co-transfection of Venus-Gβ1γ2 and masGRK3ct-Nluc-HA with an excess amount of GαoA (pink). Effects of Nb5 and Nb17 were examined by co-transfection of Venus-Gβγ and masGRK3ct-Nluc-HA with either Nb5 (greencyan) or Nb17 (purple). Experiments were performed with Gβ subtypes 1–4 + Gγ2 pairs. Each bar represents the mean of six replicates. Similar results were obtained in three independent experiments. Results are expressed as the mean ± SEM. One-way ANOVA with Tukey's post hoc multiple comparison test relative to the Gβ1γ2/GRK3ct control, ***$P \leq 0.001$, $n = 6$ replicates. **f** Western blot quantification of the expression levels of Gβ1–4, GRK3, Gα, Nb5, and Nb17 (Full blots are shown in Supplementary Figure 4a)

between Nb5 and Gβ subtypes 1–4. In contrast, Nb17 had no effect on the BRET ratio suggesting no interaction with Gβ subtypes (Fig. 4e, purple). Additionally, western blot analyses of the transfected components were performed to verify similar expression levels of the exogenous proteins (Fig. 4f). Interestingly, the transfection of either Gα or Nb5, but not Nb17 slightly increased the expression levels of Venus-Gβ1–4 subunits, suggesting the formation of a proteolytically stable complex of Gα and Nb5 with Venus-Gβγ dimer. Overall, these experiments suggest that Nb5 binds to all combinations of Gβ subtypes 1–4 and Gγ and suppresses Gβγ signaling mediated through protein–protein interactions near the Gβγ hotspot. These results

are suggestive of Nb5's potential broad utility to influence Gβγ signaling in various cell types.

**Nb5 inhibits Gβγ-mediated GIRK signaling in striatum neurons.** GIRK channels, found primarily in CNS neurons and atrial myocytes, respond to GPCR-mediated signaling through Gβγ binding events. Activation of GIRKs affects the flow of K⁺ ions across cell membranes that attenuate cellular electrical excitability. The capability of Nb5 to modulate Gβγ signaling was examined in medium spiny neurons (MSNs) from mouse striatum. GIRK channels (GIRK2; kir3.2) were virally over-expressed in the MSNs and the resulting outward currents were used as indicators of GPCR-mediated GIRK2 activation. Indirect pathway MSNs (iMSNs) expressed D2 dopamine receptors (D2Rs)[34] whereas direct pathway MSNs (dMSNs) expressed M4 muscarinic acetylcholine receptors (M4Rs)[35]. Both D2Rs and M4Rs are G$_{i/o}$-coupled receptors capable of gating GIRK channels through Gβγ signaling.

A single electrical stimulation in the striatum evoked the release of dopamine from neuronal dopaminergic terminals, resulting in a D2R-mediated inhibitory post-synaptic current

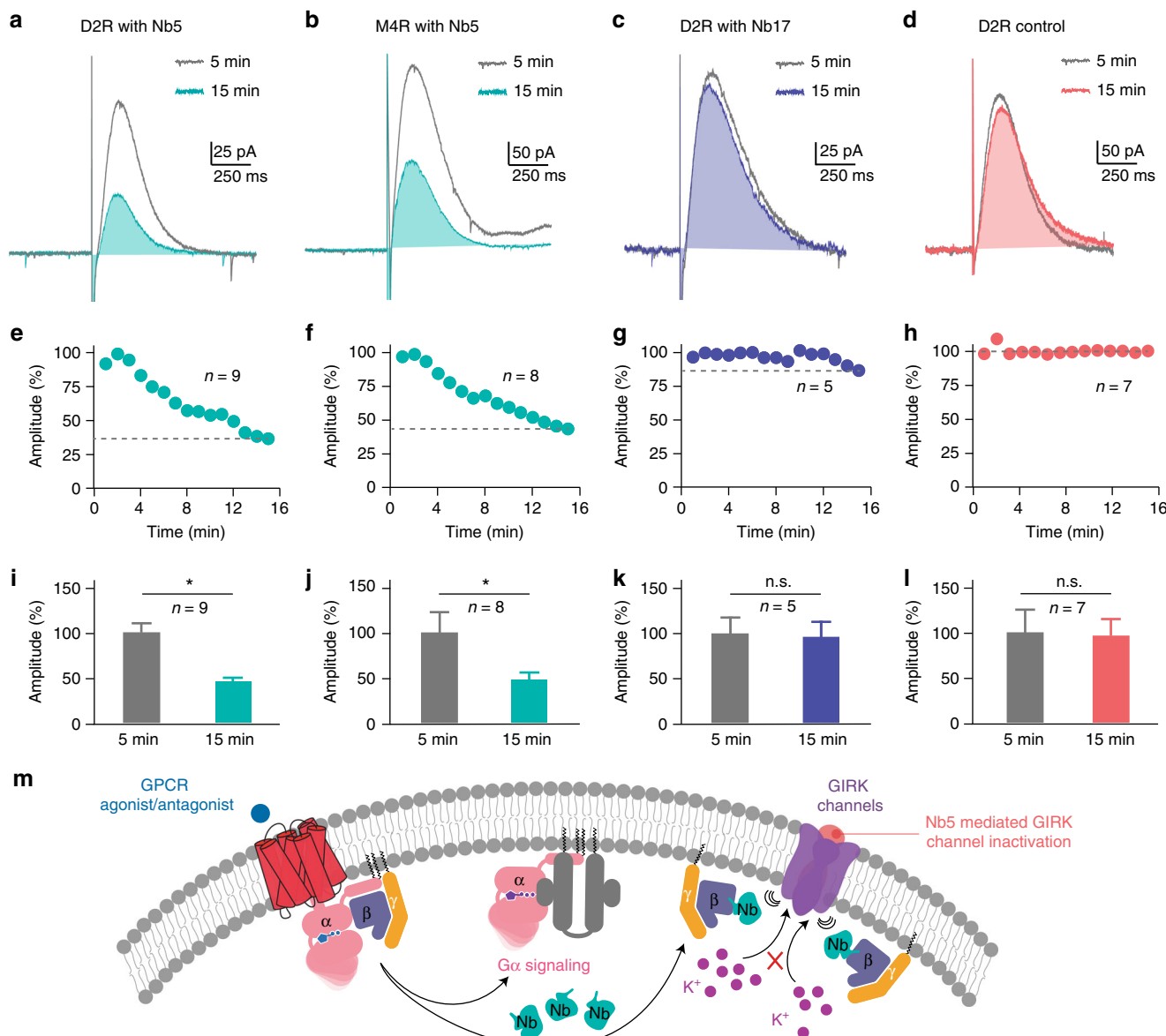

**Fig. 5** Inhibition of Gβγ-mediated GIRK signaling by Nb5. **a** Representative whole-cell recording of a GIRK2+ striatal medium spiny neuron (MSN) treated with 10 μM of intracellular Nb5. A single electrical stimulation evoked a D2R-IPSC that showed a significant amplitude reduction 15 min post-treatment with Nb5 (greencyan) as compared to the averaged trace observed at 5 min post-treatment with Nb5 (grey). **b** Representative trace of an M4-IPSC that displays an amplitude reduction 15 min post-treatment with Nb5 (greencyan) as compared to the averaged trace observed at 5 min post-treatment with Nb5 (grey). But no significant change was noted in D2R-IPSC either exposed to 10 μM intracellular Nb17 (**c**) or without nanobody treatment (**d**). Time course measurements showing a decrease in D2R-IPSC (**e**) and M4R-IPSC (**f**) amplitudes upon their exposure to 10 μM Nb5. Time course measurements of D2R-IPSC amplitudes recorded either with Nb17 (**g**) or without nanobody treatment (**h**). Bar graph quantification of D2R-IPSC (n = 9) (**i**) and M2R-IPSC (n = 8) (**j**) amplitudes recorded with Nb5. Bar graph quantification of D2R-IPSC amplitudes recorded (n = 5) either with 10 μM Nb17 (**k**) or without nanobody treatment (**l**), revealing no significant change in amplitudes over 15 min. Results are expressed as the mean ± SEM. *P < 0.05, Student's t-test **m** Proposed model of GIRK inhibition by Nb5

(D2R-IPSC) in postsynaptic GIRK2 positive iMSNs. Interestingly, inclusion of 10 μM Nb5 in the recording pipette attenuated D2R-IPSCs by about 60% ($208.12 \pm 47.08$ pA in 1 min; $75.18 \pm 9.40$ pA after 15 min, $n = 9$, $P < 0.05$, Student's $t$-test) (Fig. 5a, e, i). In contrast, including the irrelevant nanobody, Nb17 at the same concentration had no significant effect on the D2R-IPSCs amplitudes ($210.05 \pm 44.81$ pA in 1 min; $186.66 \pm 34.04$ pA after 15 min, $n = 5$, $P > 0.05$, Student's $t$-test) (Fig. 5c, g, k). Similarly, there was no effect on the D2R-IPSCs in postsynaptic GIRK2 positive iMSNs in the absence of nanobody treatment ($271.50 \pm 73.95$ pA in 1 min; $257.20 \pm 50.23$ pA after 15 min, $n = 7$, $P > 0.05$, Student's $t$-test) (Fig. 5d, h, l). To ensure that the effect of Nb5 was not D2R-specific, M4R-mediated inhibitory post-synaptic current (M4R-IPSC) measurements were also made on post-synaptic GIRK2 positive dMSNs. As expected, Nb5 attenuated M4-IPSCs evoked by electrical stimulation by about 60% ($430.84 \pm 81.09$ pA in 1 min; $179.68 \pm 22.70$ pA after 15 min, $n = 8$, $P < 0.05$, Student's $t$-test) (Fig. 5b, f, j). The effect of Nb5 in these GIRK inhibition assays was more demonstrable compared to its effects in the BRET assays. This likely is due to differences in the mode of delivery and thereby the effective concentration of Nb5 attained in cells in the two assay systems. While, Nb5 was co-transfected in human embryonic kidney cells 293 in BRET assays, GIRK recordings were performed by introducing an internal solution containing 10 μM of Nb5 directly to the neuronal exons. These results, together with the crystal structure of the $G\beta_1\gamma_1$-Nb5 complex demonstrate that Nb5 interferes with $G\beta\gamma$ dimer association to GIRK channels, thereby suppressing $G\beta\gamma$-regulated GIRK signaling (Fig. 5m).

**Nb5 acts as a control-switch for GPCR-mediated $G\beta\gamma$ signaling**. Next, the effect of Nb5 was investigated on cellular pathways that are often dysregulated such disease states as cancer. Here the PI3K-protein kinase B/AKT (PI3K-PKB/AKT)[17,18] and MAPK-extracellular signal-regulated kinase (ERK)[20] pathways were examined that are co-regulated by GPCR-mediated $G\beta\gamma$ signaling (Fig. 6a). The PI3K-PKB/AKT and MAPK/ERK pathways respond to a variety of cellular stimuli, and hence play a key role in diverse cellular processes, including cell survival, growth, proliferation, angiogenesis, metabolism, and migration. Because of the importance of these cellular pathways in cancer progression and metastasis, a Chinese hamster ovary cell line stably expressing human apelin receptor (APJ) was transfected with Nb5 cDNA and its effect upon the phosphorylation of AKT and ERK proteins downstream of APJ activation was assessed. Parental CHO-APJ cells were treated with 1 μM apelin over 0–30 min. Cell lysates then were collected and analyzed by immunoblotting to obtain optimum conditions to observe GPCR-mediated phosphorylation changes. Such changes were found most pronounced 5 min post-treatment with apelin (Supplementary Figure 3a). Interestingly, a significant decrease in the phosphorylation of both AKT and ERK1/2 proteins was observed in CHO-APJ-Nb5 cells 5 min post-treatment with apelin (Fig. 6b, c). In contrast, CHO-APJ cells, CHO-APJ-vector cells, and CHO-APJ-Nb17 cells showed no visible change in phosphorylation levels of either AKT or ERK proteins during the same time interval (Fig. 6b, compare lanes marked with asterisks). Additionally, no significant change was observed in the expression level of APJ with either Nb5 or Nb17 transfection (Supplementary Figure 3b). These results demonstrate a suppressive effect of Nb5 on APJ-mediated $G\beta\gamma$ signaling, further supporting its capability to modulate $G\beta\gamma$-mediated pathways governed by multiple GPCRs.

APJ signaling is regulated by the $G_{i/o}$ class of G proteins wherein both $G\alpha_i$ and $G\beta\gamma$ act either together or separately to affect AC-mediated intracellular cAMP production (Fig. 6d)[36,37].

To determine the effect of Nb5 on intracellular cAMP levels, the forskolin-stimulated accumulation of cAMP was measured in response to apelin treatment. Treatment of CHO-APJ cells with apelin inhibited the forskolin-stimulated accumulation of cAMP in a dose-dependent manner (Fig. 6e). CHO-APJ cells co-expressing Nb5 (greencyan) showed a reduced inhibition of cAMP accumulation at a higher dosage of apelin when compared to either parental CHO-APJ cells (grey) or CHO-APJ cells expressing the negative control Nb17 (purple). This result demonstrates that Nb5 modulates the apelin-induced inhibition of cAMP accumulation in CHO-APJ cells. In addition, Nb5 presumably had no effect on the $G\alpha_{i/o}$-GTP-mediated cAMP signaling after the loss of the $G\beta\gamma$-mediated cAMP component in cells transfected with Nb5 (Fig. 6e, greencyan, Supplementary Table 4). These results also are consistent with the higher affinity of $G\alpha$ compared to Nb5 for the $G\beta\gamma$ dimer. Overall, the binding of Nb5 to $G\beta\gamma$ negatively regulates $G_{i/o}$ signaling, at least partially by inhibiting $G\beta\gamma$-associated cAMP signaling.

To further confirm the specificity of Nb5 towards $G\beta\gamma$-induced signaling, we tested its effects on two well-established agonist-induced $G\alpha$ signaling pathways (Fig. 6f)[33]. Here, $G\alpha_q$ signaling was monitored in HEK293T/17 cells transfected with the M3 muscarinic acetylcholine receptor (M3R) and using a $Ca^{2+}$ sensor (CalFluxVTN). $G\alpha_s$ signaling was monitored in HEK293T/17 cells transfected with the dopamine D1 receptor (D1R) and using a cAMP sensor (Nluc-Epac-VV). The transfected cells were then stimulated with acetylcholine or dopamine to activate M3R or D1R, respectively. Real-time monitoring of the resultant responses clearly demonstrated that Nb5 had no effect on the elevation of intracellular $Ca^{2+}$ induced by the activation of M3R-$G\alpha_q$ (Fig. 6g, h) or on D1R-$G\alpha_s$-induced cAMP production (Fig. 6i, j) in living cells.

## Discussion
The significance of $G\beta\gamma$ signaling in various cellular functions has received appropriate recognition only recently in comparison to $G\alpha$ signaling. The $G\beta\gamma$ dimer functions as a negative regulator when bound to the $G\alpha$ subunit. Additionally, the $G\beta\gamma$ dimer regulates many downstream events depending on its interaction with different effector molecules. Many of these downstream events are dysregulated in neurological disorders and cancer, making the $G\beta\gamma$ dimer a critical drug target. However, the ability of $G\beta\gamma$ subunits to play essential roles in various cellular functions, including the formation of heterotrimeric G proteins, implies the potential for numerous side effects when $G\beta\gamma$ is the pharmacological target. In this study, it was found that Nb5, a nanobody against $G\beta_1$, can cause selective inhibition of $G\beta\gamma$-mediated signaling while leaving GTP-bound $G\alpha_q$ and $G\alpha_s$-mediated signaling intact. $G_t$ activation analyses showed that Nb5 causes a GPCR-independent shift in the dynamic equilibrium of heterotrimeric $G_t$ into its dissociated subunits in both membranes and in solution. Both in vitro SPR kinetics and $G_t$ activation assays suggest a tight interaction between Nb5 and the $G\beta\gamma$ dimer. $G\beta\gamma$ binding proteins, including $G\alpha$, PDC, GRKs, and PI3K are key regulators of GPCR signaling known to interact with the $G\beta\gamma$ dimer with nanomolar affinities. A similar binding affinity of Nb5 with $G\beta_1\gamma_1$ ensures competition between Nb5 and these $G\beta\gamma$ regulatory proteins in modulating $G\beta\gamma$-mediated signaling events. Indeed, combined differential HDX and x-ray crystallography studies demonstrate a possible competition between Nb5 and $G\beta\gamma$ regulatory proteins due to their overlapping binding interfaces. In fact, Sc index analysis that precisely measures correlations of directions suggests a structural advantage of Nb5 over most $G\beta\gamma$ regulatory proteins except $G\alpha$ for $G\beta\gamma$ binding. Furthermore, the binding interface between Nb5 and

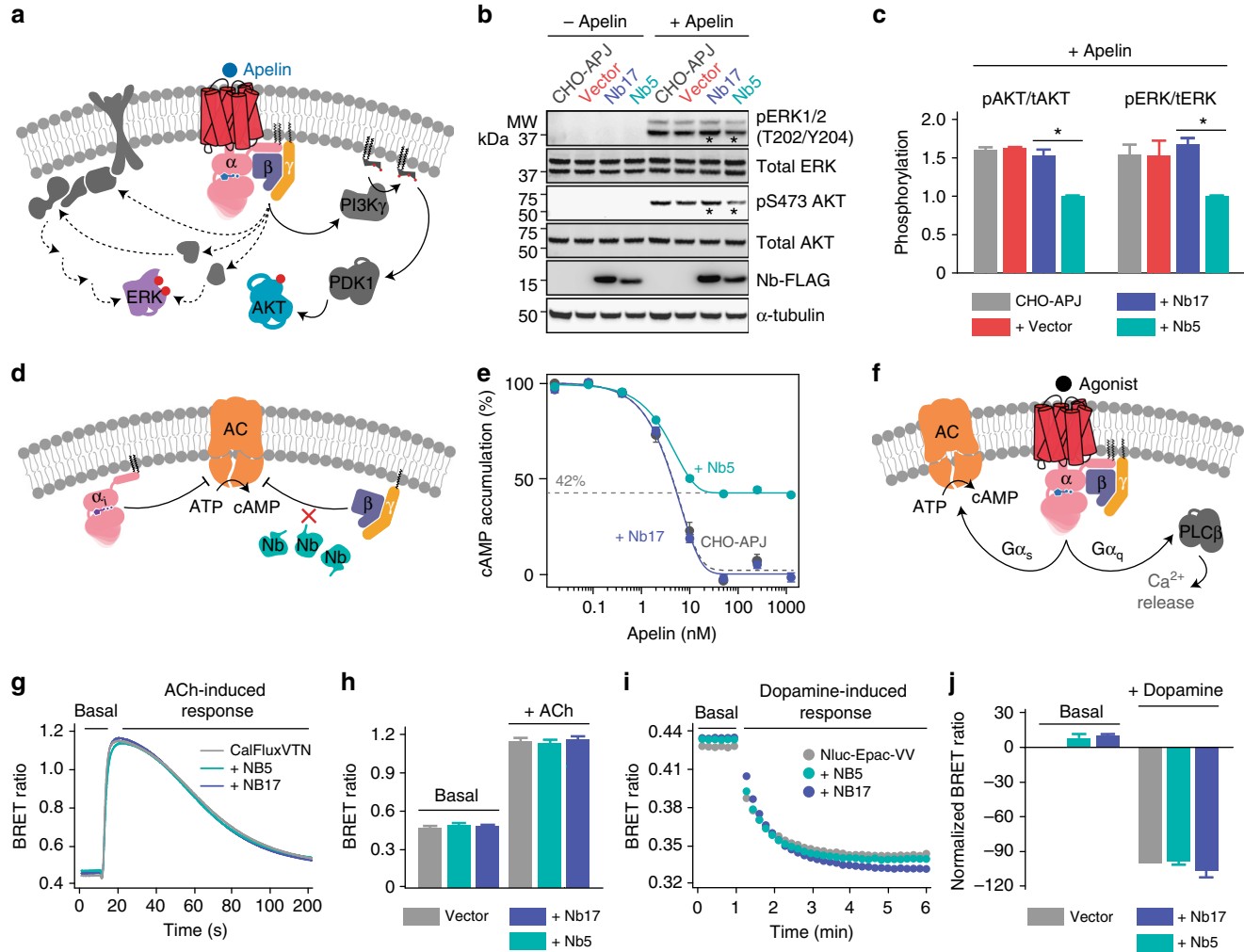

**Fig. 6** Nb5 as a control-switch for GPCR-mediated Gβγ signaling. **a** Schematic diagram of the GPCR-mediated Gβγ signaling network that controls various cellular functions. **b** Parental CHO-APJ cells, CHO-APJ cells transfected with pcDNA3.1 (+) empty vector, transfected CHO-APJ-Nb5 and CHO-APJ-Nb17 cells were treated with 1 μM apelin over 0–5 min to investigate the effect of Nb5 on the phosphorylation of ERK1/2 and AKT. Nb5 significantly decreased the phosphorylation of both ERK1/2 and AKT as compared to the Nb17 control (compare lanes marked with asterisks). Full blots are reported in Supplementary Figure 4b. **c** No significant effect of Nb17 transfection (purple) was observed on the phosphorylation of either ERK1/2 or AKT as compared to parental CHO-APJ cells (black) and CHO-APJ cells transfected with pcDNA3.1 (+) empty vector (red). Results are expressed as the mean ± SEM, $n = 3$ replicates, $*P < 0.05$, Student's $t$-test. **d** Schematic diagram showing the effect of Gαi and Gβγ signaling on cellular cAMP levels. **e** Dose-response curves of apelin-induced inhibition of cAMP accumulation in parental CHO-APJ cells (grey) and CHO-APJ cells expressing either Nb5 (greencyan) or Nb17 (purple). Apelin concentration is plotted on a semi-log scale. Results are expressed as the mean ± SEM, $n = 4$ replicates. **f** Schematic diagram of GPCR-mediated second-messenger regulation by Gαq and Gαs. **g** The effect of nanobodies on Gαq signaling events was determined in HEK293T/17 cells transfected with M3R and CalFluxVTN. Each trace represents the mean of the responses measured in six wells. **h** Basal BRET ratio and ACh-induced maximum amplitude shown as a bar graph. Similar results were obtained in three independent experiments. Results are expressed as the mean ± SEM, $n = 6$ replicates. Two-way ANOVA with Tukey's post hoc multiple comparison test. **i** Effect of nanobodies on Gαs signaling was evaluated in HEK293T/17 cells transfected with D1R and Nluc-Epac-VV. Each trace represents the mean of the responses measured in six wells. **j** Basal BRET ratio and dopamine-induced maximum amplitude shown as a bar graph. Each bar represents the mean of three independent experiments. Results are expressed as the mean ± SEM, $n = 6$ replicates. Two-way ANOVA with Tukey's post hoc multiple comparison test

Gβ1γ1 has the highest Sc index among several tested protein-Nb interactions[38,39]. Nb5-mediated Gβ purification from mouse brain and cell-based BRET assays demonstrate the specificity of Nb5 towards Gβ subtypes 1–4. No evidence for binding between Nb5 and Gβ5 was observed, likely because the sequence of Gβ5 differs from that of other Gβ subtypes with only 53% sequence identity to its closest Gβ subtype versus 80–90% sequence identity between other Gβ subtypes. Overall, these results identify Nb5 as a potential negative regulator of Gβγ signaling in various cell types.

The M119 class of small-molecule inhibitors, which includes M119, gallein and M119K, are potent Gβγ antagonists. They remain the most extensively validated inhibitors of Gβγ signaling. Studies to date indicate that these molecules bind to a Gβγ hot spot on the top surface of the β subunit with apparent affinities in the high nM to low μM range[40]. The M119 class of inhibitors displays a limited number of chemical moieties that interact with the Gβγ dimer. Therefore, structure activity relationship studies to achieve Gβ selectivity would have limited practicality. Nb5, on the other hand, binds Gβγ with a low nM affinity and has a larger area of interaction with the Gβγ dimer (>1030 Å²). These features indicate that Nb5 might serve as a better template for structure activity relationship studies to achieve Gβ selectivity. Unlike nanobodies, the M119 class of inhibitors can be delivered orally

and intraperitoneally to modulate Gβγ-dependent pathways, which admittedly takes them one step closer to clinical applications. In addition, the M119 class of inhibitors has the advantage that they specifically target the formation of Gβγ-GRK2 complexes[40]. Challenges remain for the development of nanobody-based clinical therapies due to the dependence of this approach on efficient gene transduction methods. Moreover, the ability of Nb5 to affect general Gβγ signaling raises concerns for possible off-target effects when pursued therapeutically[41]. Nevertheless, there may be applications for Nb5-based treatments where the beneficial effects outweigh the side effects. The C-terminal domain of GRK2 (βARK-ct), PDC and several affinity matured peptides, including the SIRK peptide family, comprise an alternative class of potent Gβγ inhibitors that are mechanistically similar to Nb5[42–44]. While βARK-ct and PDC have a large area of interaction with the Gβγ dimer (>1000 Å$^2$), no atomic level information is available for SIRK or other peptides except that they bind to the same Gβγ hot spot with high nM affinities[44,45] and selectively inhibit the binding of Gβγ effector molecules. Several cell-permeating versions of SIRK peptides have been shown to effectively modulate ERK1/2 and MAPK signaling in arterial smooth muscle cells[43]. Treatment strategies involving Nb5, βARK-ct, PDC, and members of the SIRK peptide family would all require gene transduction methods, however this may be worthy of significant effort given their potential to treat various Gβγ-related disorders.

Many neurological and neuromuscular disorders including Parkinson disease, Alzheimer disease, bipolar disorder, multiple sclerosis, and other age-related disorders are aggravated by dysregulated ion channel function. Ion channels like GIRK play a crucial role in maintaining ion homeostasis in neurons, determining their membrane potential and neurotransmitter secretion. The lack of endogenous GIRK channels in MSNs and their effective coupling to various GPCR signaling pathways makes them a useful tool for rapid sensing and monitoring of Gβγ-regulated GIRK signaling[35]. Whole-cell patch clamping of GIRK2 overexpressed MSNs was performed to investigate the role of Nb5 as a negative regulator of Gβγ-mediated signaling. Treatment with Nb5 decreased both D2R- and M4R-IPSCs amplitudes, and thus inhibited Gβγ-regulated GIRK channel opening. This serves as one of the first examples wherein a nanobody modulates a GPCR-mediated Gβγ-signaling event in any cell type. A body of evidence demonstrates that Gβγ subunits play significant roles in regulating downstream G protein signaling including the MAP kinase pathway. Previous studies have shown that Gβγ antagonist βARK1-(495–689) inhibits Gα$_i$-mediated ERK activation in a Gβγ dependent manner[46,47]. Extensive studies on small molecule-based and peptide-based Gβγ inhibitors show that Gi-coupled GPCR-mediated ERK signaling is altered by M119 class and SIRK Gβγ inhibitors[40,43]. Our data also suggest that Nb5 reduces downstream phosphorylation events in Gα$_i$-linked AKT and ERK kinase pathways. Notably, Nb5 does not completely ablate either GIRK activation or downstream AKT and ERK phosphorylation, suggesting its capability to suppress dysregulated pathways with critical roles in cancer progression and metastasis.

The Gβγ dimer classically was thought to inhibit AC activity by binding and inhibiting stimulatory Gα subunits. However, recent studies have shown that Gβγ can stimulate several AC isoforms through cross-talk between Gα-mediated and Gβγ-mediated cAMP signaling[19,48,49]. Co-expression of Nb5 in CHO-APJ cells showed a 58% reduction in apelin-mediated inhibition of cAMP accumulation when compared to either parental cells or CHO-APJ co-expressing Nb17. This result demonstrates that Nb5 modulates the forskolin-stimulated accumulation of cAMP in CHO-APJ cells by inhibiting the Gβγ-mediated cAMP signaling component. Additionally, Nb5 showed no significant effect on either the M3R-Gα$_q$-induced intracellular Ca$^{2+}$ elevation or D1R-Gα$_s$-induced cAMP production. These results show that Nb5 does not alter the GTP-bound Gα$_q$ and Gα$_s$-mediated signaling component in living cells. Overall, Nb5 serves as a dynamic scavenger of the Gβγ dimer and thereby causes partial inhibition of Gβγ-mediated signaling. The remaining Nb5-free Gβγ supports the canonical Gα-GTP-mediated signaling that includes the GPCR-mediated GDP-GTP exchange from Gα, and signaling termination by GTP hydrolysis and re-association of Gα-GDP with Gβγ. However at higher concentrations, Nb5 might affect the Gα-GTP signaling component due to increased sequestration of Gβγ dimers (Fig. 1d,e).

In summary, the ability of nanobodies to act as crystallization chaperones by either stabilizing intrinsically flexible regions or shielding aggregating surfaces in a protein is well established[50], but their ability to serve functional roles in cellular signaling has not been well studied. This work highlights the functional importance of these genetically-encodable antibody fragments in controlling various aspects of GPCR-signaling pathways. Interestingly, Nb5 acts as a specific inhibitor of various Gβγ-mediated signaling events that regulate several 'undruggable' targets such as GIRK channels, ERK, and AKT kinases. The ability of Nb5 to suppress but not completely ablate Gβγ signaling makes it a beneficial tool for future nanobody-based therapeutics to modulate cellular signaling. This work on Nb5 provides a proof of principle and serves as one of the first examples wherein a nanobody modulates a GPCR-mediated Gβγ-signaling event in any cell type. Future experiments will focus on increasing the Gβ selectivity of Nb5 by using both structure guided mutational studies and development of Gβ selective phage display libraries. The ease of production and genetic manipulation of nanobodies are advantageous characteristics for achieving the goal of engineering variants with specificity toward different Gβ subtypes and even Gβγ combinations. This opens up the possibility of designing targeted treatments for various excitatory neurological conditions and inhibitors for blocking cancer progression. However, the development of nanobody-based clinical therapies relies heavily on the advancement of specific and efficient gene transduction methods such as CRISPR/Cas9 and viral-gene transduction. Additional strategies for intracellular nanobody delivery include coupling the Nbs to cell-penetrating peptides such as penetratin[51]. Elucidating nanobody translocation across the CNS could also provide a tremendous advantage in treating neurological disorders. Such strategies might include internalization via either clathrin-coated vesicles[52] or Trojan horse technology using transferrin and insulin receptors[53]. The comprehensive biochemical and structural analyses of Nb5 reported in this study provide a basis for further research studies to turn nanobodies into a viable class of therapeutics. We believe that nanobody-based therapeutics will find a niche in cases where 'undruggable' protein targets are involved in multiple signaling pathways and where conventional therapeutic approaches induce unacceptable side effects.

## Methods

**Reagents**. Antibodies recognizing phosphorylated-ERKs 1/2 (αpT202/pY204-ERK1/2, catalog no. 9106), total ERKs 1/2 (αERK1/2, catalog no. 4695), phosphorylated-Akt (αpS473-Akt, catalog no. 4058), total Akt (αAkt, catalog no. 9272), and anti-α-tubulin (catalog no. 2144) were from Cell Signaling Technology (Danvers, MA). Anti-APJ antibody (3C3–7, catalog no. MABN1846) was from Millipore Sigma (St. Louis, MO). Polyclonal αFlag antibody (catalog no. A190-101A) was from Bethyl Laboratories, Inc. (Montgomery, TX). Anti-Flag antibody (catalog no. F7425), αHA antibody (catalog no. 11867423001), αGAPDH antibody (catalog no. MAB374) and αGFP antibody (catalog no. 11814460001) were from Millipore Sigma (St. Louis, MO) and used for Western blotting analysis in Fig. 4f. Anti-Gαo antibody (catalog no. sc-387) was from Santa Cruz (Dallas, TX). IRDye-conjugated anti-mouse IgGs (catalogs P/N 925-32210 and P/N 925-68070), and anti-rabbit IgGs (P/N 925-32211 and P/N 925-68071) were from LI-COR (Lincoln,

NE). HRP-anti-goat IgG was from ThermoFisher Scientific, Inc (Waltham, MA). Apelin-13 (catalog no. 057-18) was from Phoenix Pharmaceuticals, Inc (Burlingame, CA). IBMX (catalog no. 2845) and Forskolin (catalog no. 1099) were from R&D systems, Inc (Minneapolis, MN). CHO-K and HEK293T/17 cells (catalog nos. CRL-9618 and CRL11268, respectively) were from the American Type Culture Collection (Manassas, VA) and have no mycoplasma contamination. All antibodies were used at 1:1000 dilution.

A pCMV5 plasmid encoding $G\alpha_{oA}$ was a gift from Dr. Hiroshi Itoh (Nara Institute of Science and Technology, Japan). Plasmids encoding Venus 156-239-$G\beta_1$, and Venus 1-155-$G\gamma2$ were gifts from Dr. Nevin A. Lambert (Augusta University)[54]. The masGRK3ct-Nluc-HA constructs were constructed by introducing HA tag at the C-terminus of masGRK3ct-Nluc reported previously[33]. Amino acids 156–239 of Venus were fused to a GGSGGG linker at the N-terminus of $G\beta_2$ (GenBank: AF501883), $G\beta_3$ (GenBank: M31328), or $G\beta_4$ (GenBank: AF300648) to construct Venus 156–239-$G\beta$ subunits. Nb5-Flag and Nb17-Flag constructs were based on the primary sequences of Nb5 and Nb17 wherein the 6 × His-tag was replaced by the FLAG-tag (GATTACAAGGATGACGACGATAA GTAG). Human M3 muscarinic acetylcholine receptor and human dopamine D1 receptor in pcDNA3.1(+) were purchased from cDNA Resource Center (Bloomsberg, PA). Nluc-Epac-VV and CalFluxVTN in pcDNA3.1(+) were reported previously[33]. Adeno-associated virus (AAV) AAV9.hSyn.tdTomato.T2A. mGIRK2-1-A22A.WPRE.bGH was obtained from the University of Pennsylvania Viral Core[35].

**Generation and phage selection of nanobodies**. Nanobodies against the $G\beta_1\gamma_1$ dimer were prepared by a previously published protocol[10]. Briefly, a llama (Lama glama) was immunized weekly for six weeks with 1 mg of purified bovine $G\beta_1\gamma_1$ dimer. Peripheral blood lymphocytes from anti-coagulated blood were then used to prepare cDNA clones that served as templates for amplification of open reading frames encoding the variable domains of the heavy-chain only antibodies. PCR fragments were then ligated into the pMESy4 phage display vector and transformed in *E. coli* TG1 cells (Lucigen, Middleton, WI) to create a nanobody library of 4 × $10^9$ transformants. After super infection with M13 helper phage[10], the display library was added to antigen-coated wells of MaxiSorp ELISA plates and selections were performed in buffer containing 10 mM HEPES, pH 7.5, 100 mM NaCl, 2 mM MgCl₂, 1 mM EDTA, and 1 mM DTT. After washing, phage were eluted by incubating the antigen-coated wells with 100 μl of trypsin (250 μg ml⁻¹) for 30 min. Freshly grown TG1 cells were then infected with the eluted phage and grown overnight at 37 °C. A total of 184 colonies were randomly picked and analyzed with ELISA. Testing for specific binding to the $G\beta_1\gamma_1$ dimer resulted in 14 families wherein all nanobodies were produced as soluble His-tagged proteins in the *E. coli* periplasmic region[10].

**Nanobody expression and purification**. Nanobodies (Nb5 and Nb17) were expressed and purified as described before[10]. Briefly, nanobodies bearing a C-terminal His-tag were transformed in *E. coli* WK6 (Su⁻) cells[55]. Small scale cultures of 5 ml were grown in LB media containing 2% glucose, 1 mM MgCl₂ and 50 μg ml⁻¹ ampicillin. Large scale cultures of 2 L were grown to $OD_{600}$ = 0.9–1.0 at 37 °C in TB media containing 0.1% glucose, 1 mM MgCl₂, and 50 μg ml⁻¹ ampicillin. Cultures were induced with 1 mM IPTG and grown overnight at 28 °C. Cells were harvested by centrifugation at 11,000 *g* for 20 min at 4 °C. The periplasmic fraction was extracted by re-suspending cells in ice-cold TES buffer containing 0.2 M Tris, pH 8.0, 0.5 mM EDTA, and 0.5 M sucrose. Following a 1 h incubation at 4 °C, cells were supplemented with 4 × diluted TES buffer to achieve a final buffer composition of 0.1 M Tris, pH 8.0, 0.25 mM EDTA, and 0.25 M sucrose. After 1 h incubation at 4 °C, cells were removed by centrifugation at 11,000 *g* for 30 min at 4 °C. The periplasmic extract was filtered through a 0.22 μm vacuum driven filter and subjected to immobilized-Ni²⁺ affinity chromatography followed by size exclusion chromatography (SEC) on a Superdex 200 10/300 GL column equilibrated with 20 mM bis-Tris propane, pH 7.0 and 100 mM NaCl. Peak protein fractions were pooled and concentrated to 10 mg ml⁻¹.

**Rh purification**. All experimental procedures were carried out in a darkroom under dim red light (>670 nm). Bovine rod outer segments (ROS) were prepared as described elsewhere[34]. ROS were washed with isotonic and hypotonic buffer to remove both soluble and membrane-associated ROS proteins[56,57]. Rh was purified by a protocol described previously[58]. Briefly, native Rh membranes were solubilized by a zinc/alkyl-glucoside extraction method and centrifuged at 100,000 *g* for 40 min to extract Rh. Clear supernatants were loaded on a 1D4-coupled CNBr-activated Sepharose 4B column and washed with buffer containing 10 mM MES, pH 6.4, 100 mM NaCl, and 0.02% *n*-dodecyl β-D-maltoside (DDM) to either dispose of excess retinal or achieve further purification. Finally, purified Rh was eluted with 0.5 mg ml⁻¹ of TETSQVAPA nanopeptide (from the Rh C-terminal sequence).

**Purification of heterotrimeric Gₜ**. Bovine rod outer segments (ROS) were prepared as described elsewhere[34] in a darkroom under dim red light (>670 nm). ROS were resuspended in isotonic buffer containing 20 mM HEPES, pH 7.5, 100 mM NaCl, 1 mM DTT and 5 mM MgCl₂. Resuspended ROS were then centrifuged at 31,000 *g* at 4 °C for 25 min to remove soluble and some membrane-associated

proteins. The pellet was then gently homogenized twice in hypotonic buffer containing 5 mM HEPES, pH 7.5, 1 mM EDTA, and 1 mM DTT by manually passing the solution through a glass-to-glass homogenizer. The homogenized suspension was centrifuged at 40,000 *g* for 30 min at 4 °C. Supernatants from the two hypotonic washes were pooled and centrifuged multiple times at 40,000 *g* for 30 min at 4 °C to completely remove any residual ROS pellet. The clear supernatant was dialyzed against the equilibrating buffer containing 10 mM HEPES, pH 7.5, 2 mM MgCl₂, and 1 mM DTT for 3 h at 4 °C. For purification of heterotrimeric Gₜ, the hypotonic solution was loaded onto a C10/10 column (GE Healthcare) with 6 mL of pre-equilibrated propyl-agarose resin. Next, the column was washed with 30 resin volumes of the equilibration buffer followed by 2 resin volumes of the same buffer containing 50 mM NaCl. Bound proteins were eluted with 50 mL of equilibration buffer containing 0.5 M NaCl. The eluate containing heterotrimeric Gₜ was concentrated and loaded onto a Superdex 200 10/300 GL column equilibrated with equilibration buffer containing 0.1 M NaCl. Fractions containing heterotrimeric Gₜ were combined, concentrated to about 2 mg ml⁻¹ and used for further analyses.

**Nanobody-mediated shift in heterotrimeric Gₜ equilibrium**. ROS were resuspended in isotonic buffer containing 20 mM HEPES, pH 7.5, 100 mM NaCl, 1 mM DTT, and 5 mM MgCl₂. For Gₜ dissociation experiments in ROS membranes, the resuspended ROS (2 mg ml⁻¹ Rh) were either illuminated with a 150-W fiber light in the presence of 250 μM GTP, co-treated with 250 μM GTP and 3 μM Nb5, or treated with either 3 μM Nb5 or 3 μM Nb17 for 30 min at 4 °C. The resuspensions were then centrifuged at 100,000 *g* at 4 °C for 20 min and the supernatants were analyzed by SDS-PAGE. In solution Gₜ dissociation experiments were conducted on supernatants obtained after ROS hypotonic washes. Hypotonic extracts were combined and centrifuged multiple times at 40,000 *g* for 30 min at 4 °C to ensure complete removal of the residual ROS pellet. Next, the hypotonic extracts (1 μM Gₜ) were either supplemented with 1 μM Rh and illuminated with a 150-W fiber light in the presence of 250 μM GTP and 2 μM Nb5, co-treated with 250 μM GTP and 2 μM Nb5, or treated with either 2 μM Nb5 or 2 μM Nb17 for 30 min at 4 °C. Then these treated hypotonic extracts were subjected to immobilized-Ni²⁺ affinity chromatography. A small volume of Ni²⁺-NTA resin (250 μl) pre-equilibrated with 10 mM HEPES, pH 7.5, 0.1 M NaCl, 2 mM MgCl₂ and 1 mM DTT was added to the treated hypotonic extracts. After 1 h of incubation, the resin was washed with 50 resin volumes of equilibration buffer and bound proteins were eluted in equilibration buffer containing 300 mM imidazole, pH 7.5. Eluted fractions were then analyzed by SDS-PAGE.

**Gₜ activation assay**. The intrinsic fluorescence increase from $G\alpha_t$ was measured with a L55 luminescence spectrophotometer (PerkinElmer Life Sciences) operating at excitation and emission wavelengths of 300 and 335 nm, respectively[23,59]. The ratio of Gₜ to Rh was 20:1, with Gₜ at a concentration of 1000 nM and Rh at 50 nM. Gₜ was pre-incubated with a two molar excess of either Nb5 or Nb17 to determine their effects on Rh*-mediated Gₜ activation rates. This was followed by the addition of 300 μM GTPγS (Sigma-Aldrich) to determine the GTPγS-induced complex dissociation and fluorescence changes[58]. Samples were bleached for 1 min with a Fiber-Light delivered through a 480–520 nm long pass wavelength filter prior to the fluorescence measurements. Gₜ activation rates were determined for the first 100 s in a Gₜ activation assay[58].

**$G\beta_1\gamma_1$ alone and $G\beta_1\gamma_1$-Nb5 complex purification**. Bovine ROS were washed with isotonic buffer containing 20 mM HEPES, pH 7.5, 100 mM NaCl, 1 mM DTT, and 5 mM MgCl₂. For $G\beta_1\gamma_1$ purification, ROS washed with isotonic buffer were illuminated with a 150-W fiber light (NCL-150, Volpi, USA) for 15 min and then washed twice with hypotonic buffer containing 5 mM HEPES, pH 7.5, 1 mM EDTA, and 1 mM DTT supplemented with 250 μM GTP. The hypotonic wash extracts were pooled and applied to a Blue Sepharose CL-GB column pre-equilibrated with 10 mM HEPES, pH 7.7, 6 mM MgCl₂, 1 mM EDTA, and 1 mM DTT. $G\beta_1\gamma_1$ obtained from the unbound flow-through was further purified by SEC on a Superdex 200 10/300 GL column (GE Life sciences) equilibrated with 50 mM Tris-HCl, pH 7.5, 100 mM NaCl, 2 mM MgCl₂ and 3 mM DTT. For purification of the $G\beta_1\gamma_1$-Nb5 complex, ROS washed with isotonic buffer were washed twice with hypotonic buffer and the washes were combined for further purification. Hypotonic wash extracts were then incubated with a 1:2 molar ratio of Nb5 for 1 h at 4 °C. This was followed by the addition of Ni²⁺-NTA resin pre-equilibrated with 50 mM Tris-HCl, pH 7.5, 100 mM NaCl, 5 mM imidazole, 2 mM MgCl₂, and 1 mM DTT. Following 1 h of incubation, the resin was washed with 50 resin volumes of equilibration buffer and the $G\beta_1\gamma_1$-Nb5 complex was eluted with equilibration buffer containing 300 mM imidazole. The Ni²⁺-NTA purified $G\beta_1\gamma_1$-Nb5 complex was then loaded on a Superdex 200 10/300 GL column equilibrated with 50 mM Tris-HCl, pH 7.5, 100 mM NaCl, 2 mM MgCl₂, and 3 mM DTT.

**$G\beta_1\gamma_1$-Nb5 complex crystallization**. SEC fractions were concentrated to 4.9 mg ml⁻¹ and then used for crystallization. The concentrated protein was supplemented with 2 mM DTT. Crystallization screens by the sparse matrix crystallization method were carried out by both the hanging-drop and sitting drop vapor diffusion methods. Each hanging drop was prepared on a siliconized coverslip by mixing equal volumes of

Gβ$_1$γ$_1$-Nb5 complex and reservoir solution. The reservoir solution contained 25% (w/v) PEG 3350 in 0.1 M Bis-Tris-HCl, pH 5.5–5.7, and 0.2–0.3 M MgCl$_2$. Crystals appeared in 2 days at 4 °C and reached 50–80 μm in their longest dimension within 5 days. Crystals were harvested directly from the mother liquor into dual-thickness microloops (MiTeGen, LLC) and plunge-frozen in liquid nitrogen.

**Diffraction data collection and structural refinement.** X-ray data collection of the Gβ$_1$γ$_1$-Nb5 complex was performed at −173 °C. Diffraction data were collected at the NE-CAT-24-ID-E beamline. Data were integrated with XDS and scaled using XSCALE[60]. Initial phases for the Gβ$_1$γ$_1$-Nb5 complex were obtained by molecular replacement using the Gβ$_1$γ$_1$-phosducin complex and TssK nanobody nb18 structures as search models (PDB accession: 1A0R[21], 5M2W[61]) with the CCP4 program PHASER[62,63]. Initial models were improved by multiple rounds of REFMAC ver. 5.8[62] refinement against the Gβ$_1$γ$_1$-Nb5 complex dataset and manual model adjustments with Coot 0.8.8[64]. The final models had agreement factors R$_{free}$ and R$_{cryst}$ of 24.7% and 20%, respectively. The stereochemical quality of the Gβ$_1$γ$_1$-Nb5 complex model was assessed with the Molprobity[65] and wwPDB validation servers[66]. Details of the diffraction data collection and structural refinement statistics are provided in Supplementary Table 2. Coordinates and structure factor amplitudes were deposited in the PDB (PDB accession: 6B20). The dimer interface shape complementarity was calculated by Sc[29] and PISA[28].

**Differential HDX.** Amide H/D exchange was performed as described previously[67]. Briefly, 10 μg of purified Gβ$_1$γ$_1$ or Gβ$_1$γ$_1$-Nb5 complex were diluted in 70 μL of ice cold D$_2$O to obtain a final concentration of 80% D$_2$O. The solution was incubated on ice for 10 min to achieve steady state deuterium exchange conditions. This was followed by quenching the reaction with ice cold buffer containing D$_2$O in formic acid (Sigma-Aldrich) to obtain a final pH of 2.5. Non-deuterated samples were quenched with buffer containing H$_2$O in formic acid, pH 2.5. Samples were digested with freshly prepared 5–20 μg pepsin (Worthington, Lakewood, NJ) solution prepared in H$_2$O. All samples were digested over a time range of 1–5 min on ice. Next, samples (100 μL) were loaded onto a C8 trap (2.1 mm, Thermo Scientific) and a C18 (20 mm × 2.0 mm, Phenomenex) column using a temperature-controlled Accela 600 autosampler and pump (Thermo Scientific) with a temperature set to 4 °C. Separation was achieved with an Agilent 1100 HPLC system (Agilent Technologies, Santa Clara, CA) at a flow rate of 0.1 ml min$^{-1}$. Peptides were eluted with a gradient from 98% of buffer A containing 0.1% (v/v) formic acid in H$_2$O and 2% of buffer B containing 0.1% (v/v) formic acid in acetonitrile to 2% of buffer A and 98% of buffer B. The eluent was injected into a Thermo Finnigan LXQ (Thermo Scientific, Waltham, MA) MS equipped with an electrospray ionization source operated in the positive ion mode with other parameters adjusted as follows: activation type was set to collision induced dissociation, normalized collision energy to 35 kV, capillary temperature to 370 °C, source voltage to 5 kV, capillary voltage to 43 V, tube lens to 105 V, and then MS spectra were collected over a 200–2000 mass range. To avoid contamination from previous runs, each production run was followed by a 10 μl mock injection of buffer A, followed by residual peptide elution with the gradient profile described above. Each run was also followed by a 20 min equilibration run with 98% buffer A and 2% buffer B.

**Differential HDX data analysis.** mzXML files were generated from raw data with the MassMatrix file conversion tool. Peptides were identified by searching against the primary sequence of bovine G protein beta 1 (GBB1; Uniprot ID: P62871) and G protein gamma 1 (GNGT1; Uniprot ID: P02698) using MassMatrix v3.10. Search parameters for peptide identification were as follows: precursor ion tolerance, 3.0 Da; variable modifications, farnesylation and methylation of the N-terminus; minimal peptide length, 6 amino acids; minimal pp score, 5; pptag score, 1.3; maximal number of combinations of different modification sites for a peptide match with modifications, 1; and maximal number of candidate peptide matches for each spectrum output in the result, 1. Raw data in the form of the relative signal intensity (percent) as a function of m/z were extracted with Xcalibur version 2.1.0. Qual Browser was used for recently described semiautomated peak detection, and a deconvolution procedure was performed with HXExpress[68].

**SPR analyses.** SPR data acquisition was carried out using the Biacore T100 SPR instrument (Biacore, GE Healthcare) at 25 °C. Purified Nb5 (50 ng ml$^{-1}$) was injected at a flow rate of 10 μl min$^{-1}$ in 10 mM sodium acetate, pH 5.5 to achieve capture level between 400–500 resonance units (RU) on a CM5 sensor chip according to directions in the manufacturer's amine coupling kit. After Nb5 immobilization, the surface was blocked with 1 M ethanolamine at pH 8.5, followed by regeneration using 50 mM NaOH. The interaction experiments were performed using running buffer containing 10 mM HEPES pH 7.5, 150 mM NaCl, and 1 mM Tris(2-carboxyethyl)phosphine and 0.1% Tween-20. Binding experiments were carried out using a Gβ$_1$γ$_1$ concentration range of 0.6–312 nM in running buffer at a flow rate of 30 μl min$^{-1}$. The association and dissociation kinetics for Gβ$_1$γ$_1$ were monitored for 140 and 300 s, respectively. SPR data processing and analysis were performed using Biacore T100 Evaluation Software (GE, version 2.0.3). For kinetic analyses, data were locally fit to a 1:1 Langmuir model to obtain on-rate and off-rate constants.

**G protein extraction from mouse brain.** G proteins were extracted from mouse brain using a protocol described elsewhere with a few modifications[69]. Brains from 32 C57BL/6 J mice (Jackson laboratory) were thawed in 50 ml buffer containing 20 mM Tris-HCl, pH 8.0, 1 mM EDTA, 1 mM DTT, 3 mM MgCl$_2$ and halt protease inhibitor cocktail (ThermoFisher). Thawed brains were homogenized and centrifuged at 39,000 g for 20 min at 4 °C. Membrane pellets were homogenized in 50 ml buffer A containing 20 mM Tris-HCl, pH 8.0, 1 mM EDTA, 1 mM DTT and halt protease inhibitor cocktail and washed twice after adding buffer A by centrifugation at 39,000 g for 20 min. Pellets then were resuspended and washed by centrifugation in buffer A supplemented with 0.1 M NaCl and 0.1% (w/v) sodium cholate. Next, membranes were homogenized and solubilized in buffer A containing 2% (w/v) sodium cholate for 2 h at 4 °C. The clear supernatant obtained after centrifugation at 186,000 g was loaded onto a DEAE Sepharose column equilibrated with buffer A containing 20 mM NaCl and 0.2% sodium cholate. Bound proteins were eluted with a linear gradient of NaCl (0–500 mM) in buffer A with 0.2% sodium cholate. The eluate was desalted, supplemented and incubated with a two molar excess of either Nb5 or Nb17 for 1 h and then subjected to immobilized-Ni$^{2+}$ affinity chromatography. The Ni$^{2+}$-NTA resin was equilibrated with buffer A containing 0.1 M NaCl, 0.2% sodium cholate and 5 mM imidazole and this was added to the eluate obtained during DEAE Sepharose purification. Following 1 h of incubation, the resin was washed with 50 resin volumes of buffer A containing 0.1 M NaCl, 0.2% sodium cholate and 5 mM imidazole. The Gβγ-Nb5 complex then was eluted with buffer A containing 0.1 M NaCl, 0.1% sodium cholate and 300 mM imidazole. The Ni$^{2+}$-NTA purified Gβγ-Nb5 complex was desalted and loaded onto 500 μl of Talon resin packed into a Pierce™ disposable column (ThermoFisher) equilibrated with buffer A containing 0.1 M NaCl, 0.1% sodium cholate, and 5 mM imidazole. The Talon resin was washed with 50 resin volumes of equilibration buffer and the purified Gβγ-Nb5 complex was eluted with buffer A containing 0.1 M NaCl, 0.1% sodium cholate and 300 mM imidazole.

**In-gel protein digestion of Gβ subtypes.** The excised SDS-PAGE band was destained in a solution containing 50% (v/v) ethanol with 5% (v/v) acetic acid in water. Next, the gel band was treated with acetonitrile and 5 μM DTT followed by its alkylation with iodoacetamide. Complete in-gel protein digestion was achieved by treatment with 50 ng of trypsin and chymotrypsin proteases prepared in 50 mM ammonium bicarbonate for 16 h at room temperature. Peptides were extracted by washing the gel twice with 30 μL of 50% (v/v) acetonitrile with 5% (v/v) formic acid in water. Extracts were combined and evaporated to <10 μL in a Speedvac and then resuspended in 1% (v/v) acetic acid to achieve a final volume of 30 μL. Next, the sample was loaded on to a Dionex 15 cm × 75 μm id Acclaim Pepmap C18, 2 μm, 100 Å reversed phase capillary chromatography column attached to a Finnigan LTQ-Obitrap Elite hybrid mass spectrometer system. Peptides were eluted with a gradient of 0.1% (v/v) formic acid in acetonitrile at a flow rate of 0.3 μL min$^{-1}$ and these were injected into the electrospray ionization source of the mass spectrometer on-line. The nano-electrospray ion source was operated at 1.9 kV. The digest was analyzed using the data-dependent multitask capability of the instrument acquiring full scan mass spectra to determine peptide molecular weights and product ion spectra to obtain the amino acid sequence in successive instrument scans. Data were analyzed by using all CID spectra collected in the experiment to search whole mouse UniProtKB databases with the search program Sequest. Search parameters for peptide identification were as follows: minimum precursor mass, 350 Da; maximum precursor mass, 5000 Da; maximum missed cleavage sites, 2 for trypsin and 6 for chymotrypsin digestion; minimum peptide length, 6; maximum peptide length, 144; precursor mass tolerance, 10 ppm; fragment mass tolerance, 0.8 Da; static modification, carbamidomethylation; dynamic modification, oxidation for trypsin and oxidation and phosphorylation for chymotrypsin digestion; maximum dynamic modifications per peptide, 4; and maximum Delta Cn (degree of match between the scores of the possible peptide spectral matches), 0.05. Sequest searches were also performed against five Gβ subtypes using the same parameters to confirm the identities of the peptides and the sequence coverage. Data are available via ProteomeXchange with identifier PXD009503.

**Gβ subunit selectivity of Nb5.** HEK293T/17 cells (American Type Culture Collection, Manassas, VA) were chosen for this analysis because of their high transfectability[70]. Cells were grown in culture medium (Dulbecco's modified Eagle's medium) supplemented with 10% fetal bovine serum, MEM non-essential amino acids (Life Technologies), 1 mM sodium pyruvate, and antibiotics (100 units ml$^{-1}$ penicillin and 100 μg ml$^{-1}$ streptomycin) at 37 °C in a humidified incubator containing 5% CO$_2$. For transfection, 6-cm culture dishes were coated during incubation for 10 min at 37 °C with 2.5 ml of Matrigel solution (approximately 10 μg ml$^{-1}$ growth factor-reduced Matrigel (BD Biosciences) in culture medium). Cells were seeded into the 6-cm dishes containing Matrigel solution at a density of 3.5 × 10$^6$ cells/dish. After 4 h, expression constructs (total 10 μg/dish) were transfected into the cells using PLUS (10 μl/dish) and Lipofectamine LTX (12 μl/dish) reagents. Gα$_{oA}$, Venus 1–155 Gβ, Venus-156–239-Gγ2, masGRK3ct-Nluc, and nanobody constructs were used at a 2:1:1:1:12 ratio (ratio 1 = 0.42 μg of plasmid DNA). Empty vector pcDNA3.1(+) was employed to normalize the amounts of transfected DNA.

Cellular measurements of BRET between Venus-Gβγ and masGRK3ct-Nluc-HA were performed to examine the Gβ selectivity of Nb5 in living cells. Sixteen to twenty-four hours post-transfection, HEK293T/17 cells were washed once with BRET buffer (PBS containing 0.5 mM MgCl$_2$ and 0.1% glucose) and detached by gentle pipetting with BRET buffer. Cells were harvested by centrifugation at 500 $g$ for 5 min and resuspended in BRET buffer. Approximately 50,000–100,000 cells per well were distributed in 96-well flat-bottomed white microplates (Greiner Bio-One). The Nluc substrate, furimazine, purchased from Promega, was used according to the manufacturer's instructions. BRET measurements were made with a micro plate reader (POLARstar Omega; BMG Labtech). All measurements were performed at room temperature. The BRET ratio was determined by calculating the ratio of the light emitted by the Venus-Gβγ (535 nm with a 30 nm band path width) over the light emitted by the masGRK3ct-Nluc-HA (475 nm with a 30 nm band path width). BRET assays for real-time monitoring of cAMP and Ca$^{2+}$ were performed in the same manner as described above. At approximately 16–24 h post-transfection, cells were stimulated with either 100 μM acetylcholine (ACh) or dopamine. Full blots are shown in Supplementary Figure 4a.

**Stereotaxic injection.** All animal procedures were performed with the approval of the Institutional Animal Care and Use Committee at Case Western Reserve University. To allow the expression of G protein-coupled inwardly rectifying potassium channels (GIRKs, Kir3.2) in striatal MSNs, wild type (WT) C57BL6 mice (Jackson Laboratory) were injected with adeno-associated virus encoding mGIRK2 (AAV. GIRK2, TdTomato) into the dorsal striatum at postnatal day 21. Stereotaxic injections were performed while mice were under anesthesia. AAV.GIRK2, 300 nL, was injected into one hemisphere of the dorsal striatum. The injection coordinates were: AP + 1.15 mm, ML + 1.825 mm, DV −3.325 mm (relative to the bregma). Animals were allowed to recover for ~3 weeks after surgery.

**Brain slice preparation.** Three weeks after surgery, mice were euthanized and coronal brain slices containing the striatum were made in ice cold cutting solution containing 75 mM NaCl, 2.5 mM KCl, 6 mM MgCl$_2$, 0.1 mM CaCl$_2$, 1.2 mM NaH$_2$PO$_4$, 25 mM NaHCO$_3$, 2.5 mM D-glucose, and 50 mM sucrose bubbled with 95% O$_2$ and 5% CO$_2$. Slices were incubated for 1 h at 32 °C in artificial cerebrospinal fluid (ACSF) containing: 126 mM NaCl, 2.5 mM KCl, 1.2 mM MgCl$_2$, 2.4 mM CaCl$_2$, 1.2 mM NaH$_2$PO$_4$, 21.4 mM NaHCO$_3$, 11.1 mM D-glucose and 10 μM MK-801 bubbled with 95% O$_2$ and 5% CO$_2$. Slices then were transferred to a recording chamber and perfused with ACSF at 32 °C containing picrotoxin (100 μM) and DNQX (10 μM).

**Electrophysiological recordings.** Striatal neurons were visualized with an Olympus BX51 fluorescence microscope, and GIRK2 expressing MSNs were identified by the presence of tdTomato. Electrical stimulation was applied with a monopolar extracellular stimulating electrode filled with ACSF. A single stimulation (1 ms, 20–35 μA) was used to evoke the release of neurotransmitters in the striatum. Whole-cell voltage clamp recordings were performed with an Axopatch 200B amplifier. Patch pipettes were filled with an internal solution containing 135 mM D-gluconate(K), 20 mM NaCl, 1.5 mM MgCl$_2$, 10 mM HEPES(K), pH 7.4, 10 mM BAPTA-tetrapotassium, 1 mg ml$^{-1}$ ATP, 0.1 mg ml$^{-1}$ GTP, 1.5 mg ml$^{-1}$ phosphocreatine, and 10 μM of either Nb5 or Nb17 (275 mOsm). Recordings were acquired with Axograph X (Axograph Scientific) at 10 kHz and filtered to 2 kHz for analysis. Neurons were held at −60 mV. No series resistance compensation was used, and neurons were discarded if the series resistance exceeded 15 MΩ.

**Cell culture and transfection.** A Chinese hamster ovary cell-apelin receptor (CHO-APJ) stable cell line (American Type Culture Collection, Manassas, VA) was established in the lab. Briefly, human apelin receptor (APJ) cDNA was synthesized (GenScript, Piscataway, NJ) and inserted into pLNCX2 (catalog no. 631503, Clontech, Inc.). Stable cell lines were generated by viral infection and G418 selection at a concentration of 800 μg ml$^{-1}$. For expression of Nb5 and Nb17 in mammalian cells, cDNAs of the nanobodies were sub-cloned into pcDNA3.1 (+) (Catalog no: V79020, Thermo Fisher Scientific, Inc) with a Flag tag at the C-terminus. CHO-APJ cells were transiently transfected with lipofectamine 2000 (Invitrogen, Carlsbad, CA). All cells were incubated overnight at 37 °C in serum-free medium containing 1% BSA before treatment with 1 μm apelin or a vehicle control. For immunoblotting, proteins on the blots were detected with the indicated antibodies using the Odyssey Fc Imaging System (LI-COR Biosciences, Lincoln, NE). The intensity of the bands was quantified using Image Studio Lite 5.2 software. Full blots are reported in Supplementary Figure 4b.

**cAMP assay.** CHO-APJ cells expressing Nb5 or Nb17 were plated onto a 96-well plate at 80,000 cells/well and incubated at 37 °C overnight in 5% CO$_2$. On the day of the cAMP assay, cells were treated with IBMX at a final concentration of 750 μM for 20 min at room temperature. Cells then were treated with forskolin (20 μM), together with apelin-13 at various concentrations for 10 min. After these reactions, cells were lysed and their intracellular cAMP levels were measured according to the procedures of the CatchPoint Cyclic-AMP Fluorescent Assay Kit (Molecular Devices, Sunnyvale, CA USA). Fluorescence with excitation/emission at 530/590 nm was read with a Flexstation3 plate reader (Molecular Devices).

**Statistics.** Differential HDX of Gβ$_1$γ$_1$ alone and the Gβ$_1$γ$_1$-Nb5 complex was performed on three independent repeats and statistical significance was determined with two-tailed student's $t$-test where $P$ values of <0.01 were considered significant. The BRET assays were performed on six replicates where one-way ANOVA with Tukey's post hoc multiple comparison test was applied relative to the Gβ$_1$γ$_2$/GRK3ct control. The $P$ values of <0.001 were considered significant and the results were expressed as mean ± SEM. Nb5 electrophysiological recordings were obtained from nine striatum neurons for the measurement of D2R-IPSCs and eight neurons for the measurements of M4R-IPSCs. Nb17 electrophysiological recordings were obtained from five striatum neurons for the measurement of D2R-IPSCs. The amplitudes obtained from individual recordings were then averaged and the significance ($P < 0.05$) was determined with two-tailed student's $t$-test. The real-time monitoring of Ca$^{2+}$ was performed on six replicates where two-way ANOVA with Tukey's post hoc multiple comparison test was applied. The BRET-based cAMP assay was performed on three independent experiments where two-way ANOVA with Tukey's post hoc multiple comparison test was applied.

**Data availability statement.** Data supporting the findings of this manuscript are available from the corresponding author upon reasonable request. Coordinates and structure factor amplitudes of the Gβ$_1$γ$_1$-Nb5 complex are deposited in the PDB (PDB accession: 6B20). The mass spectrometry proteomics data have been deposited to the ProteomeXchange Consortium via the PRIDE partner repository with the dataset identifier PXD009503.

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

## Acknowledgements

We thank Dr. Leslie T. Webster, Jr. and members of the Palczewski laboratory for their helpful comments regarding this manuscript. We thank Dr. Hossein Heidari Torkabadi, Eva Beke, Nickolas K. Skamangas, and Dr. Yinghua Chen for technical assistance with mouse brain isolation, nanobody production, western blotting and SPR, respectively. We also thank INSTRUCT, part of the European Strategy Forum on Research Infrastructures and the Research Foundation—Flanders (FWO) for their nanobody discovery support. This research was supported in part by grants from the National Institutes of Health [EY009339, EY027283, and EY024864 (K.P.); and DA35821 and NS95809 (C.P.F.)], core grants [P30EY011373 and P30EY025585 (P.D.K.)], and the Department of Veterans Affairs Grant [IK2BX002683 (P.D.K.)]. This work is based upon research conducted at the Northeastern Collaborative Access Team beamlines, which are funded by the National Institute of General Medical Sciences from the National Institutes of Health (P41 GM103403). The Eiger 16M detector on 24-ID-E beam line is funded by a NIH-ORIP HEI grant (S10OD021527). This research used resources of the Advanced Photon Source, a U.S. Department of Energy (DOE) Office of Science User Facility operated for the DOE Office of Science by Argonne National Laboratory under Contract No. DE-AC02-06CH11357. K.P. is the John H. Hord Professor of Pharmacology.

## Author contributions

S.G. and K.P. designed the research and wrote the paper; S.G. and H.J. performed the cell transfections and cAMP assays. I.M. and K.A.M. performed the BRET assays and contributed to the written paper. Y.C. and C.P.F. did the electrophysiological recordings and contributed to the written paper. E.P. and J.S. performed the nanobody discovery and contributed to the written paper. S.G. and T.O. carried out the differential HDX data acquisition and analyses. S.G. performed all the biochemical assays and crystallized the $G\beta_1\gamma_1$-Nb5 complex. S.G. did all the protein purifications and mass spectrometry analyses. S.G. and P.D.K. analyzed the x-ray diffraction data. P.L.S. made intellectual contributions to the study and contributed to the written paper. All authors have reviewed and edited the manuscript.

## Additional information

**Competing interests:** The authors declare no competing interests.

