## [Peer Review File · Nature Communications]

Reviewers' comments:

Reviewer #1 (Remarks to the Author):

This paper describes and characterizes a novel nanobody, Nb5, which binds G $\beta\gamma$ with high affinity (~40 nM) at an interface that partially overlaps the G α - and effector-binding surface of G $\beta\gamma$. The nanobody interacts with all G β subunits except G $\beta 5$. Importantly, Nb5 promotes dissociation of G α -GDP from G $\beta\gamma$. As expected for a G $\beta\gamma$ "scavenger", Nb5 partly inhibits the association of G $\beta\gamma$ with GRK3, the activation of ectopically expressed GIRK channels by two GPCRs in striatal medium spiny neurons, and the activation of Akt and ERK1/2 by the apelin/APJ via a G $\beta\gamma$ /PI3K-dependent pathway. On the other hand, it is proposed that Nb5 does not interfere with G α -GTP mediated signaling, as indicated by preservation of almost 60% of G αi -induced inhibition of forskolin-activated adenylyl cyclase in APJ-expressing cells.

The part of the study describing the Nb5 discovery, production as a recombinant protein, characterization of its effect on Gt heterotrimer, and its interaction and complex with G $\beta\gamma$, including crystal structure and mass spectroscopy, are impeccable. Inhibition of GIRK channels is also clear-cut. It is safe to say that the paper clearly establishes this novel nanobody as a new interactor of G β with distinct G $\beta\gamma$ -binding and G $\beta\gamma$ -scavenging activity. The development and meticulous characterization of this reagent are important tools for future mechanistic and clinical studies.

On the side of mechanism, the authors point out two major novel aspects. The first new mechanism is the promotion of dissociation of G α -GDP from G $\beta\gamma$ by Nb5. Fig. 1 presents strong evidence supporting this mechanism. I wonder, however, how unique this Nb5 effect is. Is it known, or have the authors tested, whether other G $\beta\gamma$ binding partners/scavengers of similar affinity, such as phosducin or β ARK, do not act similarly? If this is the case, Nb5 is not as unique as suggested.

The major mechanistic highlight of the study is the suggestion that Nb5 is a selective tool to inhibit G $\beta\gamma$ signaling without affecting G α signaling. The main supporting evidence is the (expected) inhibition of GIRK signaling (Fig. 5) and the data of Fig. 6. The latter is an important figure but, unfortunately, the weakest in terms of data. First, in Fig. 6b, it looks like even Nb17 reduces phosphorylation compared to control (no Nb). Second, there is no quantitative summary of the effect of Nb5 on phosphorylation of ERK and Akt. The extent of inhibition is unclear. Third, it is not shown whether Nb5 affects in any way the activation of AC (cAMP accumulation), only normalized data are shown. Finally, it is unclear why cAMP results of Fig. 6 are reported in Discussion. Most importantly, it appears to me that the proposed mechanism, in which G $\beta\gamma$ signaling branch is silenced whereas the G α signaling is intact, is not sufficiently supported by data and may be an overstatement. It needs to be discussed how GPCR-G α signaling can proceed in the absence of G $\beta\gamma$; can G α be activated by the receptor in the absence of G $\beta\gamma$, how this process will be terminated and "restarted" on repetitive exposures to agonist/transmitter? The presented results may be explained in a simpler and more conventional way. Indeed, Nb5 attenuates the G $\beta\gamma$ -mediated GIRK signaling by ~50-60%, indicating that at least 40% of G $\beta\gamma$ remains available for re-association with G α . Similarly, G $\beta\gamma$ -mediated signaling to ERK and Akt is reduced (by how much?) but not abolished. Thus, Nb5 produces partial inhibition of G $\beta\gamma$ signaling pathways, by partial dynamic scavenging of G $\beta\gamma$. The Nb5-free G $\beta\gamma$ may suffice to support much of the canonical signaling that includes dissociation of G $\beta\gamma$ and the concomitant GDP-GTP exchange, and later termination by GTP hydrolysis and reassociation of G α -GDP with G $\beta\gamma$. In the same line, the 58% decrease in cAMP accumulation is interpreted as G αi -mediated component after selective "loss of the G $\beta\gamma$ -mediated cAMP signaling component"; there is no data to support the latter assertion, because the initial contribution of the G $\beta\gamma$ -dependent component, as well as extent of loss of G $\beta\gamma$ from the system, have not been estimated in this system.

In view of lack of selectivity toward different G β subunits and uncertainty as to the intactness of GPCR signaling via G α , I think it is premature to declare that Nb5 is "a versatile tool to achieve

selective modulation of GPCR-signaling”.

Minor technical points.

1. Line 98: define mAb.
2. Fig. 1 legend: “...initial rate of activation by Rh*” (not of Rh*)
3. Fig. 3 legend: maybe it is possible to better define the color code in “e”; does black numbering refer to residues in contact with Nb5 only, or also with Ga?
4. Fig. 4e: the maximum BRET signal was determined by co-transfection with Gβ1γ2? Or the different Gβ’s + Gγ2? “Experiments were performed with Gβ1, Gβ2, ...” – unclear.
5. line 209: residues 80-99 and 111-118 of Gβ1 (not Gβ1γ1).
6. Line 271: Activation of GIRKs affects the flow of K⁺ ions.

Reviewer #2 (Remarks to the Author):

Gulati et al. developed a llama-derived nanobody (Nb5) against Gβγ dimer, which induces dissociation of Gβγ dimer from Ga subunit. The authors identified NB5-binding interface on Gβ by HDX-MS and X-ray crystallography; the binding interface is a common binding site (so-called Gβγ hot spot) for Ga subunit, GRK, and GIRK. Nb5 binds Gβ1-4 and inhibits Gβ signaling while it does not affect Ga signaling. The author claims that NB5 has a therapeutic potential as it selectively inhibits Gβγ signaling whereas GPCR-targeting drugs inhibit both Ga and Gβγ signaling. Overall, this study is well-designed for the nanobody development and identification of the binding interface. However, a few concerns are raised about the effects of Nb5 on Gβγ signaling and the therapeutic potential of Nb5.

Here are detailed comments.

1. The main concern is that the authors claimed that inhibiting Gβγ has a better therapeutic potential over GPCR-targeting drugs. However, would targeting Gβγ be more promiscuous? It would affect signal transduction of all GPCRs. Moreover, in the G protein signaling cycle, Ga activation is terminated when GTP is hydrolyzed to GDP not solely by re-interaction with Gβγ. Without forming a heterotrimer with Gβγ, the efficacy of GPCR-mediated GDP release from Ga is lowered, and GDP-bound Ga is less stable as a monomer than heterotrimer. Therefore, Nb5 can also inhibit Ga activity.
2. In page 6 first paragraph, it is written: “all 3 Gβ1γ1-positive Nbs belonged to the same nanobody family with an identical complementarity determining region 3 (CDR3) and displayed similar biochemical properties.” Please provide appropriate data or references.
3. In page 7 line 99, please change Fig. 2b to Fig. 2a.
4. In Fig 4a, please provide the sequence of Gβ5.
5. In Fig 4b, what are the Nb5-coeluted proteins near 50 kDa? Are these proteins Nb5-binding non-Gβ proteins?
6. In Fig 4e, please provide the expression levels of Gβ, GRK3, Ga, Nb5, and Nb17.
7. In Fig 6b, when lane 2 is compared to lane 6, it suggests that Nb17 also inhibits ERK and AKT phosphorylation. Therefore, it is not correct to state “CHO-APJ-Nb17 cells showed no visible change in phosphorylation levels of either AKT or ERK proteins during the same time interval”
8. It is well known that Gai can also trigger ERK activation. Please provide experimental data or references whether apelin-induced ERK or AKT activation is mediated by Gai or Gβγ.
9. In Fig 6d, please provide control data that does not express Nb5 or Nb7.
10. In the text, the Results section ends without explaining Fig 6c and 6d, and the explanation for these figures are in the Discussion section. Please re-arrange the text and provide the explanations of Fig 6c and Fig 6d in the Result section.
11. Based on the references that the authors provided, Gβγ stimulates adenylyl cyclase. It would be easy for the readers if the authors directly state that Gβγ stimulates adenylyl cyclase rather than “influence the activity of several AC”.
12. In page 12, “But Nb5 had no effect on the Ga-GTP-mediated cAMP signaling component as evidenced by its maintenance after the loss of the Gβγ-mediated cAMP signaling component in

cells transfected with Nb5 (Fig. 6d, greencyan).” However, Fig 1d demonstrates that pre-incubation of Gt with Nb5 decreases the initial Gt activation rate. Would these two results show inconsistency? Moreover, I am not convinced that Fig 6d demonstrate that Nb5 inhibits the AC stimulatory effect of Gβγ and not affect Gai. The authors stated in the Abstract “Gβγ also serves as a negative

62 regulator of Ga that is essential for G protein inactivation”. If Nb5 prevents Gβγ from binding to Ga, would this increase the Ga activity?

13. With protein identification by mass spectrometry-based proteomics in Fig 4, are there proteins identified other than Gβ proteins? Please provide detailed database search parameters in the Methods and a full list of identified proteins.

14. In page 29, “Note the decrease in Gy1 Coomassie staining in the presence of 300 mM imidazole during protein elution.” Please explain why the Gy1 Coomassie staining is decreased in the presence of 300 mM imidazole.

15. Please provide PDB ID for Suppl Fig 1c.

Reviewer #3 (Remarks to the Author):

In these studies a llama-derived nanobody (Nb5) was identified that tightly binds to the Gβγ dimer. Differential hydrogen/deuterium exchange and crystallography studies suggest that Nb5 binds a common binding site for the Gβγ dimer.

Interestingly, Nb5 has affinity for combinations of β- and γ-subtypes, allowing a broad applicability. Nb5 impairs the activation of both the Gβγ-regulated G-protein-gated potassium channels and the Gβγ-mediated PI3K and MAPK pathways. The authors claim that Nb5 has an inhibitory effect on Gβγ-mediated signaling, and no effect on Ga-GTP-mediated signaling, therefore serving as a versatile tool to achieve selective modulation of GPCR-signaling.

The nanobody described in these studies, Nb5, is an interesting tool, that may serve to further characterize the role of Gβγs in GPCR mediated signaling. Complementary approaches are used to characterize binding of this nanobody to Gβγs. However, the claims made with respect to the lack of Nb5 effects on Galpha mediated signaling need controls and experiments to further substantiate this claim.

Major points:

- Nb5 is suggested to only affect Gβγ signaling and leave Galpha signaling intact. The authors state that since the apparent affinity of Nb5 is lower than the affinity of the Galpha unit to the Gβγ dimer that it cannot affect Galpha signaling (line 244). This statement is not true. At high concentrations of Nb5, Nb5 may still inhibit Galpha signaling. Overexpression of the Nb inside the cell could therefore affect Galpha signaling.

- To show that Nb5 has no effect on Galpha mediated signaling include an experiment (Galpha dependent) showing that with increasing concentrations of Nb5 no effect on Galpha signaling is apparent.

- In figure 5 the authors showed the effects of Nb5 on D2 and M4 mediated GIRK signaling. The authors should include control experiments showing signaling in the absence of Nb5 and Nb17 too. Overexpression of proteins may affect the protein machinery of cells and indirectly affect signaling.

- The authors should include control experiments in figure 6D showing cAMP signaling in the absence of Nb5 and Nb17 too. Is cAMP signaling in the absence of Nbs similar as co-expression with Nb17 only?

- In figure 6 the authors showed that the effects of Nb5 on apelin-mediated phosphorylation of AKT and ERK in CHO cells. Both Nb5 but also Nb17 appear to impair phosphorylation of Akt and ERK, when compared to cells not expressing Nb5 nor Nb17. Include quantification of blots. How does one explain the decrease of apelin-induced phosphorylation of Akt and ERK in Nb17 expressing cells?

- Does expression of Nb5 and Nb17 affect expression of D2/M4 or apelin receptors?

Minor comments:

- Include the results of figure 6d into the results section.

- No referral has been made to recent studies (Ghosh et al 2017) using nanobodies targeting barrestin in the introduction. These studies are also an example of the use of nanobodies targeting GPCR signaling rather than receptors.

Reviewer #4 (Remarks to the Author):

The manuscript entitled

The manuscript entitled "Targeting G protein-coupled receptor signaling at the G protein level with a selective nanobody inhibitor" was an extremely well written article and contained some very important and novel findings. The authors show convincing data that they have selected a nanobody (Nb5) that targets the G β gamma dimer specifically. This nanobody was able to inhibit G β gamma subunits 1-4 but not 5 and also inhibit the regulation of GIRK channels. The use of SPR and BRET convincing supports the conclusions of the authors. Unlike the nanobodies that are progressing through the clinic Nb5 has to operate inside the cell. This makes the therapeutic role for this nanobody more complicated, however the last part of the discussion provides a clear-eyed summary of the difficulties here.

I have only a few minor comments;

1) It would have been nice to see the sequences of the nanobodies representing the 14 families perhaps as a supplementary figure (or at least the sequence of the Nb5.

2) The authors state on page 4 that "recognition. However, only nanobodies (Nbs) derived from the variable region of camelid 106 heavy chain are endowed with favorable characteristics in terms of size, solubility, affinity and 107 ease of production" The use of the word only here is rather definitive and I believe that the single domain IgNARs and the recently reported i-body (Griffiths et al J. Biol. Chem. 2016, 291, 12641–12657) could claim to have similar properties

Reviewer #5 (Remarks to the Author):

The report by Gulati et.al. entitled "Targeting G protein coupled receptor signaling at the G protein

level with a selective inhibitor" details the development of a nanobody (Nb5) that can interact with G protein beta-gamma subunits. This shifts the equilibrium of the G protein heterotrimer to be disassociated and selectively inhibits beta-gamma mediated cell signaling. As beta-gamma mediated cell signaling is an important communication pathway in the cell. This signal biasing tool might be useful to understand and develop new models for cell signaling. However, there are a number of issues with this report.

Major concerns:

- The authors do not actually use this tool to address any biological questions on signaling in the context of this report, so they have not proven that this nanobody will be useful to the field.
- There are already Gbg specific inhibitors reported in the literature such as Gallein, M119, M119K. This decreases the novelty of the nanobody reported here. A comparison of Nb5 and these known inhibitors would be important to show whether Nb5 has advantages.

Concerns with data interpretation:

- The authors suggest that there is a therapeutic potential of this inhibitor. However, they also state that "Nb5 responds to all combinations of β - and γ -subtypes, indicating its broad applicability to various cell types". This may not be consistent with therapeutic applications.

- This study shows a direct interaction between Nb5 and the G-beta 1 subtype (by using X-ray crystallography, HDX and SPR). The interaction between Gb2-4 and Nb5 is shown indirectly and does not test all of the Gbg combinations. They are assuming Nb5 will interact and inhibit all Gbg mediated cell signaling. This needs to be clarified.

- Because of Nb5s exhibits some nonspecific behavior, the authors expect broad applicability to various cell types and cellular signaling. If Nb5 affects various cell types and signaling pathways, it will also increase potential side effects of Nb5 as a therapeutic usage. Until they develop Gbg subtype specific nanobody, the idea of therapeutic potential of nanobody on Gbg signaling may not be feasible.

- In figure 1a, why isn't the Ggamma subunit in the Rh+GTP+Nb5, GTP+Nb5, Nb5 conditions as Nb7 and control?

- In figure 1, the G alpha t nucleotide exchange looks extremely slow compare to Gai or Gas. In the presence of Gbg, nucleotide exchange should be slower than with just the alpha subunit. In figure 1d, it looks like nucleotide exchange is reaching more than 65-75% of the maximum level less than 3 mins in the dark and in the presence of Nb5 (which is practically the rate of Galphat in this study). Why the exchange rate is so fast in the dark and in the presence of Nb5? It would be nice to show the raw data without normalization.

- The BRET signal in the presence of Nb5 statistically decreases compared to Nb7 in Figure 4. However, it is still significantly higher than in the presence of the Galpha subunit, which indicates that Gbg can still interact with GRK in the presence of Nb5. This result contradicts with Figure 5, which is showing at least 50% GIRK channel inhibition with Nb5. This discrepancy needs to be clarified.

- The effect of Nb5 on the cellular pathways were investigated in figure 6. Compared to control (without any NB) experiments, ERK and AKT phosphorylation significantly decreases not just in the presence of Nb5 but also with Nb7 (figure 6b). This result is in opposition to the rest of the data, which are consistent with Nb7 not showing any inhibitory effects. Are there any non-specific interactions between the Nb7 and the other signaling proteins in the cell? If so, this then raises a question about if there are any non-specific interaction between Nb5 and the other proteins in the cell.

•Recognizing the ongoing debate in the field on how to define resolution of crystal structures, it could be worth reporting in the legend to supplementary table 2 the resolution of data where more traditional data collection statistics are observed (for example $I/\sigma > 2$ at a resolution of "x") as well as describe the rationale for cutting the data at the proposed resolution. By giving a statement like this, it helps readers to compare the accuracy of the model to more historic standards.

Reviewer #6 (Remarks to the Author):

In this manuscript, Gulati et al. identify a nanobody (Nb5) that binds Gbeta/gamma dimers with high affinity to prevent the interaction of dimers to Galpha-t and downstream effectors. The data strongly support these conclusions and the manuscript is very well written. On the other hand, discussions regarding the significance of the work with respect to previous studies seems unnecessarily narrow. That is, the hotspot on Gbeta/gamma dimers that is bound by Nb5 is also used by affinity-matured peptides and small molecules to modulate signaling mediated by Gbeta/gamma dimers (1,2). A detailed discussion of the merits of the various methods to target Gbeta/gamma dimers seems warranted. This discussion should also include the use of the C-terminal domain of GRK2 (bARK-ct). This genetically-encoded domain effectively sequesters Gbeta/gamma dimers in much the same fashion as Nb5 and has been used for approximately two decades to silence signaling by Gbeta/gamma dimers. What preferential uses do the authors envision for Nb5 relative to bARK-ct?

1. Bonacci TM, Mathews JL, Yuan C, Lehmann DM, Malik S, Wu D, Font JL, Bidlack JM and Smrcka AV (2006). Differential targeting of Gbeta/gamma-subunit signaling with small molecules. *Science* 312, 443-446.
2. Goubaeva F, Ghosh M, Malik S, Yang J, Hinkle PM, Griendling KK, Neubig RR and Smrcka AV (2003). Stimulation of cellular signaling and G protein subunit dissociation by G protein beta/gamma subunit-binding peptides. *J Biol Chem* 278, 19634-19641.

Minor considerations:

1. In the abstract, the authors state that "Nb5 has no effect on Galpha-GTP signaling events..." This statement does not seem to be supported by the data in the sense that sequestration of Gbeta/gamma should prevent GPCR-mediated activation of Galpha subunits. Indeed, this property is presumably the cause of the depression in rhodopsin-catalyzed activation of Galpha-t (Fig. 1d-e). Please clarify.
2. In the introduction that authors state that, "While GPCRs serve as the largest class of drug-targeted membrane proteins, only a handful of GPCR-mediated signaling events have been targeted therapeutically with either small molecules or peptide modulators." While technically true, this statement is somewhat misleading since drugs that target GPCRs continue to comprise the largest set of drugs based on target class. Indeed, the reference used by the authors to support their statement refers to GPCRs as a "privileged target family." Consider rewording.
3. The description of the methods for the SPR experiments is absent and should be added. Also, SPR is prone to several artefacts arising from the requirement to affix the ligand to the sensor chip. It is best practice to measure affinities using each partner as the affixed ligand with the expectation that measure affinities will not change. Unfortunately, most experimenters do not carry out this control and it would be unreasonable to require it here. However, had it been done, Gbeta/gamma would be the ligand and the setup would have easily allowed the direct measurement of Galpha-t binding to Gbeta/gamma. Instead, the authors reference a measure of

the affinity between heterotrimer subunits based on fluorescent proteins at membranes. This measurement is likely confounded by the experimental design and the authors should find a more appropriate reference. Finally, it seems the authors missed a simple experiment to verify the direct competition of Galpha-t and Nb5 for binding to Gbeta/gamma. Pre-incubation of Gbeta/gamma with varying concentration of Galpha-t-GDP would be expected to reduce sensorgram responses; conversely, pre-incubation with Galpha-t-GDP in the presence of aluminum fluoride should have no effect on the sensorgrams.

4. The crystal structure is of high quality. In light of the structural work, it probably makes sense to relegate the HDX-MS to supplemental information, especially since it is not clear why some regions experience increased exchange and other show decreased exchange. On a related note, Galpha-t is shown on the right hand panel of Figure 2e, but not on the left. The authors should state explicitly that Galpha-t is not shown on the left for the sake of clarity.

5. Gbeta1-4 are highly conserved and in contrast to Ggamma subunits rarely, if ever, dictate preferential activation of effectors. Therefore, given that Nb5 binds exclusively to Gbeta1, it is not remarkable that "Nb5 responds to all combinations of beta- and gamma- subtypes." Perhaps reword and remove multiple sequence alignment (Fig. 4a).

6. Figure 5j is somewhat confusing. Is there a better way to indicate that the GIRK channel does not open? Right now, it appears that the channel is cycling potassium around inside the cell. Similarly, "Nb5 mediated GIRK channel activation" has a pointer on the outside of the cell. Perhaps direct the pointer closer to the action?

Response to Reviewers' comments:

We thank the reviewers for their careful reading of our manuscript and for their positive assessment of our work. We have addressed the reviewer's comments below, and have incorporated their suggestions in the revised manuscript. Reviewers' comments are in black text and our responses are in blue text.

Reviewers' comments:

Reviewer #1 (Remarks to the Author):

This paper describes and characterizes a novel nanobody, Nb5, which binds G $\beta\gamma$ with high affinity (~40 nM) at an interface that partially overlaps the G α - and effector-binding surface of G $\beta\gamma$. The nanobody interacts with all G β subunits except G $\beta 5$. Importantly, Nb5 promotes dissociation of G α -GDP from G $\beta\gamma$. As expected for a G $\beta\gamma$ "scavenger", Nb5 partly inhibits the association of G $\beta\gamma$ with GRK3, the activation of ectopically expressed GIRK channels by two GPCRs in striatal medium spiny neurons, and the activation of Akt and ERK1/2 by the apelin/APJ via a G $\beta\gamma$ /PI3K-dependent pathway. On the other hand, it is proposed that Nb5 does not interfere with G α -GTP mediated signaling, as indicated by preservation of almost 60% of G αi -induced inhibition of forskolin-activated adenylyl cyclase in APJ-expressing cells.

The part of the study describing the Nb5 discovery, production as a recombinant protein, characterization of its effect on Gt heterotrimer, and its interaction and complex with G $\beta\gamma$, including crystal structure and mass spectroscopy, are impeccable. Inhibition of GIRK channels is also clear-cut. It is safe to say that the paper clearly establishes this novel nanobody as a new interactor of G β with distinct G $\beta\gamma$ -binding and G $\beta\gamma$ -scavenging activity. The development and meticulous characterization of this reagent are important tools for future mechanistic and clinical studies.

On the side of mechanism, the authors point out two major novel aspects. The first new mechanism is the promotion of dissociation of G α -GDP from G $\beta\gamma$ by Nb5. Fig. 1 presents strong evidence supporting this mechanism. I wonder, however, how unique this Nb5 effect is. Is it known, or have the authors tested, whether other G $\beta\gamma$ binding partners/scavengers of similar affinity, such as phosducin or β ARK, do not act similarly? If this is the case, Nb5 is not as unique as suggested.

The authors appreciate the reviewer's comment. We agree it is important to include a comparison of the known G $\beta\gamma$ binding partners/scavengers of similar affinity with Nb5 for G $\beta\gamma$ binding. In response to the reviewer's comment, the following text has been added to the discussion section of the revised manuscript.

Inserted text 1: "

The M119 class of small-molecule inhibitors, which includes M119, gallein and M119K, are potent G $\beta\gamma$ antagonists. They remain the most extensively validated inhibitors of G $\beta\gamma$ signaling. Studies to date indicate that these molecules bind to a G $\beta\gamma$ "hot spot" on the top surface of the β subunit with apparent affinities in the high nM to low μ M range ¹. The M119 class of inhibitors displays a limited number of chemical moieties that interact with the G $\beta\gamma$ dimer. Therefore, structure activity relationship studies to achieve G β selectivity would have limited practicality. Nb5, on the other hand, binds G $\beta\gamma$ with a low nM affinity and has a larger area of interaction with the G $\beta\gamma$ dimer (>1030 Å²). These features indicate that Nb5 might serve as a better template for structure activity relationship studies to achieve G β selectivity. Unlike nanobodies, the M119 class of inhibitors can be delivered orally and intraperitoneally

to modulate G $\beta\gamma$ –dependent pathways, which admittedly takes them one step closer to clinical applications. Challenges remain for the development of nanobody-based clinical therapies due to the dependence of this approach on efficient gene transduction methods. The C-terminal domain of GRK2 (β ARK-ct), PDC and several affinity matured peptides, including the SIRK peptide family, comprise an alternative class of potent G $\beta\gamma$ inhibitors that are mechanistically similar to Nb5²⁻⁴. While β ARK-ct and PDC have a large area of interaction with the G $\beta\gamma$ dimer (>1000 Å²), no atomic level information is available for SIRK or other peptides except that they bind to the same G $\beta\gamma$ “hot spot” with high nM affinities^{4,5} and selectively inhibit the binding of G $\beta\gamma$ effector molecules. Several cell-permeating versions of SIRK peptides have been shown to effectively modulate ERK1/2 and MAPK signaling in arterial smooth muscle cells³. Treatment strategies involving Nb5, β ARK-ct, PDC and members of the SIRK peptide family would all require gene transduction methods, however this may be worthy of significant effort given their potential to treat various G $\beta\gamma$ -related disorders.

Inserted text 2: “The ease of production and genetic manipulation of nanobodies are advantageous characteristics for achieving the goal of engineering variants with specificity toward different G β subtypes and even G $\beta\gamma$ combinations. This opens up the possibility of designing targeted treatments for various excitatory neurological conditions and inhibitors for blocking cancer progression.”

1. The major mechanistic highlight of the study is the suggestion that Nb5 is a selective tool to inhibit G $\beta\gamma$ signaling without affecting G α signaling. The main supporting evidence is the (expected) inhibition of GIRK signaling (Fig. 5) and the data of Fig. 6. The latter is an important figure but, unfortunately, the weakest in terms of data. First, in Fig. 6b, it looks like even Nb17 reduces phosphorylation compared to control (no Nb).

The authors thank the reviewer for raising this point. The apelin receptor mediated phosphorylation changes in PI3K-PKB/AKT and MAPK/ERK pathways were studied for both parental and nanobody co-expressing CHO-APJ cells. However, these experimental groups were not studied simultaneously, and therefore Nb17 co-expressed CHO-APJ cells seem to have reduced phosphorylation of AKT and ERK1/2 proteins compared to the parental CHO-APJ cells. This inconsistency could be explained by the difference in the confluency of cells (total number of cells loaded on the gels) and the difference in the exposure time of HRP-conjugated secondary antibodies between the experimental groups.

In response to the reviewer’s comment, the effect of nanobodies on the phosphorylation of both AKT and ERK proteins was re-examined in parental CHO-APJ cells incubated with transfection reagents only, CHO-APJ cells transfected with pcDNA3.1 (+) empty vector, CHO-APJ-Nb17 and CHO-APJ-Nb5 cells, simultaneously. These results show that Nb17 has no significant effect on the phosphorylation of AKT and ERK proteins as compared to either parental CHO-APJ cells or CHO-APJ cells transfected with pcDNA3.1 (+) empty vector. However, a significant decrease in the phosphorylation of both AKT and ERK1/2 proteins was observed in CHO-APJ-Nb5 cells 5 min post-treatment with apelin. These results have been incorporated in Fig. 6b of the revised manuscript.

2. Second, there is no quantitative summary of the effect of Nb5 on phosphorylation of ERK and Akt. The extent of inhibition is unclear.

The quantification of western blots has been included in Fig.6 of the revised manuscript.

3. Third, it is not shown whether Nb5 affects in any way the activation of AC (cAMP accumulation), only normalized data are shown.

The activation of AC or cAMP accumulation was directly achieved by treating CHO-APJ cells with 20 μ M forskolin. The forskolin-stimulated accumulation of cAMP was then measured in response to apelin treatment to determine the effect of Nb5 on intracellular cAMP levels. Nb5 does not affect the forskolin-mediated activation of AC when compared to either Nb17 or parental cell line.

In response to the reviewer's comment, the raw data for the cAMP assay has been added as Supplementary Table 4 of the revised supplementary information.

4. Finally, it is unclear why cAMP results of Fig. 6 are reported in Discussion.

The authors concur with the reviewer. The results explaining Figure 6c and 6d have been moved to the results section in the revised manuscript.

5. Most importantly, it appears to me that the proposed mechanism, in which G $\beta\gamma$ signaling branch is silenced whereas the G α signaling is intact, is not sufficiently supported by data and may be an overstatement. It needs to be discussed how GPCR-G α signaling can proceed in the absence of G $\beta\gamma$; can G α can be activated by the receptor in the absence of G $\beta\gamma$, how this process will be terminated and "restarted" on repetitive exposures to agonist/transmitter? The presented results may be explained in a simpler and more conventional way. Indeed, Nb5 attenuates the G $\beta\gamma$ -mediated GIRK signaling by ~50-60%, indicating that at least 40% of G $\beta\gamma$ remains available for re-association with G α . Similarly, G $\beta\gamma$ -mediated signaling to ERK and Akt is reduced (by how much?) but not abolished. Thus, Nb5 produces partial inhibition of G $\beta\gamma$ signaling pathways, by partial scavenging of G $\beta\gamma$. The Nb5-free G $\beta\gamma$ may suffice to support much of the canonical signaling that includes dissociation of G $\beta\gamma$ and the concomitant GDP-GTP exchange, and later termination by GTP hydrolysis and reassociation of G α -GDP with G $\beta\gamma$. In the same line, the 58% decrease in cAMP accumulation is interpreted as G α_i -mediated component after selective "loss of the G $\beta\gamma$ -mediated cAMP signaling component"; there is no data to support the latter assertion, because the initial contribution of the G $\beta\gamma$ -dependent component, as well as extent of loss of G $\beta\gamma$ from the system, have not been estimated in this system. In view of lack of selectivity toward different G β subunits and uncertainty as to the intactness of GPCR signaling via G α , I think it is premature to declare that Nb5 is "a versatile tool to achieve selective modulation of GPCR-signaling".

The authors thank the reviewer for his/her positive assessment of our work. The reviewer is correct about the possibility that Nb5 might indirectly affect G α signaling due to the sequestration of G $\beta\gamma$ by Nb5. This study introduces Nb5 as a dynamic scavenger of G $\beta\gamma$ dimer that results in a partial inhibition of both G $\beta\gamma$ -mediated GIRK activation (58% inhibition) and the downstream AKT and ERK phosphorylation (30-40% inhibition). As the reviewer mentioned, the remaining Nb5-free G $\beta\gamma$ dimers might suffice to support the canonical G protein signaling that includes dissociation of G $\beta\gamma$ and the concomitant GDP-GTP exchange from G α , and later termination by GTP hydrolysis and re-association of G α -GDP with G $\beta\gamma$.

However in response to the reviewer's comment, we performed experiments to monitor the effect of Nb5 on agonist-induced G α signaling. G α_q signaling was monitored in HEK293T/17 cells transfected with the m3 muscarinic acetylcholine receptor (M3R) and using a Ca²⁺ sensor (CalFluxVTN). G α_s signaling was monitored in HEK293T/17 cells transfected with the dopamine D1 receptor (D1R) and using a cAMP sensor (Nluc-Epac-VV). The transfected cells were then stimulated with acetylcholine or dopamine to activate M3R or D1R, respectively. Real-time monitoring of the resultant responses clearly showed no effect of Nb5 on the elevation of intracellular Ca²⁺ induced by M3R-G α_q or on D1R-G α_s -induced cAMP production. These results show that Nb5 does not alter G α_q and G α_s signaling in a living cell. These findings also are consistent with the higher affinity of G α compared to Nb5

for the Gβγ dimer, supporting the inability of Nb5 to affect Gα signaling. However at higher concentrations, Nb5 might affect Gα signaling by increased sequestration of Gβγ as shown in Fig. 1d-e.

These findings have been added in the revised manuscript as Fig 6f-j. The quantification of AKT and ERK western blots has been included in Fig.6c of the revised manuscript.

Additionally for further clarification, the following text has been added to the discussion section of the revised manuscript.

Inserted text: “Additionally, Nb5 showed no significant effect on either the M3R-Gα_q-induced intracellular Ca²⁺ elevation or D1R-Gα_s-induced cAMP production. These results show that Nb5 does not alter the GTP-bound Gα_q and Gα_s-mediated signaling component in living cells. Overall, Nb5 serves as a dynamic scavenger of the Gβγ dimer and thereby causes partial inhibition of Gβγ-mediated signaling. The remaining Nb5-free Gβγ supports the canonical Gα-GTP-mediated signaling that includes the GPCR-mediated GDP-GTP exchange from Gα, and signaling termination by GTP hydrolysis and re-association of Gα-GDP with Gβγ. However at higher concentrations, Nb5 might affect the Gα-GTP signaling component due to increased sequestration of Gβγ dimers.”

Minor technical points.

1. Line 98: define mAb.

The abbreviation mAb has been defined in the revised manuscript.

2. Fig. 1 legend: “...initial rate of activation by Rh*” (not of Rh*)

The suggested correction has been incorporated into the legend of Fig.1 in the revised manuscript.

3. Fig. 3 legend: maybe it is possible to better define the color code in “e”; does black numbering refer to residues in contact with Nb5 only, or also with Gα?

The authors appreciate this suggestion. The color code has been clarified in the legend of Fig. 3 in the revised manuscript.

4. Fig. 4e: the maximum BRET signal was determined by co-transfection with Gβ1γ2? Or the different Gβ's + Gγ2? “Experiments were performed with Gβ1, Gβ2, ...” – unclear.

The maximum BRET signal was determined by co-transfection with different Gβ subtypes + Gγ₂ pairs. For further clarification, the text in the legend for Fig. 4 has been modified in the revised manuscript.

Modified text 1: “The maximum BRET signal was determined by co-transfection of different Venus-Gβ subtypes + Gγ₂ pairs and masGRK3ct-Nluc-HA (grey)”

Modified text 2: “Experiments were performed with Gβ subtypes 1-4 + Gγ₂ pairs.”

5. line 209: residues 80-99 and 111-118 of Gβ1 (not Gβ1γ1).

The authors appreciate this comment. Correction has been made in the revised manuscript.

6. Line 271: Activation of GIRKs affects the flow of K⁺ ions.

Correction has been made in the revised manuscript.

Reviewer #2 (Remarks to the Author):

Gulati et al. developed a llama-derived nanobody (Nb5) against Gβγ dimer, which induces dissociation of Gβγ dimer from Gα subunit. The authors identified NB5-binding interface on Gβ by HDX-MS and X-ray crystallography; the binding interface is a common binding site (so-called Gβγ hot spot) for Gα subunit, GRK, and GIRK. Nb5 binds Gβ1-4 and inhibits Gβ signaling while it does not affect Gα signaling. The author claims that NB5 has a therapeutic potential as it selectively inhibits Gβγ signaling whereas GPCR-targeting drugs inhibit both Gα and Gβγ signaling. Overall, this study is well-designed for the nanobody development and identification of the binding interface. However, a few concerns are raised about the effects of Nb5 on Gβγ signaling and the therapeutic potential of Nb5.

Here are detailed comments.

1. The main concern is that the authors claimed that inhibiting Gβγ has a better therapeutic potential over GPCR-targeting drugs. However, would targeting Gβγ be more promiscuous? It would affect signal transduction of all GPCRs. Moreover, in the G protein signaling cycle, Gα activation is terminated when GTP is hydrolyzed to GDP not solely by re-interaction with Gβγ. Without forming a heterotrimer with Gβγ, the efficacy of GPCR-mediated GDP release from Gα is lowered, and GDP-bound Gα is less stable as a monomer than heterotrimer. Therefore, Nb5 can also inhibit Gα activity.

The authors appreciate the reviewer's questions. The reviewer is correct that targeting Gβγ will be relatively more promiscuous as compared to targeting individual GPCRs. However, targeted gene transduction/delivery methods will prove extremely beneficial in this case. Additionally, Nb5 has a large interaction surface area with Gβγ dimer ($>1030 \text{ \AA}^2$) and will serve as a template to achieve selectivity towards Gβ subtypes and even Gβγ combinations. The authors believe that targeting GPCR signaling through both the receptor and Gβγ will provide an effective therapeutic strategy.

The reviewer is correct about the possibility that Nb5 might indirectly affect Gα signaling due to the sequestration of Gβγ by Nb5 resulting in either a slow GPCR-mediated GDP release from Gα or decrease stability of Gα subunit. This study introduces Nb5 as a dynamic scavenger of Gβγ dimer that results in a partial inhibition of both Gβγ-mediated GIRK activation and the downstream AKT and ERK phosphorylation. The remaining Nb5-free Gβγ dimers may suffice to support the canonical G protein signaling that includes dissociation of Gβγ and the concomitant GDP-GTP exchange from Gα, and later termination by GTP hydrolysis and re-association of Gα-GDP with Gβγ.

However in response to the reviewer's comment, we performed experiments to monitor the effect of Nb5 on agonist-induced Gα signaling. Gα_q signaling was monitored in HEK293T/17 cells transfected with the m3 muscarinic acetylcholine receptor (M3R) and using a Ca²⁺ sensor (CalFluxVTN). Gα_s signaling was monitored in HEK293T/17 cells transfected with the dopamine D1 receptor (D1R) and using a cAMP sensor (Nluc-Epac-VV). The transfected cells were then simulated with acetylcholine or dopamine to activate M3R or D1R, respectively. Real-time monitoring of the resultant responses clearly showed no effect of Nb5 on the elevation of intracellular Ca²⁺ induced by M3R-Gα_q or on D1R-Gα_s-induced cAMP production. These results show that Nb5 does not alter Gα_q and Gα_s signaling in a living cell. These findings also are consistent with the higher affinity of Gα compared to Nb5 for the Gβγ dimer, supporting the inability of Nb5 to affect Gα signaling. However at higher concentrations, Nb5 might affect Gα signaling by increased sequestration of Gβγ as shown in Fig. 1d-e.

These findings have been added in the revised manuscript as Fig 6f-j.

Additionally for further clarification, the following text has been added to the discussion section of the revised manuscript.

Inserted text: “Additionally, Nb5 showed no significant effect on either the M3R-G α_q -induced intracellular Ca²⁺ elevation or D1R-G α_s -induced cAMP production. These results show that Nb5 does not alter the GTP-bound G α_q and G α_s -mediated signaling component in living cells. Overall, Nb5 serves as a dynamic scavenger of the G $\beta\gamma$ dimer and thereby causes partial inhibition of G $\beta\gamma$ -mediated signaling. The remaining Nb5-free G $\beta\gamma$ supports the canonical G α -GTP-mediated signaling that includes the GPCR-mediated GDP-GTP exchange from G α , and signaling termination by GTP hydrolysis and re-association of G α -GDP with G $\beta\gamma$. However at higher concentrations, Nb5 might affect the G α -GTP signaling component due to increased sequestration of G $\beta\gamma$ dimers.”

2. In page 6 first paragraph, it is written: “all 3 G $\beta_1\gamma_1$ -positive Nbs belonged to the same nanobody family with an identical complementarity determining region 3 (CDR3) and displayed similar biochemical properties.” Please provide appropriate data or references.

The authors appreciate and concur with the reviewer’s suggestion. The sequences of Nb5 family members and their binding analysis with G $\beta_1\gamma_1$ have been included as supplementary Fig. 1 in the revised supplementary material.

3. In page 7 line 99, please change Fig. 2b to Fig. 2a.

The correction has been made in the revised manuscript.

4. In Fig 4a, please provide the sequence of G β_5 .

Figure 4a has been revised to show the sequence of G β_5 .

5. In Fig 4b, what are the Nb5-coeluted proteins near 50 kDa? Are these proteins Nb5-binding non-G β proteins?

As noted by the reviewer, there are some co-eluted proteins with molecular weight in the range of 45-50 kDa. The authors performed in-gel protein digestion followed by mass-spectrophotometry analyses of the co-eluted proteins, and identified unique peptides from actin, tubulin and ATP synthase subunits alpha and beta. The authors believe that these co-eluted proteins are contaminants that were bound non-specifically to the Talon resin during purification. Nevertheless, to assess the possibility of interaction between these co-eluted proteins and Nb5, we used a computational approach that utilizes protein structure information for the prediction of possible protein-protein interactions. This approach has been used earlier for studying several viral-human proteome interactions⁶⁻⁸ as well as for non-viral pathogens causing tropical diseases⁹. This approach takes advantage of the shape complementarity between two proteins to predict a possible interaction with a common binding partner. In this study, we performed shape complementarity analyses between the G β subunit and the co-eluted proteins (actin, tubulin and ATP synthase subunits alpha and beta) using the Dali web server¹⁰. This server measures structural similarities by comparing the intramolecular distances between amino acids using the sum-of-pairs method. When distance matrices of two proteins share the same or similar features in approximately the same positions, they can be said to have similar folds with similar-length loops connecting their secondary structure elements^{11,12}. However, we did not find shape complementarity between G β subunit and any of the co-eluted proteins making the interaction between Nb5 and the co-eluted proteins unlikely.

6. In Fig 4e, please provide the expression levels of G β , GRK3, G α , Nb5, and Nb17.

The authors concur with the reviewer's suggestion. The expression levels of Gβ1-4, GRK3, Gα, Nb5, and Nb17 have been added in the revised manuscript as Fig. 4f.

7. In Fig 6b, when lane 2 is compared to lane 6, it suggests that Nb17 also inhibits ERK and AKT phosphorylation. Therefore, it is not correct to state "CHO-APJ-Nb17 cells showed no visible change in phosphorylation levels of either AKT or ERK proteins during the same time interval"

The authors thank the reviewer for raising this point. The apelin receptor mediated phosphorylation changes in PI3K-PKB/AKT and MAPK/ERK pathways were studied for both parental and nanobody co-expressing CHO-APJ cells. However, these experimental groups were not studied simultaneously, and therefore Nb17 co-expressed CHO-APJ cells seem to have reduced phosphorylation of AKT and ERK1/2 proteins compared to the parental CHO-APJ cells. This inconsistency could be explained by the difference in the confluency of cells (total number of cells loaded on the gels) and the difference in the exposure time of HRP-conjugated secondary antibodies between the experimental groups.

In response to the reviewer's comment, the effect of nanobodies on the phosphorylation of both AKT and ERK proteins was re-examined in 4 cell types simultaneously: parental CHO-APJ cells incubated with transfection reagents only, CHO-APJ cells transfected with pcDNA3.1 (+) empty vector, CHO-APJ-Nb17 cells and CHO-APJ-Nb5 cells, simultaneously. These results show that Nb17 has no significant effect on the phosphorylation of AKT and ERK proteins as compared to either parental CHO-APJ cells or CHO-APJ cells transfected with pcDNA3.1 (+) empty vector. However, a significant decrease in the phosphorylation of both AKT and ERK1/2 proteins was observed in CHO-APJ-Nb5 cells 5 min post-treatment with apelin.

These results have been incorporated in Fig. 6b of the revised manuscript. Additionally, the quantification of western blots has been included in Fig.6c of the revised manuscript.

8. It is well known that Gαi can also trigger ERK activation. Please provide experimental data or references whether apelin-induced ERK or AKT activation is mediated by Gαi or Gβγ.

A body of evidence demonstrates that Gβγ subunits play significant roles in regulating downstream G protein signaling including the MAP kinase pathway. Past studies have shown that Gβγ antagonist βARK1-(495-689) inhibits Gα_i-mediated ERK activation in a Gβγ dependent manner^{13,14}. Extensive studies on small molecule- and peptide-based Gβγ inhibitors show that Gi-coupled GPCR-mediated ERK signaling is altered by M119 class and SIRK Gβγ inhibitors^{1,3}. Our data also suggest that Nb5 inhibits Gα_i-linked ERK and PI3 kinase pathways, at least in part through inhibiting Gβγ. Overall, these studies suggest that Gβγ-mediated signaling pathways may be as important as those regulated by Gα subunits¹⁵⁻¹⁷.

To address the reviewer's comment, we have added this summary into the Discussion section of the revised manuscript.

9. In Fig 6d, please provide control data that does not express Nb5 or Nb7.

In response to the reviewer's comment, Fig. 6e has been updated to include the control data of parental CHO-APJ cells without expressed nanobodies.

10. In the text, the Results section ends without explaining Fig 6c and 6d, and the explanation for these figures are in the Discussion section. Please re-arrange the text and provide the explanations of Fig 6c and Fig 6d in the Result section.

The authors concur with the reviewer. The results explaining Figure 6c and 6d have been moved to the results section in the revised manuscript.

11. Based on the references that the authors provided, G $\beta\gamma$ stimulates adenylyl cyclase. It would be easy for the readers if the authors directly state that G $\beta\gamma$ stimulates adenylyl cyclase rather than “influence the activity of several AC”.

The authors appreciate the reviewer’s suggestion. For further clarification, the statement has been replaced with “G $\beta\gamma$ can stimulate several AC isoforms” in the revised manuscript.

12. In page 12, “But Nb5 had no effect on the G α -GTP-mediated cAMP signaling component as evidenced by its maintenance after the loss of the G $\beta\gamma$ -mediated cAMP signaling component in cells transfected with Nb5 (Fig. 6d, greencyan).” However, Fig 1d demonstrates that pre-incubation of G t with Nb5 decreases the initial G t activation rate. Would these two results show inconsistency? Moreover, I am not convinced that Fig 6d demonstrate that Nb5 inhibits the AC stimulatory effect of G $\beta\gamma$ and not affect G α_i . The authors stated in the Abstract “G $\beta\gamma$ also serves as a negative regulator of G α that is essential for G protein inactivation”. If Nb5 prevents G $\beta\gamma$ from binding to G α , would this increase the G α activity?

The reviewer makes a very good point that pre-incubation of G t with Nb5 decreases the initial G t activation rate which might be inconsistent with the above statement. Notably, the G t activation assays were performed using 2 molar excess of Nb5 relative to G t concentration. Even though G α has a higher affinity towards G $\beta\gamma$ dimer as compared to Nb5, at higher concentrations Nb5 significantly reduces the G t activation rates of photoactivated rhodopsin as shown in Fig1d-e. However, in a cell environment Nb5 is present at relatively lower concentration, and therefore acts as a dynamic scavenger of already released G $\beta\gamma$ resulting in the partial inhibition of the G $\beta\gamma$ signaling. This is also demonstrated by the partial inhibition of both G $\beta\gamma$ -mediated GIRK activation and the downstream AKT and ERK phosphorylation. The remaining Nb5-free G $\beta\gamma$ dimers might suffice to support the canonical G protein signaling that includes the GPCR-mediated GDP-GTP exchange from G α , and later termination by GTP hydrolysis and re-association of G α -GDP with G $\beta\gamma$.

Additionally, we performed experiments to monitor the effect of Nb5 on agonist-induced G α_s and G α_q signaling events. Real-time monitoring of M3R-G α_q mediated Ca²⁺ elevation and D1R-G α_s mediated cAMP production clearly demonstrated no effect of Nb5 on these signaling events. These results show that Nb5 does not alter either G α_q or G α_s signaling in a living cell .

These findings have been added in the revised manuscript as Fig 6f-j.

13. With protein identification by mass spectrometry-based proteomics in Fig 4, are there proteins identified other than G β proteins? Please provide detailed database search parameters in the Methods and a full list of identified proteins.

There were additional proteins detected other than G β subtypes. However, their sequest score which determines the quality of hits based on the number of ions in the MS/MS spectrum that match with the experimental data was considerably lower than G β_1 , G β_2 and G β_4 subtypes. Nevertheless, unique peptides from those proteins were detected during sequest analysis which points towards their presence in the sample gel band as possible contaminants bound non-specifically to the Talon resin. Some of these contaminants are SNARE proteins and dehydrogenases that are expressed in neurons and do not have shape complementarity to G $\beta\gamma$ dimer.

In response to reviewer's suggestions, database search parameters have been added in the methods section of the revised manuscript. Additionally, a full list of identified proteins has been added as Supplementary Table 3 of the revised supplementary information.

14. In page 29, "Note the decrease in Gy1 Coomassie staining in the presence of 300 mM imidazole during protein elution." Please explain why the Gy1 Coomassie staining is decreased in the presence of 300 mM imidazole.

The authors appreciate the reviewer's insightful query. The only difference between Rh+GTP+Nb5, GTP+Nb5, Nb5 only conditions and Nb7 only and control conditions is the presence of 300 mM imidazole used during immobilized-Ni²⁺ affinity chromatography purification. Removal of excess imidazole with size exclusion chromatography or other similar techniques does not resolve the Gy smearing and Coomassie staining artifact on SDS-polyacrylamide gels. The crystal structure of the G $\beta_1\gamma_1$ -Nb5 complex obtained from similar preparations (Fig. 1b) also negates the possibility of Gy₁ dissociation from the G $\beta_1\gamma_1$ complex in the presence of imidazole. The authors speculate that the observed smearing of Gy₁ is due to the salt/charge effect caused by imidazole binding, and consequently affecting the migration profile of Gy₁ on SDS-polyacrylamide gels. This SDS-polyacrylamide artifact has been previously observed with several post-translationally modified proteins in our laboratory and by others in the field.

15. Please provide PDB ID for Suppl Fig 1c.

The PDB ID has been added in Fig 2c of the revised Supplementary Material.

Reviewer #3 (Remarks to the Author):

In these studies a llama-derived nanobody (Nb5) was identified that tightly binds to the G $\beta\gamma$ dimer. Differential hydrogen/deuterium exchange and crystallography studies suggest that Nb5 binds a common binding site for the G $\beta\gamma$ dimer.

Interestingly, Nb5 has affinity for combinations of β - and γ -subtypes, allowing a broad applicability. Nb5 impairs the activation of both the G $\beta\gamma$ -regulated G-protein-gated potassium channels and the G $\beta\gamma$ -mediated PI3K and MAPK pathways. The authors claim that Nb5 has an inhibitory effect on G $\beta\gamma$ -mediated signaling, and no effect on G α -GTP-mediated signaling, therefore serving as a versatile tool to achieve selective modulation of GPCR-signaling.

The nanobody described in these studies, Nb5, is an interesting tool, that may serve to further characterize the role of G $\beta\gamma$ s in GPCR mediated signaling. Complementary approaches are used to characterize binding of this nanobody to G $\beta\gamma$ s. However, the claims made with respect to the lack of Nb5 effects on Galpha mediated signaling need controls and experiments to further substantiate this claim.

Major points:

1. Nb5 is suggested to only affect G $\beta\gamma$ signaling and leave Galpha signaling intact. The authors state that since the apparent affinity of Nb5 is lower than the affinity of the Galpha unit to the G $\beta\gamma$ dimer that it cannot affect Galpha signaling (line 244). This statement is not true. At high concentrations of Nb5, Nb5 may still inhibit Galpha signaling. Overexpression of the Nb inside the cell could therefore affect Galpha signaling.

The reviewer raises a good point that at high concentrations Nb5 might inhibit G α signaling by increased G $\beta\gamma$ sequestration as shown in Fig. 1d-e. However, in a cellular environment Nb5 is present at relatively low concentration, and therefore acts as a partial dynamic scavenger of already released G $\beta\gamma$ resulting in the partial inhibition of the G $\beta\gamma$ signaling. This is also demonstrated by the partial inhibition of both G $\beta\gamma$ -mediated GIRK activation and the downstream AKT and ERK phosphorylation. The remaining Nb5-free G $\beta\gamma$ dimers may suffice to support the canonical G protein signaling that includes dissociation of G $\beta\gamma$ and the concomitant GDP-GTP exchange from G α , and later termination by GTP hydrolysis and re-association of G α -GDP with G $\beta\gamma$.

However in response to the reviewer's comment, the authors have conducted additional experiments to monitor the effect of Nb5 on agonist-induced G α signaling. G α_q signaling was monitored in HEK293T/17 cells transfected with the m3 muscarinic acetylcholine receptor (M3R) and using a Ca²⁺ sensor (CalFluxVTN). G α_s signaling was monitored in HEK293T/17 cells transfected with the dopamine D1 receptor (D1R) and using a cAMP sensor (Nluc-Epac-VV). The transfected cells were then stimulated with acetylcholine or dopamine to activate M3R or D1R, respectively. Real-time monitoring of the resultant responses clearly showed no effect of Nb5 on the elevation of intracellular Ca²⁺ induced by M3R-G α_q or on D1R-G α_s -induced cAMP production. These results show that Nb5 does not alter either G α_q or G α_s signaling in a living cell. These findings also are consistent with the higher affinity of G α compared to Nb5 for the G $\beta\gamma$ dimer, supporting the inability of Nb5 to affect G α signaling. However as mentioned by the reviewer, higher concentrations or overexpression of Nb5 might affect G α signaling due to an increased sequestration of G $\beta\gamma$.

These findings have been added in the revised manuscript as Fig 6f-j. Additionally in response to the reviewer's comment and for further clarification, the following text has been added to the results section of the revised manuscript.

Inserted text: "Additionally, Nb5 showed no significant effect on either the M3R-G α_q -induced intracellular Ca²⁺ elevation or D1R-G α_s -induced cAMP production. These results show that Nb5 does not alter the GTP-bound G α_q and G α_s -mediated signaling component in living cells. Overall, Nb5 serves as a dynamic scavenger of the G $\beta\gamma$ dimer and thereby causes partial inhibition of G $\beta\gamma$ -mediated signaling. The remaining Nb5-free G $\beta\gamma$ supports the canonical G α -GTP-mediated signaling that includes the GPCR-mediated GDP-GTP exchange from G α , and signaling termination by GTP hydrolysis and re-association of G α -GDP with G $\beta\gamma$. However at higher concentrations, Nb5 might affect the G α -GTP signaling component due to increased sequestration of G $\beta\gamma$ dimers."

2. To show that Nb5 has no effect on G α mediated signaling include an experiment (G α dependent) showing that with increasing concentrations of Nb5 no effect on G α signaling is apparent.

The authors appreciate the reviewer's comment. However, due to a non-linear relationship between the transfection efficiency and concentration of DNA, it's technically challenging to quantify and control the expression of Nb5 in cells. Hence, we have conducted additional experiments to further confirm the G α -independent nature of Nb5. Real-time monitoring of G α_q and G α_s signaling clearly showed no effect of Nb5 on M3R-G α_q -induced intracellular Ca²⁺ elevation and on D1R-G α_s -induced cAMP production. These results suggest that Nb5 does not alter either G α_q or G α_s signaling in a living cell.

3. In figure 5 the authors showed the effects of Nb5 on D2 and M4 mediated GIRK signaling. The authors should include control experiments showing signaling in the absence of Nb5 and Nb17 too. Overexpression of proteins may affect the protein machinery of cells and indirectly affect signaling.

Figure 5 has been revised to include D2R controls showing the GIRK currents in the absence of nanobodies. Additionally, the following text has been added to the results section of the revised manuscript.

Inserted text: “Similarly, there was no effect on the D2R-IPSCs in postsynaptic GIRK2 positive iMSNs in the absence of nanobody treatment (271.50 ± 73.95 pA in 1 min; 257.20 ± 50.23 pA after 15 min, $n = 7$, $P > 0.05$, Student’s *t*-test) (Fig. 5d,h,i).”

4. The authors should include control experiments in figure 6D showing cAMP signaling in the absence of Nb5 and Nb17 too. Is cAMP signaling in the absence of Nbs similar as co-expression with Nb17 only?

In response to the reviewer’s comment, Fig. 6d (which is Fig. 6e in the revised manuscript) has been updated to include the control data of parental CHO-APJ cells without expressed nanobodies. The comparison of parental CHO-APJ cells with CHO-APJ cells co-expressing Nb17 showed no significance difference in the inhibition of forskolin-stimulated accumulation of cAMP.

5. In figure 6 the authors showed that the effects of Nb5 on apelin-mediated phosphorylation of AKT and ERK in CHO cells. Both Nb5 but also Nb17 appear to impair phosphorylation of AKT and ERK, when compared to cells not expressing Nb5 nor Nb17. Include quantification of blots. How does one explain the decrease of apelin-induced phosphorylation of Akt and ERK in Nb17 expressing cells?

The authors thank the reviewer for raising this point. The apelin receptor mediated phosphorylation changes in PI3K-PKB/AKT and MAPK/ERK pathways were studied for both parental and nanobody co-expressing CHO-APJ cells. However, these experimental groups were not studied simultaneously, and therefore Nb17 co-expressed CHO-APJ cells seem to have reduced phosphorylation of AKT and ERK1/2 proteins compared to the parental CHO-APJ cells. This inconsistency could be explained by the difference in the confluency of cells (total number of cells loaded on the gels) and the difference in the exposure time of HRP-conjugated secondary antibodies between the experimental groups.

In response to the reviewer’s comment, the effect of nanobodies on the phosphorylation of both AKT and ERK proteins was re-examined in 4 cell types simultaneously: parental CHO-APJ cells incubated with transfection reagents only, CHO-APJ cells transfected with pcDNA3.1 (+) empty vector, CHO-APJ-Nb17 cells and CHO-APJ-Nb5 cells. These results show that Nb17 has no significant effect on the phosphorylation of AKT and ERK proteins as compared to either parental CHO-APJ cells or CHO-APJ cells transfected with pcDNA3.1 (+) empty vector. However, a significant decrease in the phosphorylation of both AKT and ERK1/2 proteins was observed in CHO-APJ-Nb5 cells 5 min post-treatment with apelin.

These results have been incorporated in Fig. 6b of the revised manuscript. Additionally, the quantification of western blots has been included in Fig.6 of the revised manuscript.

6. Does expression of Nb5 and Nb17 affect expression of D2/M4 or apelin receptors?

The effect of Nb5 or Nb17 on the D2 or M4 receptor-mediated GIRK channel activation was tested with whole-cell patch clamping of striatum neurons. The nanobodies were delivered as an internal solution in the neuronal exons, and therefore should not affect the expression of D2/M4 receptors. However, CHO-APJ cells used to study the phosphorylation of AKT and ERK1/2 proteins and cAMP assays were transfected with nanobodies. Hence, we quantified the expression levels of the apelin receptor in parental CHO-APJ cells, CHO-APJ cells transfected with pcDNA3.1 (+) empty vector, CHO-APJ-Nb17, and CHO-APJ-Nb5 cells. As expected, no significant differences were observed in the expression levels of the apelin receptor between these experimental groups.

These results have been included in Supplementary fig. 3b of the revised manuscript.

Minor comments:

7. Include the results of figure 6d into the results section.

The authors concur with the reviewer. The results explaining Figure 6c and 6d have been moved to the results section in the revised manuscript.

8. No referral has been made to recent studies (Ghosh et al 2017) using nanobodies targeting barrestin in the introduction. These studies are also an example of the use of nanobodies targeting GPCR signaling rather than receptors.

In response to the reviewer's comment, the suggested reference has been incorporated into the Introduction of the revised manuscript.

Modified text: "With the exception of β -arrestin specific antibody fragments¹⁸, all small-molecule-, antibody- and nanobody-based approaches developed to date target GPCR-mediated signaling at the GPCR level^{19,20}, and thereby are GPCR-specific and cannot be used universally."

Reviewer #4 (Remarks to the Author):

The manuscript entitled "Targeting G protein-coupled receptor signaling at the G protein level with a selective nanobody inhibitor" was an extremely well written article and contained some very important and novel findings. The authors show convincing data that they have selected a nanobody (Nb5) that targets the G β gamma dimer specifically. This nanobody was able to inhibit G β gamma subunits 1-4 but not 5 and also inhibit the regulation of GIRK channels. The use of SPR and BRET convincing supports the conclusions of the authors. Unlike the nanobodies that are progressing through the clinic Nb5 has to operate inside the cell. This makes the therapeutic role for this nanobody more complicated, however the last part of the discussion provides a clear-eyed summary of the difficulties here.

I have only a few minor comments;

1. It would have been nice to see the sequences of the nanobodies representing the 14 families perhaps as a supplementary figure (or at least the sequence of the Nb5).

The authors appreciate and concur with the reviewer's suggestion. The sequences of Nb5 family members have been included as Fig. S1a in the revised supplementary material.

2. The authors state on page 4 that "recognition. However, only nanobodies (Nbs) derived from the variable region of camelid 106 heavy chain are endowed with favorable characteristics in terms of size, solubility, affinity and 107 ease of production" The use of the word only here is rather definitive and I believe that the single domain IgNARs and the recently reported i-body (Griffiths et al J. Biol. Chem. 2016, 291, 12641–12657) could claim to have similar properties

The authors concur with the reviewer that vNARs and i-bodies have similar biochemical and biophysical characteristics as nanobodies. In response to the reviewer's suggestion, the word "only" has been removed and the following text has been added to the Introduction of the revised manuscript.

Inserted text: “vNARs^{21,22}, the variable domain of shark-derived new antigen receptors and their human equivalents, i-bodies²³ derived from the I-set immunoglobulin superfamily are other examples of single-domain antibody-like molecules with biochemical and biophysical properties similar to Nbs. Recently, i-bodies that selectively block CXCR4 β -arrestin recruitment were reported²³.”

Reviewer #5 (Remarks to the Author):

The report by Gulati et.al. entitled “Targeting G protein coupled receptor signaling at the G protein level with a selective inhibitor” details the development of a nanobody (Nb5) that can interact with G protein beta-gamma subunits. This shifts the equilibrium of the G protein heterotrimer to be disassociated and selectively inhibits beta-gamma mediated cell signaling. As beta-gamma mediated cell signaling is an important communication pathway in the cell. This signal biasing tool might be useful to understand and develop new models for cell signaling. However, there are a number of issues with this report.

Major concerns:

1. The authors do not actually use this tool to address any biological questions on signaling in the context of this report, so they have not proven that this nanobody will be useful to the field.

This study serves as the first comprehensive structure-function study communicating the ability of a nanobody to modulate a GPCR-mediated G $\beta\gamma$ -signaling event in any cell type. Interestingly, Nb5 acts as a specific inhibitor of various G $\beta\gamma$ -mediated signaling events that regulate several ‘undruggable’ targets such as GIRK channels, ERK, and AKT kinases. This opens new avenues for nanobody-assisted modulation of cellular signaling to treat various excitatory neurological conditions and cancer progression. Although these results demonstrate a novel therapeutic application of nanobodies to inhibit intracellular events mediated by protein-protein interactions, additional scientific endeavors are required to make Nb5 selective towards different G β subtypes for immediate translational purposes. In addition to the applications of Nb5 in GPCR signaling modulation, this study also describes nanobodies as versatile protein purification reagents as evidenced by the ability of Nb5 to selectively pull out G $\beta\gamma$ dimers from crude rod outer segment and mouse brain extracts.

To address the reviewer’s comment, we have added this summary in the Discussion section to further highlight the applications of our study.

2. There are already G $\beta\gamma$ specific inhibitors reported in the literature such as Gallein, M119, M119K. This decreases the novelty of the nanobody reported here. A comparison of Nb5 and these known inhibitors would be important to show whether Nb5 has advantages.

The authors appreciate the reviewer’s comment. We agree it is important to include a comparison of the known G $\beta\gamma$ inhibitors like M119 family members with Nb5 for G $\beta\gamma$ binding. In response to the reviewer’s comment, the following text has been added to the discussion section of the revised manuscript.

Inserted text: “The M119 class of small-molecule inhibitors, which includes M119, gallein and M119K, are potent G $\beta\gamma$ antagonists. They remain the most extensively validated inhibitors of G $\beta\gamma$ signaling. Studies to date indicate that these molecules bind to a G $\beta\gamma$ “hot spot” on the top surface of the β subunit with apparent affinities in the high nM to low μ M range¹. The M119 class of inhibitors displays a limited number of chemical moieties that

interact with the Gβγ dimer. Therefore, structure activity relationship studies to achieve Gβ selectivity would have limited practicality. Nb5, on the other hand, binds Gβγ with a low nM affinity and has a larger area of interaction with the Gβγ dimer (>1030 Å²). These features indicate that Nb5 might serve as a better template for structure activity relationship studies to achieve Gβ selectivity. Unlike nanobodies, the M119 class of inhibitors can be delivered orally and intraperitoneally to modulate Gβγ –dependent pathways, which admittedly takes them one step closer to clinical applications. Challenges remain for the development of nanobody-based clinical therapies due to the dependence of this approach on efficient gene transduction methods. The C-terminal domain of GRK2 (βARK-ct), PDC and several affinity matured peptides, including the SIRK peptide family, comprise an alternative class of potent Gβγ inhibitors that are mechanistically similar to Nb5²⁻⁴. While βARK-ct and PDC have a large area of interaction with the Gβγ dimer (>1000 Å²), no atomic level information is available for SIRK or other peptides except that they bind to the same Gβγ “hot spot” with high nM affinities^{4,5} and selectively inhibit the binding of Gβγ effector molecules. Several cell-permeating versions of SIRK peptides have been shown to effectively modulate ERK1/2 and MAPK signaling in arterial smooth muscle cells³. Treatment strategies involving Nb5, βARK-ct, PDC and members of the SIRK peptide family would all require gene transduction methods, however this may be worthy of significant effort given their potential to treat various Gβγ-related disorders.”

Concerns with data interpretation:

3. The authors suggest that there is a therapeutic potential of this inhibitor. However, they also state that “Nb5 responds to all combinations of β- and γ-subtypes, indicating its broad applicability to various cell types”. This may not be consistent with therapeutic applications.

The authors appreciate the reviewer’s comment. Nb5 binding to all combinations of Gβ- and Gγ-subtypes is both a boon and a bane. Responding to all combinations of β- and γ-subtypes increases the applications of Nb5 in various cell types, but also increases the possibility of potential side-effects. However, the ease of production and genetic manipulation of nanobodies opens several avenues to increase the selectivity of Nb5 towards specific Gβ subtypes and even Gβγ combinations. The comprehensive biochemical and structural analyses of Nb5 reported here provide a basis for further biomedical undertakings to make nanobodies a new class of therapeutics.

To address the reviewer’s comment, we have added the following text in the Discussion section to further highlight this summary.

Inserted text 1: “The ease of production and genetic manipulation of nanobodies are advantageous characteristics for achieving the goal of engineering variants with specificity toward different Gβ subtypes and even Gβγ combinations. This opens up the possibility of designing targeted treatments for various excitatory neurological conditions and inhibitors for blocking cancer progression.”

”

Inserted text 2: “The comprehensive biochemical and structural analyses of Nb5 reported in this study provide a basis for further research studies to turn nanobodies into a viable class of therapeutics.”

4. This study shows a direct interaction between Nb5 and the G-beta 1 subtype (by using X-ray crystallography, HDX and SPR). The interaction between Gb2-4 and Nb5 is shown indirectly and does not test all of the Gbg combinations. They are assuming Nb5 will interact and inhibit all Gbg mediated cell signaling. This needs to be clarified.

Nb5 is expected to bind all combinations of Gβ- and Gγ-subtypes, and therefore is expected to suppress Gβγ signaling which is mediated through the Gβγ hotspot. Interestingly, Nb5

does not completely ablate G $\beta\gamma$ signaling as demonstrated by the GIRK activation and downstream AKT and ERK phosphorylation assays reported in the manuscript. This highlights the capability of Nb5 to suppress dysregulated G $\beta\gamma$ signaling pathways while maintaining a considerable amount of free G $\beta\gamma$ available to carry out downstream signaling.

The authors appreciate the reviewer's comment, and for further clarity the following text has been added to the Results of the revised manuscript.

Inserted text: "Overall, these experiments suggest that Nb5 binds to all combinations of G β subtypes 1-4 and G γ and suppresses G $\beta\gamma$ signaling mediated through protein-protein interactions near the G $\beta\gamma$ hotspot. These results are suggestive of Nb5's potential broad utility to influence G $\beta\gamma$ signaling in various cell types."

5. Because of Nb5s exhibits some nonspecific behavior, the authors expect broad applicability to various cell types and cellular signaling. If Nb5 affects various cell types and signaling pathways, it will also increase potential side effects of Nb5 as a therapeutic usage. Until they develop Gbg subtype specific nanobody, the idea of therapeutic potential of nanobody on Gbg signaling may not be feasible.

The authors concur with the reviewer's point about the potential side-effects of Nb5 due to its ability to bind all combinations of G β - and G γ -subtypes. However, the ability of Nb5 to suppress but not completely ablate G $\beta\gamma$ signaling makes it a valuable tool for future nanobody-based therapeutics to modulate cellular signaling. As described in the current manuscript, Nb5 acts as proof of principle and serves as one of the first examples wherein a nanobody modulates a GPCR-mediated G $\beta\gamma$ -signaling event in any cell type. Future experiments will focus on increasing the G β selectivity of Nb5 by using both structure guided mutational analyses and G β selective phage display libraries. To address the reviewer's comment, the following text has been added to the Discussion section.

Inserted text: "The ability of Nb5 to suppress but not completely ablate G $\beta\gamma$ signaling makes it a beneficial tool for future nanobody-based therapeutics to modulate cellular signaling. This work on Nb5 provides a proof of principle and serves as one of the first examples wherein a nanobody modulates a GPCR-mediated G $\beta\gamma$ -signaling event in any cell type. Future experiments will focus on increasing the G β selectivity of Nb5 by using both structure guided mutational studies and development of G β selective phage display libraries."

6. In figure 1a, why isn't the Ggamma subunit in the Rh+GTP+Nb5, GTP+Nb5, Nb5 conditions as Nb7 and control?

The authors appreciate the reviewer's insightful query. The only difference between Rh+GTP+Nb5, GTP+Nb5, Nb5 only conditions and Nb7 only and control conditions is the presence of 300 mM imidazole used during immobilized-Ni²⁺ affinity chromatography purification. Removal of excess imidazole with size exclusion chromatography or other similar techniques does not resolve the G γ smearing and Coomassie staining artifact on SDS-polyacrylamide gels. The crystal structure of the G $\beta_1\gamma_1$ -Nb5 complex obtained from similar preparations (Fig. 1b) also negates the possibility of G γ_1 dissociation from the G $\beta_1\gamma_1$ complex in the presence of imidazole. The authors speculate that the observed smearing of G γ_1 is due to the salt/charge effect caused by imidazole binding, and consequently affecting the migration profile of G γ_1 on SDS-polyacrylamide gels. This SDS-polyacrylamide artifact has been previously observed with several post-translationally modified proteins in our laboratory and by others in the field.

7. In figure 1, the G alpha t nucleotide exchange looks extremely slow compare to Gai or Gas. In the presence of Gbg, nucleotide exchange should be slower than with just the alpha subunit. In figure 1d, it looks like nucleotide exchange is reaching more than 65-75% of the

maximum level less than 3 mins in the dark and in the presence of Nb5 (which is practically the rate of Galphat in this study). Why the exchange rate is so fast in the dark and in the presence of Nb5? It would be nice to show the raw data without normalization.

The authors appreciate the reviewer's insightful query. Conditions for the G_t activation assay were chosen such that the G_t activation rate was the same as that determined by GTP γ S-induced complex dissociation. The G_t activation rates were calculated based on the exponential slopes of the representative spectra^{24,25}. Based on these analyses the initial G_t activation rates of Rh^* ($k_{initial}$) in the presence of Nb5 and in the dark are $0.0060 \pm 0.00011 \text{ s}^{-1}$ and $0.001 \pm 0.00010 \text{ s}^{-1}$, respectively. These rates are 16.9 % and 2.8 % of the initial G_t activation rate of Rh^* ($k_{initial} = 0.0353 \pm 0.00485 \text{ s}^{-1}$). The initial G_t activation rates of these samples have been quantified in Fig. 1e for comparison. Additionally, Fig 1d shows the representative raw data for the G_t activation assays.

8. The BRET signal in the presence of Nb5 statistically decreases compared to Nb7 in Figure 4. However, it is still significantly higher than in the presence of the Galpha subunit, which indicates that Gbg can still interact with GRK in the presence of Nb5. This result contradicts with Figure 5, which is showing at least 50% GIRK channel inhibition with Nb5. This discrepancy needs to be clarified.

The reviewer makes the correct assessment that $G\beta\gamma$ can still interact with GRK in the presence of Nb5 in the BRET assays which support our conclusion that Nb5 does not completely ablate $G\beta\gamma$ signaling. The reviewer also notes that the effects observed in the BRET assays are less pronounced than the GIRK channel inhibition assays. This variable effect is likely due to the differences in the mode of delivery and the effective concentration of Nb5 in cells. While, Nb5 was co-transfected in human embryonic kidney cells 293 in BRET assays, GIRK recordings were performed by introducing an internal solution containing 10 μM of Nb5 in the neuronal exons."

For further clarity the following text has been added in the Results section of the revised manuscript.

Inserted text: "The effect of Nb5 in these GIRK inhibition assays was more demonstrable compared to its effects in the BRET assays. This likely is due to differences in the mode of delivery and thereby the effective concentration of Nb5 attained in cells in the two assay systems. While, Nb5 was co-transfected in human embryonic kidney cells 293 in BRET assays, GIRK recordings were performed by introducing an internal solution containing 10 μM of Nb5 directly to the neuronal exons."

9. The effect of Nb5 on the cellular pathways were investigated in figure 6. Compared to control (without any NB) experiments, ERK and AKT phosphorylation significantly decreases not just in the presence of Nb5 but also with Nb7 (figure 6b). This result is in opposition to the rest of the data, which are consistent with Nb17 not showing any inhibitory effects. Are there any non-specific interactions between the Nb17 and the other signaling proteins in the cell? If so, this then raises a question about if there are any non-specific interaction between Nb5 and the other proteins in the cell.

The authors thank the reviewer for raising this point. The apelin receptor mediated phosphorylation changes in PI3K-PKB/AKT and MAPK/ERK pathways were studied for both parental and nanobody co-expressing CHO-APJ cells. However, these experimental groups were not studied simultaneously, and therefore Nb17 co-expressed CHO-APJ cells seem to have reduced phosphorylation of AKT and ERK1/2 proteins compared to the parental CHO-APJ cells. This inconsistency is not due to non-specific interactions between Nb17 with other cellular signaling proteins and could be explained by the difference in the confluency of cells (total number of cells loaded on the gels) and the difference in the exposure time of HRP-conjugated secondary antibodies between the experimental groups.

In response to the reviewer's comment, the effect of nanobodies on the phosphorylation of both AKT and ERK proteins was re-examined in 4 cell types simultaneously: parental CHO-APJ cells incubated with transfection reagents only, CHO-APJ cells transfected with pcDNA3.1 (+) empty vector, CHO-APJ-Nb17 cells and CHO-APJ-Nb5 cells. These results show that Nb17 has no significant effect on the phosphorylation of AKT and ERK proteins as compared to either parental CHO-APJ cells or CHO-APJ cells transfected with pcDNA3.1 (+) empty vector. However, a significant decrease in the phosphorylation of both AKT and ERK1/2 proteins was observed in CHO-APJ-Nb5 cells 5 min post-treatment with apelin.

These results have been incorporated in Fig. 6b of the revised manuscript. Additionally, the quantification of western blots has been included in Fig.6 of the revised manuscript.

10. Recognizing the ongoing debate in the field on how to define resolution of crystal structures, it could be worth reporting in the legend to supplementary table 2 the resolution of data where more traditional data collection statistics are observed (for example $I/\sigma > 2$ at a resolution of "x") as well as describe the rationale for cutting the data at the proposed resolution. By giving a statement like this, it helps readers to compare the accuracy of the model to more historic standards.

The authors concur with the reviewer. In response to the reviewer's suggestion, we have revised the Table 2 legend to include the resolution cut-off at $<I/\sigma>$ of 2.1 by adding the following text in the revised supplementary material. The rationale for data truncation at the proposed resolution has also been included in the revised legend.

Inserted text: "#Resolution bin at $<I/\sigma>$ of 2.1 is 2.70-2.62 Å for comparison with the historical standards of x-ray data truncation. The resolution bin with $<I/\sigma>$ of 1.05 is used for the resolution cut-off to include the intensities that are significantly above the noise level. Extending the data beyond $<I/\sigma>$ values of >2 have been shown to improve structure determination in many cases with no negative impact on model building²⁶⁻²⁸."

Reviewer #6 (Remarks to the Author):

In this manuscript, Gulati et al. identify a nanobody (Nb5) that binds Gbeta/gamma dimers with high affinity to prevent the interaction of dimers to Galpha-t and downstream effectors. The data strongly support these conclusions and the manuscript is very well written. On the other hand, discussions regarding the significance of the work with respect to previous studies seems unnecessarily narrow. That is, the hotspot on Gbeta/gamma dimers that is bound by Nb5 is also used by affinity-matured peptides and small molecules to modulate signaling mediated by Gbeta/gamma dimers (1,2). A detailed discussion of the merits of the various methods to target Gbeta/gamma dimers seems warranted. This discussion should also include the use of the C-terminal domain of GRK2 (bARK-ct). This genetically-encoded domain effectively sequesters Gbeta/gamma dimers in much the same fashion as Nb5 and has been used for approximately two decades to silence signaling by Gbeta/gamma dimers. What preferential uses do the authors envision for Nb5 relative to bARK-ct?

1. Bonacci TM, Mathews JL, Yuan C, Lehmann DM, Malik S, Wu D, Font JL, Bidlack JM and Smrcka AV (2006). Differential targeting of Gbeta/gamma-subunit signaling with small molecules. *Science* 312, 443-446.
2. Goubaeva F, Ghosh M, Malik S, Yang J, Hinkle PM, Griendling KK, Neubig RR and Smrcka AV (2003). Stimulation of cellular signaling and G protein subunit dissociation by G protein beta/gamma subunit-binding peptides. *J Biol Chem* 278, 19634-19641.

The authors appreciate the reviewer's comment. We agree it is important to include a comparison of the known G $\beta\gamma$ small molecule and peptide inhibitors with Nb5 for G $\beta\gamma$ binding. In response to the reviewer's comment, the following text has been added to the discussion section of the revised manuscript.

Inserted text: "The M119 class of small-molecule inhibitors, which includes M119, gallein and M119K, are potent G $\beta\gamma$ antagonists. They remain the most extensively validated inhibitors of G $\beta\gamma$ signaling. Studies to date indicate that these molecules bind to a G $\beta\gamma$ "hot spot" on the top surface of the β subunit with apparent affinities in the high nM to low μ M range ¹. The M119 class of inhibitors displays a limited number of chemical moieties that interact with the G $\beta\gamma$ dimer. Therefore, structure activity relationship studies to achieve G β selectivity would have limited practicality. Nb5, on the other hand, binds G $\beta\gamma$ with a low nM affinity and has a larger area of interaction with the G $\beta\gamma$ dimer (>1030 Å²). These features indicate that Nb5 might serve as a better template for structure activity relationship studies to achieve G β selectivity. Unlike nanobodies, the M119 class of inhibitors can be delivered orally and intraperitoneally to modulate G $\beta\gamma$ –dependent pathways, which admittedly takes them one step closer to clinical applications. Challenges remain for the development of nanobody-based clinical therapies due to the dependence of this approach on efficient gene transduction methods. The C-terminal domain of GRK2 (β ARK-ct), PDC and several affinity matured peptides, including the SIRK peptide family, comprise an alternative class of potent G $\beta\gamma$ inhibitors that are mechanistically similar to Nb5 ²⁻⁴. While β ARK-ct and PDC have a large area of interaction with the G $\beta\gamma$ dimer (>1000 Å²), no atomic level information is available for SIRK or other peptides except that they bind to the same G $\beta\gamma$ "hot spot" with high nM affinities ^{4,5} and selectively inhibit the binding of G $\beta\gamma$ effector molecules. Several cell-permeating versions of SIRK peptides have been shown to effectively modulate ERK1/2 and MAPK signaling in arterial smooth muscle cells ³. Treatment strategies involving Nb5, β ARK-ct, PDC and members of the SIRK peptide family would all require gene transduction methods, however this may be worthy of significant effort given their potential to treat various G $\beta\gamma$ -related disorders. "

Minor considerations:

1. In the abstract, the authors state that "Nb5 has no effect on G α -GTP signaling events..." This statement does not seem to be supported by the data in the sense that sequestration of G $\beta\gamma$ should prevent GPCR-mediated activation of G α subunits. Indeed, this property is presumably the cause of the depression in rhodopsin-catalyzed activation of G α -t (Fig. 1d-e). Please clarify.

The reviewer raises a good point that the sequestration of G $\beta\gamma$ should prevent GPCR-mediated activation of G α subunits. Indeed at higher concentrations, Nb5 significantly reduces the G $\beta\gamma$ activation rates of photoactivated rhodopsin as shown in Fig1d-e. However, in a cell environment Nb5 is present at relatively lower concentration, and therefore acts as a scavenger of already released G $\beta\gamma$ resulting in the partial inhibition of the G $\beta\gamma$ signaling. This is also demonstrated by the partial inhibition of both GIRK activation and the downstream AKT and ERK phosphorylation.

In response to the reviewer's comment, the authors have conducted additional experiments to monitor the effect of Nb5 on agonist-induced G α signaling. G α_q signaling was monitored in HEK293T/17 cells transfected with the m3 muscarinic acetylcholine receptor (M3R) and using a Ca²⁺ sensor (CalFluxVTN). G α_s signaling was monitored in HEK293T/17 cells transfected with the dopamine D1 receptor (D1R) and using a cAMP sensor (Nluc-Epac-VV). The transfected cells were then stimulated with acetylcholine or dopamine to activate M3R or D1R, respectively. Real-time monitoring of the resultant responses clearly showed no effect of Nb5 on the elevation of intracellular Ca²⁺ induced by M3R-G α_q or on D1R-G α_s -induced cAMP production. These results show that Nb5 does not alter either G α_q or G α_s

signaling in a living cell. These findings also are consistent with the higher affinity of G α compared to Nb5 for the G $\beta\gamma$ dimer, supporting the inability of Nb5 to affect G α signaling.

These findings have been added in the revised manuscript as Fig 6f-j. Additionally in response to the reviewer's comment and for further clarification, the following text has been added to the results section of the revised manuscript.

Inserted text: "Additionally, Nb5 showed no significant effect on either the M3R-G α_q -induced intracellular Ca²⁺ elevation or D1R-G α_s -induced cAMP production. These results show that Nb5 does not alter the GTP-bound G α_q and G α_s -mediated signaling component in living cells. Overall, Nb5 serves as a dynamic scavenger of the G $\beta\gamma$ dimer and thereby causes partial inhibition of G $\beta\gamma$ -mediated signaling. The remaining Nb5-free G $\beta\gamma$ supports the canonical G α -GTP-mediated signaling that includes the GPCR-mediated GDP-GTP exchange from G α , and signaling termination by GTP hydrolysis and re-association of G α -GDP with G $\beta\gamma$. However at higher concentrations, Nb5 might affect the G α -GTP signaling component due to increased sequestration of G $\beta\gamma$ dimers."

2. In the introduction that authors state that, "While GPCRs serve as the largest class of drug-targeted membrane proteins, only a handful of GPCR-mediated signaling events have been targeted therapeutically with either small molecules or peptide modulators." While technically true, this statement is somewhat misleading since drugs that target GPCRs continue to comprise the largest set of drugs based on target class. Indeed, the reference used by the authors to support their statement refers to GPCRs as a "privileged target family." Consider rewording.

In response to the reviewer's suggestion, we have modified the above statement in the revised manuscript to the following text.

Modified text: "While GPCRs serve as the largest class of drug-targeted membrane proteins, producing the majority of FDA-approved drugs available on the market, only a handful of GPCR-mediated signaling events have been targeted therapeutically with either small molecule or peptide modulators²⁹."

3. The description of the methods for the SPR experiments is absent and should be added. Also, SPR is prone to several artefacts arising from the requirement to affix the ligand to the sensor chip. It is best practice to measure affinities using each partner as the affixed ligand with the expectation that measure affinities will not change. Unfortunately, most experimenters do not carry out this control and it would be unreasonable to require it here. However, had it been done, Gbeta/gamma would be the ligand and the setup would have easily allowed the direct measurement of Galpha-t binding to Gbeta/gamma. Instead, the authors reference a measure of the affinity between heterotrimer subunits based on fluorescent proteins at membranes. This measurement is likely confounded by the experimental design and the authors should find a more appropriate reference. Finally, it seems the authors missed a simple experiment to verify the direct competition of Galpha-t and Nb5 for binding to Gbeta/gamma. Pre-incubation of Gbeta/gamma with varying concentration of Galpha-t-GDP would be expected to reduce sensorgram responses; conversely, pre-incubation with Galpha-t-GDP in the presence of aluminum fluoride should have no effect on the sensorgrams.

The authors thank the reviewer for recognizing our omission of SPR methodology in the original manuscript. This has been remedied by the addition of surface plasmon resonance (SPR) procedures to the Methods section in the revised manuscript.

The reviewer makes an excellent point about the artifacts that arises due to ligand immobilization to the sensor chip. The authors agree that the best practice to measure affinities is by immobilizing each interacting partner separately. However, during the SPR experiments the G $\beta_1\gamma_1$ dimer bound non-specifically to the CM5 sensor chip resulting in a

high background signal that interfered with the data acquisition and analyses. This might have been caused by the pH change introduced during $G\beta_1\gamma_1$ coupling to the sensor chip. Therefore, Nb5 was used as an immobilized ligand and varied concentrations of $G\beta_1\gamma_1$ dimer were used for the binding analysis.

The reviewer makes a good point about the potential flaws in the experimental design used in the referred research article. In response to the reviewer's comment, we have added references that utilized fluorescence flow cytometry to determine the affinities between $G\alpha$ and $G\beta\gamma$ dimers^{30,31}.

The authors have tried a direct competition assay between $G\alpha_t$ and Nb5 for $G\beta_1\gamma_1$. However, the non-specific binding of the $G\beta_1\gamma_1$ dimer to the sensor chip was a primary concern and lead to a high background signal that interfered with the SPR data acquisition. We believe that the non-specific $G\beta_1\gamma_1$ binding to the CM5 sensor chip is due to either protein misfolding or aggregation caused during the immobilization of $G\beta_1\gamma_1$.

4. The crystal structure is of high quality. In light of the structural work, it probably makes sense to relegate the HDX-MS to supplemental information, especially since it is not clear why some regions experience increased exchange and other show decreased exchange. On a related note, Galpha-t is shown on the right hand panel of Figure 2e, but not on the left. The authors should state explicitly that Galpha-t is not shown on the left for the sake of clarity.

The authors appreciate the reviewer's comment. The HDX-MS analysis of the $G\beta_1\gamma_1$ -Nb5 complex maps the key interaction sites between $G\beta_1\gamma_1$ and Nb5 in solution rather than in a crystalline state. Therefore, the authors believe that the HDX-MS analyses should be included as a main figure. In response to the reviewer's suggestion, we have added the following text to the Figure 2e legend in the revised manuscript.

Inserted text: "The $G\alpha$ subunit was omitted from the left panel for clarity."

5. Gbeta1-4 are highly conserved and in contrast to Ggamma subunits rarely, if ever, dictate preferential activation of effectors. Therefore, given that Nb5 binds exclusively to Gbeta1, it is not remarkable that "Nb5 responds to all combinations of beta- and gamma- subtypes." Perhaps reword and remove multiple sequence alignment (Fig. 4a).

The authors concur with the reviewer that $G\beta$ rarely dictates preferential activation of effectors. The binding of Nb5 to $G\beta_1$ increases the likelihood that it also binds $G\beta_2$ -4 due to their high sequence identity. In response to the reviewer's comment, the word "remarkable" has been removed and sentence has been rephrased in the revised manuscript as below.

Modified text: "Nb5 responds to all combinations of β - and γ -subtypes, indicating its broad applicability to various cell types."

Fig 4a has also been revised to include the $G\beta_5$ sequence to demonstrate the key residues that are necessary for Nb5 binding to $G\beta$ subunits.

6. Figure 5j is somewhat confusing. Is there a better way to indicate that the GIRK channel does not open? Right now, it appears that the channel is cycling potassium around inside the cell. Similarly, "Nb5 mediated GIRK channel activation" has a pointer on the outside of the cell. Perhaps direct the pointer closer to the action?

The authors appreciate this comment. Fig. 5j (which is Fig. 5m in the revised manuscript) has been revised to clarify the blockage of potassium entry into the GIRK channels and the pointer has been placed closer to the site of action.

References

1. Bonacci, T.M., *et al.* Differential targeting of Gbetagamma-subunit signaling with small molecules. *Science* **312**, 443-446 (2006).
2. Koch, W.J., Hawes, B.E., Inglese, J., Luttrell, L.M. & Lefkowitz, R.J. Cellular expression of the carboxyl terminus of a G protein-coupled receptor kinase attenuates G beta gamma-mediated signaling. *The Journal of biological chemistry* **269**, 6193-6197 (1994).
3. Goubaeva, F., *et al.* Stimulation of cellular signaling and G protein subunit dissociation by G protein betagamma subunit-binding peptides. *The Journal of biological chemistry* **278**, 19634-19641 (2003).
4. Scott, J.K., *et al.* Evidence that a protein-protein interaction 'hot spot' on heterotrimeric G protein betagamma subunits is used for recognition of a subclass of effectors. *The EMBO journal* **20**, 767-776 (2001).
5. Bonacci, T.M., Ghosh, M., Malik, S. & Smrcka, A.V. Regulatory interactions between the amino terminus of G-protein betagamma subunits and the catalytic domain of phospholipase Cbeta2. *The Journal of biological chemistry* **280**, 10174-10181 (2005).
6. Doolittle, J.M. & Gomez, S.M. Structural similarity-based predictions of protein interactions between HIV-1 and Homo sapiens. *Virology journal* **7**, 82 (2010).
7. Doolittle, J.M. & Gomez, S.M. Mapping protein interactions between Dengue virus and its human and insect hosts. *PLoS Negl Trop Dis* **5**, e954 (2011).
8. Rana, J., *et al.* Deciphering the host-pathogen protein interface in chikungunya virus-mediated sickness. *Archives of virology* **158**, 1159-1172 (2013).
9. Davis, F.P., Barkan, D.T., Eswar, N., McKerrow, J.H. & Sali, A. Host pathogen protein interactions predicted by comparative modeling. *Protein science : a publication of the Protein Society* **16**, 2585-2596 (2007).
10. Holm, L. & Rosenstrom, P. Dali server: conservation mapping in 3D. *Nucleic acids research* **38**, W545-549 (2010).
11. Holm, L. & Sander, C. Protein structure comparison by alignment of distance matrices. *Journal of molecular biology* **233**, 123-138 (1993).
12. Holm, L., Kaariainen, S., Rosenstrom, P. & Schenkel, A. Searching protein structure databases with DaliLite v.3. *Bioinformatics* **24**, 2780-2781 (2008).
13. Koch, W.J., Hawes, B.E., Allen, L.F. & Lefkowitz, R.J. Direct evidence that Gi-coupled receptor stimulation of mitogen-activated protein kinase is mediated by G beta gamma activation of p21ras. *Proceedings of the National Academy of Sciences of the United States of America* **91**, 12706-12710 (1994).
14. Neptune, E.R. & Bourne, H.R. Receptors induce chemotaxis by releasing the betagamma subunit of Gi, not by activating Gq or Gs. *Proceedings of the National Academy of Sciences of the United States of America* **94**, 14489-14494 (1997).
15. Khan, S.M., *et al.* The expanding roles of Gbetagamma subunits in G protein-coupled receptor signaling and drug action. *Pharmacol Rev* **65**, 545-577 (2013).
16. Smrcka, A.V. G protein betagamma subunits: central mediators of G protein-coupled receptor signaling. *Cellular and molecular life sciences : CMLS* **65**, 2191-2214 (2008).
17. Lin, Y. & Smrcka, A.V. Understanding molecular recognition by G protein betagamma subunits on the path to pharmacological targeting. *Molecular pharmacology* **80**, 551-557 (2011).
18. Ghosh, E., *et al.* A synthetic intrabody-based selective and generic inhibitor of GPCR endocytosis. *Nat Nanotechnol* **12**, 1190-1198 (2017).
19. Jahnichen, S., *et al.* CXCR4 nanobodies (VHH-based single variable domains) potently inhibit chemotaxis and HIV-1 replication and mobilize stem cells. *Proceedings of the National Academy of Sciences of the United States of America* **107**, 20565-20570 (2010).
20. Ogura, M., *et al.* Multicenter phase II study of mogamulizumab (KW-0761), a defucosylated anti-cc chemokine receptor 4 antibody, in patients with relapsed peripheral T-cell lymphoma and cutaneous T-cell lymphoma. *Journal of clinical*

- oncology : official journal of the American Society of Clinical Oncology* **32**, 1157-1163 (2014).
21. Kovaleva, M., Ferguson, L., Steven, J., Porter, A. & Barelle, C. Shark variable new antigen receptor biologics - a novel technology platform for therapeutic drug development. *Expert Opin Biol Ther* **14**, 1527-1539 (2014).
 22. Griffiths, K., *et al.* Shark Variable New Antigen Receptor (VNAR) Single Domain Antibody Fragments: Stability and Diagnostic Applications. *Antibodies* **2**, 66-81 (2013).
 23. Griffiths, K., *et al.* i-bodies, Human Single Domain Antibodies That Antagonize Chemokine Receptor CXCR4. *The Journal of biological chemistry* **291**, 12641-12657 (2016).
 24. Gulati, S., *et al.* Photocyclic behavior of rhodopsin induced by an atypical isomerization mechanism. *Proceedings of the National Academy of Sciences* **114**, E2608-E2615 (2017).
 25. Alexander, N.S., *et al.* Complex binding pathways determine the regeneration of mammalian green cone opsin with a locked retinal analogue. *The Journal of biological chemistry* (2017).
 26. Evans, P.R. & Murshudov, G.N. How good are my data and what is the resolution? *Acta crystallographica. Section D, Biological crystallography* **69**, 1204-1214 (2013).
 27. Diederichs, K. & Karplus, P.A. Better models by discarding data? *Acta crystallographica. Section D, Biological crystallography* **69**, 1215-1222 (2013).
 28. Karplus, P.A. & Diederichs, K. Linking crystallographic model and data quality. *Science* **336**, 1030-1033 (2012).
 29. Santos, R., *et al.* A comprehensive map of molecular drug targets. *Nature reviews. Drug discovery* **16**, 19-34 (2017).
 30. Smrcka, A.V., *et al.* NMR analysis of G-protein betagamma subunit complexes reveals a dynamic G(alpha)-Gbetagamma subunit interface and multiple protein recognition modes. *Proceedings of the National Academy of Sciences of the United States of America* **107**, 639-644 (2010).
 31. Sarvazyan, N.A., Remmers, A.E. & Neubig, R.R. Determinants of gi1alpha and beta gamma binding. Measuring high affinity interactions in a lipid environment using flow cytometry. *The Journal of biological chemistry* **273**, 7934-7940 (1998).

Reviewers' comments:

Reviewer #1 (Remarks to the Author):

The revised version of the paper satisfactorily addresses my concerns. The discussion of mechanisms and comparison with existing G β γ -binding proteins with similar effects has been improved and extended. New experiments shown in Fig. 6 as well as additions made to some other figures strengthen the conclusions, emphasizing the action of Nb5 as selective G β γ inhibitor with little effect on Ga signaling in living cells upon mild expression of Nb5. Overall, it is a thorough, well-controlled study which deserves publication in its present form. It will be of interest to a wide community of scientists interested in G protein signaling.

Reviewer #2 (Remarks to the Author):

The authors properly addressed most of the issues raised by reviewers. However, there are still a few concerns.

1. The authors claim "These results show that Nb5 does not alter the GTP-bound G α q and Gas-mediated signaling component in living cells.... However at higher concentrations, Nb5 might affect the Ga-GTP signaling component due to increased sequestration of G β γ dimers." Please provide supporting data for this statement. The authors may transfect cells with increasing amount of Nb5 DNA, measure Nb5 expression levels (probably through WB), perform functional assay (i.e. inhibition of Ga signaling), and correlate the functional effects with Nb expression level.

2. Please provide appropriate references for the statement in the response to the reviewers' comments "This SDS-polyacrylamide artifact has been previously observed with several post-translationally modified proteins in our laboratory and by others in the field."

Reviewer #3 (Remarks to the Author):

The authors have adequately addressed the major and minor points raised. Additional experiments have been performed and controls, quantification of blots as suggested have been added. These additions have improved the manuscript, which is an important contribution to the field.

Reviewer #5 (Remarks to the Author):

Thanks to authors to their hard work in improving their study. With such a significant revision, there are a two outstanding major points that arose, plus two minor points.

Major points:

First, the authors indicated "This study serves as the first comprehensive structure-function study communicating the ability of a nanobody to modulate a GPCR-mediated G β γ -signaling event in any cell type.... Nb5 might serve as a better template for structural activity studies for to Gb selectivity." If that is the case, then Nb5 can be use as a research tool which binds to all Gb subtypes and inhibits all Gbg signaling in different levels. The authors also stated that "Despite its inhibitory effect on G β γ -mediated signaling, Nb5 has no effect on GTP bound G α q and Gas-mediated signaling events in the environment of a living cell". Are they also expecting no effect on the visual system when they blocked the Gb1 in in vivo or in the animal model?

Second, the authors suggest the therapeutic potential of the Nb5. There are many Gbg-target couplings involved in diseases and disruption of these interactions has been shown to be of

potential therapeutic benefit. Various studies have also shown that blocking Gbg protein-protein interactions is an effective approach to preventing heart failure (Rockman et al., 1998), arterial restenosis (Iaccarino et al., 1999), hypertension (Koch et al., 1995), drug addiction (Yao et al., 2003), cancer metastasis (Müller et al., 2001), and prostate cancer (Bookout et al., 2003) in animal models.

However, to consider Gbg as a viable therapeutic target there are several issues that must be overcome. First, Gbg plays a central role in the function of all GPCR signaling systems. Genetic deletion of the cellular complement of Gbg's completely disrupts all GPCR signaling (Hwang et al., 2005). Thus, targeting of specific Gbg protein-protein interactions without ablating general Gbg function is a possible approach to this. Another major problem is that Gbg expression is nearly ubiquitous, so blocking all Gbg functions might have unwanted side effects.

Or to put it another way, the discussion of this Nb as a potential drug candidate is misleading because this Nb ablates general Gbg function rather than selectively targets specific Gbg-effector coupling. This would have major untoward physiological effects. In contrast, M119, M119K and Gallein only target the Gb-GRK2 complex, which makes them useful therapies.

Remaining minor concerns:

The authors provided additional information about Gt activation rate. "The authors appreciate the reviewer's insightful query. Conditions for the Gt activation assay were chosen such that the Gt activation rate was the same as that determined by GTPγS induced complex dissociation. The Gt activation rates were calculated based on the exponential slopes of the representative spectra 24,25. Based on these analyses the initial Gt activation rates of Rh* (k_{initial}) in the presence of Nb5 and in the dark are $0.0060 \pm 0.00011 \text{ s}^{-1}$ and $0.001 \pm 0.00010 \text{ s}^{-1}$, respectively. These rates are 16.9 % and 2.8 % of the initial Gt activation rate of Rh* ($k_{\text{initial}} = 0.0353 \pm 0.00485 \text{ s}^{-1}$). The initial Gt activation rates of these samples have been quantified in Fig. 1e for comparison. Additionally, Fig 1d shows the representative raw data for the Gt activation assays."

Transducin basal activation is extensively studied and shows extremely slow nucleotide exchange rate. Without active rhodopsin, the half time for activation of transducin is more than 500 minutes. (Jager et al., 1996, Marin et al., 2001 and 2002, Oldham et al., 2006, Kaya et al., 2011) which is not consistent and significantly slower than what is proposed in this study. This still needs to be clarified. It would be helpful for the readers to compare exchange rates by providing raw data rather than what is shown in Figure 1d, which is normalized.

In Figure 6g, I and J it should be basal not basel.

Reviewer #6 (Remarks to the Author):

The authors have thoughtfully addressed the concerns of this reviewer.

Response to Reviewers' comments:

We thank the reviewers again for their careful assessment of our work. We have addressed the reviewer's comments below, and have incorporated their suggestions in the revised manuscript. Reviewers' comments are in black text and our responses are in blue text.

Reviewers' comments:

Reviewer #1

The revised version of the paper satisfactorily addresses my concerns. The discussion of mechanisms and comparison with existing G $\beta\gamma$ -binding proteins with similar effects has been improved and extended. New experiments shown in Fig. 6 as well as additions made to some other figures strengthen the conclusions, emphasizing the action of Nb5 as selective G $\beta\gamma$ inhibitor with little effect on G α signaling in living cells upon mild expression of Nb5. Overall, it is a thorough, well-controlled study which deserves publication in its present form. It will be of interest to a wide community of scientists interested in G protein signaling.

We are pleased that we have satisfactorily addressed Reviewer #1's concerns.

Reviewer #2

The authors properly addressed most of the issues raised by reviewers. However, there are still a few concerns.

1. The authors claim "These results show that Nb5 does not alter the GTP-bound G α_q and G α_s -mediated signaling component in living cells.... However at higher concentrations, Nb5 might affect the G α -GTP signaling component due to increased sequestration of G $\beta\gamma$ dimers." Please provide supporting data for this statement. The authors may transfect cells with increasing amount of Nb5 DNA, measure Nb5 expression levels (probably through WB), perform functional assay (i.e. inhibition of G α signaling), and correlate the functional effects with Nb expression level.

The authors appreciate the reviewer's comment. The supporting data to this comment comes from Fig. 1d-e, where 2 molar excess concentration of Nb5 resulted in reduced G $_t$ activation rate of photoactivated rhodopsin. Due to a non-linear relationship between the cell transfection efficiency and concentration of DNA, it's technically challenging to quantify and control the expression of Nb5 in cells. Hence, we had conducted additional experiments to further confirm the G α -independent nature of Nb5. Real-time monitoring of G α_q and G α_s signaling clearly showed no effect of Nb5 on M3R-G α_q -induced intracellular Ca $^{2+}$ elevation and on D1R-G α_s -induced cAMP production. These results suggest that Nb5 does not alter either G α_q or G α_s signaling in a living cell.

For further clarity, we have modified the following text in the discussion section of the revised manuscript.

Modified text: "However at higher concentrations, Nb5 might affect the G α -GTP signaling component due to increased sequestration of G $\beta\gamma$ dimers (Fig. 1d,e)."

2. Please provide appropriate references for the statement in the response to the reviewers' comments "This SDS-polyacrylamide artifact has been previously observed with several post-translationally modified proteins in our laboratory and by others in the field."

The authors appreciate the reviewer's comment. The references for the observed SDS-polyacrylamide artifact have been added in the revised manuscript.

Modified text: "This SDS-polyacrylamide artifact has been previously observed with several post-translationally modified proteins in our laboratory and by others in the field¹⁻⁴."

Reviewer #3

The authors have adequately addressed the major and minor points raised. Additional experiments have been performed and controls, quantification of blots as suggested have been added. These additions have improved the manuscript, which is an important contribution to the field.

We are pleased that we have satisfactorily addressed Reviewer #3's concerns.

Reviewer #5

Thanks to authors to their hard work in improving their study. With such a significant revision, there are a two outstanding major points that arose, plus two minor points.

Major points:

First, the authors indicated "This study serves as the first comprehensive structure-function study communicating the ability of a nanobody to modulate a GPCR-mediated Gβγ-signaling event in any cell type.... Nb5 might serve as a better template for structural activity studies for to Gb selectivity." If that is the case, then Nb5 can be use as a research tool which binds to all Gb subtypes and inhibits all Gbg signaling in different levels. The authors also stated that "Despite its inhibitory effect on Gβγ-mediated signaling, Nb5 has no effect on GTP bound Gαq and Gαs-mediated signaling events in the environment of a living cell". Are they also expecting no effect on the visual system when they blocked the Gb1 in in vivo or in the animal model?

The authors appreciate the reviewer's insightful query. Currently, the authors are working on generating Nb5 transgenic mice to assess the effect of Nb5 on the visual photo-transduction pathway. However, significant research studies need to be conducted to confirm Nb5's effect on vision. Therefore, while being very interesting and a near future publication, the authors and the editor believe that the *in vivo* studies are beyond the scope of the current manuscript.

Second, the authors suggest the therapeutic potential of the Nb5. There are many Gbg-target couplings involved in diseases and disruption of these interactions has been shown to be of potential therapeutic benefit. Various studies have also shown that blocking Gbg protein-protein interactions is an effective approach to preventing heart failure (Rockman et al., 1998), arterial restenosis (Iaccarino et al., 1999), hypertension (Koch et al., 1995), drug addiction (Yao et al., 2003), cancer metastasis (Müller et al., 2001), and prostate cancer (Bookout et al., 2003) in animal models.

However, to consider Gbg as a viable therapeutic target there are several issues that must be overcome. First, Gbg plays a central role in the function of all GPCR signaling systems. Genetic deletion of the cellular complement of Gbg's completely disrupts all GPCR signaling (Hwang et al., 2005). Thus, targeting of specific Gbg protein-protein interactions without ablating general Gbg function is a possible approach to this. Another major problem is that Gbg expression is nearly ubiquitous, so blocking all Gbg functions might have unwanted side effects.

Or to put it another way, the discussion of this Nb as a potential drug candidate is misleading because this Nb ablates general Gbg function rather than selectively targets specific Gbg-effector

coupling. This would have major untoward physiological effects. In contrast, M119, M119K and Gallein only target the Gb-GRK2 complex, which makes them useful therapies.

The authors concur with the reviewer that Nb5 might have several unwanted side-effects due to its ability to modulate general G $\beta\gamma$ signaling. In response to reviewer's comment and for further clarity, we have added the following text in the discussion section of the revised manuscript.

Added text: "Unlike nanobodies, the M119 class of inhibitors can be delivered orally and intraperitoneally to modulate G $\beta\gamma$ –dependent pathways, which admittedly takes them one step closer to clinical applications. In addition, the M119 class of inhibitors has the advantage that they specifically target the formation of G $\beta\gamma$ -GRK2 complexes^{5,6}. Challenges remain for the development of nanobody-based clinical therapies due to the dependence of this approach on efficient gene transduction methods. Moreover, the ability of Nb5 to affect general G $\beta\gamma$ signaling raises concerns for possible off-target effects when pursued therapeutically⁷. Nevertheless, there may be applications for Nb5-based treatments where the beneficial effects outweigh the side effects."

Remaining minor concerns:

The authors provided additional information about Gt activation rate. "The authors appreciate the reviewer's insightful query. Conditions for the Gt activation assay were chosen such that the Gt activation rate was the same as that determined by GTP γ S induced complex dissociation. The Gt activation rates were calculated based on the exponential slopes of the representative spectra 24,25. Based on these analyses the initial Gt activation rates of Rh* (k_{initial}) in the presence of Nb5 and in the dark are 0.0060 ± 0.00011 s⁻¹ and 0.001 ± 0.00010 s⁻¹, respectively. These rates are 16.9 % and 2.8 % of the initial Gt activation rate of Rh* (k_{initial} = 0.0353 ± 0.00485 s⁻¹). The initial Gt activation rates of these samples have been quantified in Fig. 1e for comparison. Additionally, Fig 1d shows the representative raw data for the Gt activation assays."

Transducin basal activation is extensively studied and shows extremely slow nucleotide exchange rate. Without active rhodopsin, the half time for activation of transducin is more than 500 minutes. (Jager et al., 1996, Marin et al., 2001 and 2002, Oldham et al., 2006, Kaya et al., 2011) which is not consistent and significantly slower than what is proposed in this study. This still needs to be clarified. It would be helpful for the readers to compare exchange rates by providing raw data rather than what is shown in Figure 1d, which is normalized.

The reviewer raises an excellent point that the reported value of G_t activation rate in dark is higher as compared to the rates published previously. The authors have re-evaluated the G_t activation rate of rhodopsin in the dark and found the new k_{initial} of 0.000038 ± 0.000001 s⁻¹ (~438 minutes). This discrepancy in the k_{initial} was due to the erroneous exponential fitting of a near-linear G_t activation profile of rhodopsin in the dark. This correction has been incorporated in Figure 1 of the revised manuscript.

Additionally in response to the reviewer's comment, Figure 1d has been updated to show the raw data of the G_t activation assay.

In Figure 6g, I and J it should be basal not basel.

The authors thank the reviewer for recognizing this error. Correction has been made in Figure 6g-j of the revised manuscript.

Reviewer #6

The authors have thoughtfully addressed the concerns of this reviewer.

We are pleased that we have satisfactorily addressed Reviewer #6's concerns.

References

1. Evans, D.R., Romero, J.K. & Westoby, M. Concentration of proteins and removal of solutes. *Methods in enzymology* **463**, 97-120 (2009).
2. Drabik, A., Bodzoń-Kułakowska, A. & Silberring, J. 7 - Gel Electrophoresis. in *Proteomic Profiling and Analytical Chemistry (Second Edition)* 115-143 (Elsevier, Boston, 2016).
3. See, Y.P., Olley, P.M. & Jackowski, G. The effects of high salt concentrations in the samples on molecular weight determination in sodium dodecyl sulfate polyacrylamide gel electrophoresis. *ELECTROPHORESIS* **6**, 382-387 (1985).
4. Zhao, M., Sun, L., Fu, X. & Gong, X. Influence of Ionic Strength, pH, and SDS Concentration on Subunit Analysis of Phycoerythrins by SDS-PAGE. *Applied Biochemistry and Biotechnology* **162**, 1065-1079 (2010).
5. Bonacci, T.M., *et al.* Differential targeting of Gbetagamma-subunit signaling with small molecules. *Science* **312**, 443-446 (2006).
6. Casey, L.M., *et al.* Small molecule disruption of G beta gamma signaling inhibits the progression of heart failure. *Circulation research* **107**, 532-539 (2010).
7. Hwang, J.I., Choi, S., Fraser, I.D., Chang, M.S. & Simon, M.I. Silencing the expression of multiple Gbeta-subunits eliminates signaling mediated by all four families of G proteins. *Proceedings of the National Academy of Sciences of the United States of America* **102**, 9493-9498 (2005).

REVIEWERS' COMMENTS:

Reviewer #2 (Remarks to the Author):

The authors have adequately addressed the concerns raised.

Reviewer #5 (Remarks to the Author):

The revision addressed outstanding concerns.